# HYPERBOLIC ASSOCIATIVE MEMORY NETWORKS

## ABSTRACT

Despite the widespread success of associative memory models, such as modern Hopfield networks, in various domains, related research has still been confined to Euclidean or kernel-induced spaces. When the state space is restricted to Euclidean geometry, it becomes challenging to accurately capture hierarchical structures in the data. For instance, even in high-dimensional Euclidean space, arbitrary tree structures cannot be embedded with minimal distortion; in many tasks requiring the processing of hierarchical data, Hopfield networks based on Euclidean representations tend to introduce bias and distortion in semantic relations. To address this issue, we propose extending modern Hopfield retrieval to hyperbolic space. Specifically, we map query and memory vectors from Euclidean space to hyperbolic space via exponential maps and define an energy function based on the Minkowski inner product, with a solid theoretical foundation. The retrieval process uses Riemannian manifold optimization, combining curvature-aware gradients with exponential maps to ensure that the optimization trajectory remains on the manifold and produces stable updates. Our central view can be stated as a hierarchy-sensitivity hypothesis: when the data exhibit clear and deeper hierarchical structure, hyperbolic geometry brings statistically significant improvements; when the hierarchy is weak or only shallow, performance shows no significant difference from Euclidean modern Hopfield networks. We validate this through depth-controlled comparisons and cross-level consistency metrics, and the empirical results are consistent with the hypothesis. Accordingly, the proposed hyperbolic associative memory can serve as a plug-and-play general memory module embedded into task architectures that require hierarchical understanding, for storing and retrieving raw inputs, intermediate representations, or learned prototypes, and explicitly exploiting hierarchical information. Our method is formulated in a model-agnostic manner and applies to any hyperbolic model with constant negative curvature. We instantiate it with the Poincaré ball for experiments.

## 1 INTRODUCTION

Associative memory models, have played a crucial role in enabling neural systems to retrieve stored patterns from partial or noisy inputs. In this domain, classical Hopfield network models Hopfield (1982); Amari (1972) store memories as fixed-point attractor states in an energy landscape, leveraging Hebbian learning to recall full patterns from partial input cues through a recurrent architecture. More recently, Modern Hopfield Networks (MHNs) Vaswani et al. (2017); Widrich et al. (2020) have introduced continuous relaxations of the original formulation, theoretically achieving exponential storage capacity with respect to the number of neurons Krotov & Hopfield (2016); Demircigil et al. (2017); Ramsauer et al. (2021) and reigniting interest in associative memory mechanisms. Recent research has significantly extended the MHN framework in several directions. For example, kernelized MHNs Wu et al. (2024) introduce learnable kernels that map raw memories into a Hilbert feature space, inducing a new distance structure to capture similarity between queries and memories, thereby improving memory capacity and robustness. Sparse and structured MHNs Hu et al. (2023); Santos et al. (2024) introduce sparse entropic regularizers or Fenchel–Young losses to enhance efficiency and scalability. Latent-structured MHNs Li et al. (2025) embed continuous attractor dynamics into autoencoder latent spaces to improve semantic association and episodic memory. However, our focus is not on kernel functions or regularization mechanisms, which are still carried out in Euclidean geometry with zero curvature. Modern Hopfield networks are typically used in deep learning as energy-based retrieval layers or attention-like modules, and we follow this

line of work: we provide a geometric extension of existing MHN methods by moving the Hopfield energy and dynamics from Euclidean or Hilbert spaces to negatively curved manifolds and studying how this geometric change improves hierarchical retrieval. At the same time, our goal is to design a differentiable associative memory module for machine learning systems, rather than to construct a biologically realistic model of hippocampal or cortical circuits or to emphasize biological interpretability.

Although representing data in Euclidean space $\mathbb{R}^n$ has traditionally been the standard choice because of its computational convenience, recent studies have shown that this approach faces fundamental limitations when applied to complex data types Ganea et al. (2018). Many real-world datasets, particularly those involving graphs, taxonomies, or hierarchical relationships, exhibit an inherently non-Euclidean latent structure Bronstein et al. (2017). In such cases, Euclidean embeddings often struggle to faithfully preserve semantic proximity and hierarchical organization Gromov (1987). For example, arbitrary tree structures cannot be embedded with arbitrarily low distortion even in high-dimensional Euclidean spaces Linial et al. (1995), whereas hyperbolic spaces, owing to their exponential growth of volume, can naturally accommodate such structures even in low dimensions Krioukov et al. (2010); Nickel & Kiela (2017). Thus, in tasks of this kind (e.g. hierarchical classification, hierarchical clustering, knowledge graph completion, and graph/image/text classification or retrieval with hierarchical labels), applying associative memory mechanisms purely within Euclidean geometry may distort the underlying structural information during memory retrieval. These observations motivate us to embed the associative memory process into hyperbolic space, changing the geometry of the associative memory layer so as to represent hierarchical information.

To address these limitations, we introduce Hyperbolic Associative Memory Networks (HAMNs), the first framework that embeds modern associative memory into hyperbolic space. Specifically, we first apply exponential maps to transform query and memory vectors from Euclidean space to hyperbolic space (a constant–negative–curvature manifold), thereby leveraging the natural capacity of hyperbolic geometry to model hierarchical structures. On top of these mapped representations, we define a principled energy function using the Minkowski inner product to capture similarity relations in hyperbolic geometry. During memory retrieval, we incorporate curvature-aware Riemannian optimization Bonnabel (2013) with exponential-map updates to ensure that each update step follows the tangent direction of the hyperbolic manifold and remains strictly within hyperbolic space. On the theoretical side, we prove that, under fixed state dimension and minimum geodesic separation, formulating the Hopfield energy on a negatively curved manifold allows the number of well-separated recall wells to grow exponentially with the effective hyperbolic radius (i.e., hierarchy depth), whereas in the Euclidean case the analogous packing bound grows only polynomially with the radius. In our experiments, we instantiate the method with the Poincaré ball Nickel & Kiela (2017) due to implementation maturity, while the derivations apply equally to other hyperbolic models.

With this design, we propose a hierarchy-sensitivity hypothesis that does not presuppose pronounced hierarchical structure in all tasks or datasets; when hierarchical/tree structure does exist and is sufficiently deep, HAMNs demonstrate a stronger ability to understand, preserve, and retrieve hierarchical relations, whereas when the hierarchy is weak or essentially absent, their performance is largely on par with Euclidean MHNs. To validate this hypothesis, we conduct a systematic evaluation by controlling hierarchy depth and reporting metrics such as cross-level consistency, and the empirical results are consistent with the hypothesis. Our main contributions are summarized as follows:

- We design Hyperbolic Associative Memory Networks (HAMNs), a plug-and-play, model-agnostic associative memory module operating in hyperbolic space that can be dropped into architectures requiring hierarchical understanding to store and retrieve raw inputs, intermediate representations, or learned prototypes, explicitly leveraging hierarchical structure.

- We design a principled energy function and optimization mechanism based on hyperbolic geometry, ensuring a stable and efficient memory update process.

- With hierarchy depth controlled and cross-level consistency measured, our method achieves clear benefits on hierarchical data and competitive flat/shallow results, outperforming Euclidean Hopfield networks at representing complex structures.

## 2 PRELIMINARIES

### 2.1 MODERN HOPFIELD NETWORKS

Modern Hopfield Networks (MHNs) Demircigil et al. (2017); Ramsauer et al. (2021) extend classical associative memory models by introducing continuous state representations and modifying the energy function landscape. This modification significantly enhances the storage capacity and enables the network to retrieve stored patterns through continuous optimization dynamics.

Given a set of $N$ memory patterns $\{\xi_n \in \mathbb{R}^K\}_{n=1}^N$, organized as a memory matrix $\Xi \in \mathbb{R}^{N \times K}$, and a query state vector $s \in \mathbb{R}^K$, the energy function of MHNs is formulated as:

$$E(s, \Xi; \beta) = F_\beta\left(f_{\text{sim}}(\{\xi_n\}, s)\right) + \frac{1}{2}s^\top s \tag{1}$$

where the similarity is defined as $f_{\text{sim}}(\{\xi_n\}, s) = \{\langle \xi_n, s \rangle\}_{n=1}^N$ (dot product between $s$ and each memory), and $F_\beta(\cdot)$ is the log-sum-exponential (LSE) function: $F_\beta(z) = -\frac{1}{\beta}\log\left(\sum_{n=1}^N \exp(\beta z_n)\right)$, with $\beta > 0$ controlling the sharpness. For consistency with the rest of the paper, we will use $\theta$ as the temperature (i.e., $\theta \equiv \beta$). The associative retrieval process minimizes the energy iteratively as:

$$s^{(t+1)} = \Xi \, \text{softmax}\left(\beta \Xi^\top s^{(t)}\right) = \sum_{n=1}^N \xi_n \frac{\exp\left(\beta \xi_n^\top s^{(t)}\right)}{\sum_{n'=1}^N \exp\left(\beta \xi_{n'}^\top s^{(t)}\right)} \tag{2}$$

Under mild conditions, the update rule monotonically decreases the system energy and converges to a (meta-)stable fixed point Ramsauer et al. (2021); Widrich et al. (2020). This framework thus enables efficient pattern retrieval even from noisy or partial cues. Eq. (2) is equivalent to the readout of single-head attention, with keys=values $= X$ and query $s^{(t)}$, hence an MHN can be viewed as an energy-based realization of attention.

### 2.2 HYPERBOLIC MANIFOLDS: CONCEPTS AND INTUITION

A **hyperbolic manifold** is a Riemannian manifold $(\mathcal{M}, g)$ Cannon et al. (1997) of *constant negative curvature* $-c < 0$. Geometrically it exhibits:

Triangle angle sum $< \pi$; geodesics diverge; ball volume grows **exponentially** with radius (matching the exponential branching of trees/hierarchies). Any two points are typically joined by a unique geodesic; distances near the boundary are "magnified". Multiple *isometric* coordinate models (Poincaré ball/upper-half plane, Klein, Lorentz hyperboloid) that are mutually isometric and differ only by parametrization.

Our theory and algorithm rely only on *model-agnostic* primitives; a concrete instantiation (e.g. Poincaré ball) is deferred to implementation details.

#### 2.2.1 PRIMITIVE 1: EXPONENTIAL/LOGARITHMIC MAPS

For any $p \in \mathcal{M}$, the exponential and logarithmic maps

$$\exp_p^c : T_p\mathcal{M} \to \mathcal{M}, \qquad \log_p^c : \mathcal{M} \to T_p\mathcal{M}$$

move along geodesics and back: given a tangent vector $v \in T_p\mathcal{M}$, $\exp_p^c(v)$ is the point reached at unit time on the geodesic starting from $p$ with initial velocity $v$, while $\log_p^c$ is its local inverse. They satisfy $\exp_p^c(0) = p$ and $d(\exp_p^c)_0 = \text{id}$, so for small $v$ we have $\exp_p^c(v) \approx p + v$ in local coordinates. In practice, these maps form the bridge between Euclidean vector spaces and the manifold: we encode queries and memories in $T_p\mathcal{M}$, map them to $\mathcal{M}$ with $\exp_p^c$, and during retrieval compute a descent direction in $T_{\xi^{(t)}}\mathcal{M}$ (via $\log_{\xi^{(t)}}^c$ and the Riemannian gradient) before exponentiating it back onto the manifold.

#### 2.2.2 PRIMITIVE 2: GEODESIC DISTANCE

**Definition.** $d_\mathcal{M}(x, y)$ is the length of the shortest geodesic between $x$ and $y$ induced by $g$.

**Hierarchy intuition.** Radial distance grows roughly linearly with radius, but near the boundary any fixed Euclidean displacement is exponentially magnified, naturally separating differences in hierarchical depth (see the toy example 2.2.5).

### 2.2.3 PRIMITIVE 3: "MINKOWSKI-LIKE" INNER PRODUCT

We use $\langle x, y \rangle_M := -\cosh(d_{\mathcal{M}}(x, y))$ as the similarity in hyperbolic space. Key properties:

It enjoys two key properties: *(i) Monotonicity* — it is decreasing in $d_{\mathcal{M}}(x, y)$, equals $-1$ at $x = y$, and tends to $-\infty$ as distance increases; and *(ii) Equivalence* — in the Lorentz model this similarity is a monotone function of the Minkowski bilinear form, coinciding numerically with it when the curvature is $-1$, so the construction is model-agnostic and invariant under hyperbolic isometries.

### 2.2.4 ISOMETRY INVARIANCE

If $\phi : (\mathcal{M}, g) \to (\mathcal{M}', g')$ is an isometry, then $d_{\mathcal{M}}(x, y) = d_{\mathcal{M}'}(\phi(x), \phi(y)) \quad \Rightarrow \quad \langle x, y \rangle_M = \langle \phi(x), \phi(y) \rangle_{M'}$. Hence, any energy and update constructed from $d_{\mathcal{M}}$ and $-\cosh d_{\mathcal{M}}$ are *model-equivalent* across hyperbolic realizations (Poincaré ball, Lorentz, Klein, upper-half plane, etc.).

### 2.2.5 WHY HYPERBOLIC FOR HIERARCHIES? A TOY EXAMPLE

**Hierarchical data** We call a dataset $\mathcal{D} = \{(x_i, y_i)\}_{i=1}^{N}$ *hierarchical* if its label space $\mathcal{Y}$ is equipped with a directed acyclic graph (DAG) $(\mathcal{Y}, \mathcal{E})$ or a rooted tree, where each node $y \in \mathcal{Y}$ has a depth $\mathrm{depth}(y)$ and ancestors $\mathrm{Anc}(y)$. Evaluation is required to be *hierarchy-aware*, in the sense that it depends on the graph distance between predicted and true labels, e.g. hierarchical accuracy at multiple levels (coarse/fine), shortest-path distance in the label DAG, or correlation between tree distance and representation distance (such as cophenetic correlation (Sokal & Rohlf, 1962)).

*For simplicity in this toy example we take curvature $-c = -1$ (i.e., $c = 1$) on the Poincaré ball.*

**Euclidean space "flattens" hierarchies.** Embed a tree of depth $L$ into the plane: all nodes at level $\ell$ lie on the same-radius circle. As $L$ grows, the outermost nodes crowd the same ring and leaf–leaf distances are governed almost only by the angular gap and become very similar, so leaves from different major branches appear "about equally far" and hierarchical information is weakened.

**Hyperbolic space "pulls apart" hierarchies.** Keep angles uniform, but encode depth by hyperbolic radius:$\rho_\ell = \tanh(\alpha \ell / 2), \quad \alpha > 0$. Since $d(0, x) = 2 \operatorname{artanh} \|x\|$, any level-$\ell$ node satisfies:$d_{\mathbb{D}}(0, x_\ell) = \alpha \ell$

i.e., each additional level increases the *radial hyperbolic distance* by (approximately) a fixed amount, so different levels separate naturally. Moreover, because the metric is "magnified" near the boundary, two points on the *same level* but from *different major branches* acquire a much larger hyperbolic distance even for a tiny angular gap, whereas points within the same subtree are closer.

**Rule of thumb (consistency with hierarchy).** If two leaves have lowest common ancestor depth $a$, then the dominant term of their distance is

$$d_{\mathbb{D}}(x_i, x_j) \approx 2\alpha (L - a) \ (+ \text{ lower-order terms}),$$

which increases strictly with tree distance and is monotone in the LCA depth ("closer relatives" are more similar). Hence hyperbolic space simultaneously preserves two signals—*depth* (radial) and *branching relation* (angular)—and avoids the hierarchical "flattening" of Euclidean embeddings.

## 3 METHODOLOGY

Our proposed Hyperbolic Associative Memory Networks (HAMNs) use hyperbolic geometry to store and retrieve patterns. This section introduces the core components of HAMNs.

### 3.1 MEMORY ENCODING IN HYPERBOLIC SPACE

We first map all memories and the query onto a common hyperbolic manifold. Let $x_i^R \in \mathbb{R}^d$ $(i = 1, \ldots, N)$ denote the $N$ stored patterns in Euclidean space (these can be regarded as the

keys in memory), and let $\xi^R \in \mathbb{R}^d$ be the query pattern (the cue or initial state). Let $(\mathcal{M}, g)$ be a complete, simply connected Riemannian manifold with constant negative curvature $-c < 0$. Choose a reference point $p \in \mathcal{M}$ and fix an orthonormal frame on its tangent space $T_p\mathcal{M}$, thereby identifying $T_p\mathcal{M} \cong \mathbb{R}^d$ via an isometric isomorphism $\iota_p : \mathbb{R}^d \to T_p\mathcal{M}$. We encode using the exponential map at $p$:

$$v_i = \iota_p(x_i^R), \qquad v_\xi = \iota_p(\xi^R), \qquad x_i = \exp_p^c\big(\Pi(v_i)\big), \quad \xi = \exp_p^c\big(\Pi(v_\xi)\big), \qquad (3)$$

To avoid, in some models, mapped points becoming too close to the boundary (which may lead to numerical instability and gradient explosion), we may perform norm clipping in the tangent space *before* the exponential map. Given a clipping threshold $\texttt{clip}_{\texttt{tan}} > 0$, $\Pi(\cdot)$ denotes tangent-space norm clipping:

$$\Pi(v) \;=\; v \cdot \min\left(1, \; \frac{\texttt{clip}_{\texttt{tan}}}{\|v\| + \varepsilon}\right), \qquad \varepsilon > 0. \qquad (4)$$

Here $\varepsilon$ is a small constant for numerical stability (e.g. $10^{-5}$). After this encoding step, all memory points $x_i$ and the query point $\xi$ lie on the manifold $\mathcal{M}$.

## 3.2 ENERGY FUNCTION DESIGN

On a hyperbolic manifold, we use an energy function $E(\xi)$ to measure how well the current retrieval state $\xi$ matches the stored patterns $\{x_i\}_{i=1}^N$: the energy should be low when $\xi$ is close to some memory $x_i$, and high otherwise. To this end, we replace the Euclidean inner product by a *hyperbolic similarity*:

$$\langle x, y \rangle_M := -\cosh\big(d_\mathcal{M}(x, y)\big),$$

where $d_\mathcal{M}$ is the geodesic distance induced by the metric $g$. This similarity is identical across hyperbolic models; in particular, in the Lorentz (hyperboloid) model $\langle x, y \rangle_M$ coincides with the classical Minkowski inner product, while in other models it can be computed directly from $d_\mathcal{M}$ without explicitly mapping between models. see Appendix A.1 for a short derivation and geometric intuition.

Accordingly, for any $\xi \in \mathcal{M}$ we define the energy as

$$E(\xi) \;=\; -\frac{1}{\theta} \log\left(\sum_{i=1}^N \exp\big(\theta \langle x_i, \xi \rangle_M\big)\right) \;+\; \frac{1}{2} d_\mathcal{M}(\xi, p)^2, \qquad (5)$$

where $\theta > 0$ is a temperature parameter and $p \in \mathcal{M}$ is a fixed reference point (e.g., the origin in Poincaré coordinates). We use the *intrinsic* squared geodesic regularizer $\frac{1}{2} d_\mathcal{M}(\xi, p)^2$, which is geodesically convex in hyperbolic space and penalizes deviations from $p$.

The first term in equation 5 is a smooth approximation to the "maximum similarity": when $\theta$ is large, $-\frac{1}{\theta} \log \sum_i \exp(\theta \langle x_i, \xi \rangle_M) \approx -\max_i \langle x_i, \xi \rangle_M$, so it is minimized when $\xi$ is close to one of the memories $x_i$. The second term penalizes large geodesic deviations from $p$, suppressing excursions toward the boundary and stabilizing the optimization trajectory.

Together, these two terms yield energy minima around stored memories. When $\xi = x_k$, we have $d_\mathcal{M}(x_k, \xi) = 0$ and $\langle x_k, \xi \rangle_M = -1$, leading to a low energy; conversely, when $\xi$ is far from all memories, the energy becomes large. Further discussion of energy bounds is provided in Appendix A.3. A simple inequality relating geodesic margin, softmax mass, and a sphere-packing style capacity bound is given in Appendix B (Propositions B.1, B.2, and B.3).An illustrative 2D Poincaré-disk energy landscape for a toy three-level hierarchy is shown in Appendix A.2 (Fig. 1), which provides a visual counterpart to these analytic properties.

## 3.3 MEMORY RETRIEVAL AND OPTIMIZATION

We optimize the retrieval energy using a *Riemannian* version of the Concave–Convex Procedure (CCCP) on a Hadamard manifold (Yuille & Rangarajan, 2001). A detailed derivation and convergence discussion for our setting are provided in Appendix A.4, and optional neural-computation analogies for the softmax weights, Riemannian gradient, and closed-form CCCP step are given in Appendix A.6. Here we summarize the resulting update rules.

**CCCP decomposition** Decompose $E(\xi)$ in equation 5 into a geodesically convex term and a concave term on a Hadamard manifold:

$$E(\xi) = E_{\text{cvx}}(\xi) + E_{\text{cave}}(\xi), \qquad E_{\text{cvx}}(\xi) = \tfrac{1}{2} d_{\mathcal{M}}(\xi, p)^2, \quad E_{\text{cave}}(\xi) = -\frac{1}{\theta} \log\Big( \sum_{i=1}^{N} e^{\theta \langle x_i, \xi \rangle_M} \Big),$$

(6)

where $p \in \mathcal{M}$ is a fixed reference point and $\langle x, \xi \rangle_M := -\cosh(d_{\mathcal{M}}(x, \xi))$ denotes the hyperbolic similarity. The squared distance is geodesically convex on Hadamard manifolds.

**Softmax weights** At iteration $t$, define

$$p_i^{(t)} = \frac{\exp\big(\theta \langle x_i, \xi^{(t)} \rangle_M\big)}{\sum_{j=1}^{N} \exp\big(\theta \langle x_j, \xi^{(t)} \rangle_M\big)}.$$

(7)

**Riemannian linearization and surrogate** Let $a^{(t)} := \operatorname{grad} E_{\text{cave}}(\xi^{(t)})$ be the *Riemannian* gradient at $\xi^{(t)}$. The concave part admits the first-order (Riemannian) upper bound

$$E_{\text{cave}}(\xi) \le E_{\text{cave}}(\xi^{(t)}) + \big\langle a^{(t)}, \log_{\xi^{(t)}}(\xi) \big\rangle_{\xi^{(t)}},$$

so the CCCP surrogate reads

$$Q\Big(\xi \mid \xi^{(t)}\Big) = \tfrac{1}{2} d_{\mathcal{M}}(\xi, p)^2 + \big\langle a^{(t)}, \log_{\xi^{(t)}}(\xi) \big\rangle_{\xi^{(t)}} \quad \text{(constants dropped).}$$

(8)

**Closed-form CCCP step** The minimizer of equation 8 on a Hadamard manifold is obtained in closed form:

$$\xi^{(t+1)} = \exp_p\Big( -\operatorname{PT}_{\xi^{(t)} \to p}\big(a^{(t)}\big) \Big),$$

(9)

where $\operatorname{PT}_{\xi^{(t)} \to p}$ denotes parallel transport along the unique geodesic from $\xi^{(t)}$ to $p$. Equivalently, introducing $v^{(t)} := -\operatorname{PT}_{\xi^{(t)} \to p}\big(a^{(t)}\big)$ and a damping step size $\eta \in (0, 1]$, we use the stable update

$$\xi^{(t+1)} = \exp_p\big(\eta \, v^{(t)}\big),$$

(10)

for which $\eta = 1$ recovers the exact minimizer in equation 9.

**Intrinsic gradient** Using equation 7, the Riemannian gradient of the concave term can be written as

$$a^{(t)} = \operatorname{grad} E_{\text{cave}}(\xi^{(t)}) = -\sum_{i=1}^{N} p_i^{(t)} \operatorname{grad}_\xi \langle x_i, \xi \rangle_M \Big|_{\xi = \xi^{(t)}}.$$

(11)

**Convergence note** Under the conditions spelled out in Appendix A.4, the Riemannian CCCP iterations monotonically decrease $E(\xi)$, and any accumulation point of the sequence $\{\xi^{(t)}\}$ is a (meta-)stable stationary point of $E$. In our experiments we empirically observe convergence to a single attractor corresponding to a stored memory.

### 3.4 HYPERBOLIC HOPFIELD MODULES FOR DEEP LEARNING

Inspired by the modular design of modern Hopfield networks Ramsauer et al. (2021), we adopt a similar architecture for modularization and replace its original Euclidean update mechanism with the hyperbolic retrieval strategy proposed in this paper. Based on this formulation, we construct three core modules—*Hyperbolic Hopfield* (**HypHopfield**), *Hyperbolic Hopfield Pooling* (**HypPooling**), and *Hyperbolic Hopfield Layer* (**HypLayer**)—targeting association, aggregation, and retrieval, respectively. All three are implemented on the Poincaré ball model and can be seamlessly integrated into deep neural networks, thereby enhancing hierarchical modeling and memory capabilities. Detailed structure and implementation are provided in Appendix D.

## 4 EXPERIMENTS

*Instantiation.* All experiments instantiate HAMNs on the Poincaré ball model (constant negative curvature $-c$); the model-agnostic derivations hold for any hyperbolic realization, and concrete formulas for instantiations on common hyperbolic models are provided in Appendix C.

**Overview** We evaluate HAMNs around the "hierarchy-sensitivity hypothesis" and their practical usefulness through five groups of experiments:

(i) **CIFAR-100 hierarchical classification**: On our 2/3/4-layer label trees, **HAMNs** deliver the strongest cross-level consistency and competitive accuracy across levels, clearly outperforming **Euclidean MHNs** and the kernelized **U-Hop**; the consistency gap widens as the hierarchy deepens. **HypAttn** is strongest for shallow/mid-level retrieval, while **HypNN** excels at fine-grained recognition.

(ii) **Weak/shallow hierarchical tasks**: On classical MIL multi-instance learning and MoleculeNet molecular property prediction (where hierarchy is weak or only shallow), HAMNs perform on par with Euclidean MHNs overall, with slight advantages on a few datasets.

(iii) **Real-world taxonomy and knowledge-graph tasks on WordNet**: We also evaluate HAMNs on two WordNet benchmarks—noun hypernym prediction and WN18RR link prediction—to assess the practical usefulness of our method on complex hierarchical structures. All decoders additionally use a shared ontology-embedding front-end, and we find ontology embeddings and hyperbolic retrieval to be *complementary*: the former provides a graph-aware label prior, while the latter controls how well the retrieval layer respects hierarchical geometry under that fixed prior.

(iv) **Computation/performance comparison**: Theoretically fewer FLOPs and parameters, but due to hyperbolic operations and memory-access overhead, the current GPU implementation exhibits longer runtime and higher peak memory.

(v) **Ablation studies**: Performance is best when the curvature $c$ lies in a moderate range (approximately 0.7–2.0); using too many stored patterns slightly degrades top-level accuracy, though the method is overall robust to this hyperparameter.

## 4.1 HIERARCHICAL CLASSIFICATION ON CIFAR-100

To demonstrate that HAMNs can understand and exploit multi-level structure, we conduct hierarchical classification experiments on CIFAR-100(Krizhevsky et al., 2009). CIFAR-100 groups 100 *fine* classes into 20 *coarse* classes, yielding a balanced two-level hierarchy. Without modifying the original samples, we further cluster the 20 coarse classes into 7 "super" classes (e.g., large terrestrial vertebrates, plants, vehicles), and then group these 7 super classes into 3 "top" classes (e.g., animals, plants & natural scenes), thereby forming three- and four-level hierarchies.

On the model side, we adopt a ResNet-18(He et al., 2016) backbone with the final fully connected layer removed, and insert one of five memory/retrieval modules: (i) **HAMNs** (Using our **HypLayer**; see Appendix D.), (ii) a hyperbolic attention (**HypAttn**; (Gülçehre et al., 2019)), (iii) a lightweight hyperbolic neural block (**HypNN**; (Ganea et al., 2018)), and (iv) Euclidean-space modern Hopfield networks (**MHNs**; (Ramsauer et al., 2021)). (v) a kernelized Euclidean Hopfield (**U-Hop**; (Wu et al., 2024)). The retrieved representations are then fed into level-specific classification heads.

*Coarse–Fine Coherence Correlation* (coph_corr) measures the consistency between the model's "coarse" predictions and the "coarse" predictions obtained by aggregating its "fine" outputs.

From Table 1 we observe a clear pattern across depths. Euclidean MHNs remain reasonably strong but are never dominant: they reach the best hierarchy-consistency score (coph_corr) on the 3-layer tree, yet on the deepest 4-layer setting they trail all hyperbolic variants in both high-level accuracy and cross-level coherence, and also underperform them at the fine-grained level. U-Hop exhibits a trade-off: it attains relatively high coph_corr but at a clear cost in flat accuracy; on deeper hierarchies it still lags markedly behind hyperbolic decoders. We conjecture this depth-insensitive trend is tied to its two-stage protocol: U-Hop first learns a dataset-level kernel (independent of hierarchy depth), then trains Hopfield retrieval dynamics on top of the fixed kernel. Among the two hyperbolic baselines, HypAttn is particularly well suited to shallow hierarchies, giving the strongest performance when only a coarse–fine split is present, whereas HypNN gradually becomes the best fine-grained recognizer as the tree deepens, at the cost of weaker alignment between its predictions across levels. HAMNs (ours) offer the most balanced behaviour: already on the 3-layer hierarchy they match or exceed the baselines at all levels, and on the 4-layer hierarchy they simultaneously achieve the strongest high-level performance and the highest hierarchy-consistency, while keeping fine-grained accuracy competitive and only slightly below HypNN.

Table 1: Hierarchical classification on CIFAR-100 results.

| Model | top_acc | super_acc | coarse_acc | fine_acc | coph_corr |
|---|---|---|---|---|---|
| **CIFAR-100-2-layer** | | | | | |
| Backbone only | — | — | $64.20 \pm 0.91$ | $51.00 \pm 1.28$ | $0.6652 \pm 0.0163$ |
| HypAttn | — | — | $\mathbf{70.67 \pm 0.56}$ | $\mathbf{58.19 \pm 0.38}$ | $0.6740 \pm 0.0142$ |
| HypNN | — | — | $69.02 \pm 0.47$ | $56.82 \pm 0.41$ | $0.5938 \pm 0.0202$ |
| MHNs | — | — | $65.34 \pm 0.86$ | $49.86 \pm 0.63$ | $0.6295 \pm 0.0143$ |
| U-Hop | — | — | $62.14 \pm 0.46$ | $45.74 \pm 0.44$ | $\mathbf{0.7195 \pm 0.0993}$ |
| **HAMNs (ours)** | — | — | $70.12 \pm 0.57$ | $56.00 \pm 0.64$ | $0.6778 \pm 0.0193$ |
| **CIFAR-100-3-layer** | | | | | |
| Backbone only | — | $72.75 \pm 1.89$ | $62.58 \pm 1.35$ | $50.79 \pm 0.82$ | $0.7023 \pm 0.0164$ |
| HypAttn | — | $79.33 \pm 0.66$ | $68.68 \pm 0.78$ | $54.01 \pm 0.97$ | $0.6902 \pm 0.0240$ |
| HypNN | — | $79.40 \pm 0.66$ | $\mathbf{68.84 \pm 0.95}$ | $54.09 \pm 1.05$ | $0.7123 \pm 0.0211$ |
| MHNs | — | $79.17 \pm 0.59$ | $68.08 \pm 0.84$ | $52.89 \pm 0.97$ | $\mathbf{0.7152 \pm 0.0256}$ |
| U-Hop | — | $75.11 \pm 0.51$ | $62.85 \pm 0.55$ | $45.93 \pm 0.83$ | $0.6996 \pm 0.0717$ |
| **HAMNs (ours)** | — | $\mathbf{79.70 \pm 0.29}$ | $68.81 \pm 0.59$ | $\mathbf{54.27 \pm 0.47}$ | $0.7017 \pm 0.0658$ |
| **CIFAR-100-4-layer** | | | | | |
| Backbone only | $87.51 \pm 0.73$ | $72.68 \pm 1.85$ | $60.02 \pm 1.02$ | $47.23 \pm 0.77$ | $0.7180 \pm 0.0230$ |
| HypAttn | $90.13 \pm 0.48$ | $78.23 \pm 0.48$ | $67.74 \pm 0.93$ | $54.50 \pm 0.78$ | $0.6795 \pm 0.0143$ |
| HypNN | $90.30 \pm 0.35$ | $78.72 \pm 0.59$ | $68.29 \pm 0.80$ | $\mathbf{55.93 \pm 0.88}$ | $0.6046 \pm 0.0149$ |
| MHNs | $89.39 \pm 0.29$ | $76.97 \pm 0.44$ | $65.56 \pm 0.42$ | $49.37 \pm 0.57$ | $0.5902 \pm 0.0218$ |
| U-Hop | $88.62 \pm 0.57$ | $75.46 \pm 0.58$ | $62.46 \pm 0.97$ | $45.44 \pm 0.94$ | $0.7154 \pm 0.0680$ |
| **HAMNs (ours)** | $\mathbf{90.98 \pm 0.39}$ | $\mathbf{79.48 \pm 0.57}$ | $\mathbf{68.51 \pm 0.84}$ | $53.49 \pm 1.05$ | $\mathbf{0.7184 \pm 0.0254}$ |

**Takeaway.** On deeper label trees, negatively curved retrieval provides a better accuracy–coherence trade-off than Euclidean Hopfield baselines (including kernelized U-Hop): Euclidean methods can improve cross-level coherence but do not match hyperbolic decoders in deep-hierarchy performance, while **HAMNs** achieve the best overall balance on the deepest hierarchy.

Beyond these main results, we provide two additional analyses in Appendix E.1. First, we show that our conclusions are robust under an alternative, feature-driven hierarchy on CIFAR-100 (Appendix E.1.1). Second, we inspect the learned hyperbolic memories and find that radial coordinates and geodesic distances correlate well with the ground-truth tree structure (Appendix E.1.2).

### 4.2 REAL-WORLD HIERARCHICAL TASKS: WORDNET TAXONOMY AND WN18RR

While CIFAR-100 provides a benchmark for synthetic label trees, many applications come with a fixed ontology.

On **WordNet hypernym prediction** (App. E.4), inputs are SBERT-encoded synsets and the goal is to predict their immediate hypernyms. Under plain SBERT features, the Euclidean Hopfield baseline (MHN_Euc, U-Hop) performs very poorly: their accuracy is less than half of the hyperbolic decoders and its average graph distance is much larger, indicating that a flat Euclidean energy struggles to exploit the multi-level taxonomy. Both hyperbolic attention (HypAttn) and our HAMNs achieve substantially higher accuracy and much smaller hierarchical error, with HypAttn slightly ahead and HAMNs remaining competitive. When we equip *all* decoders with the same ontology encoder (**OntEuc**), trained as a lightweight MLP on the hypernym graph, every model improves dramatically; nevertheless, HypAttn and HAMNs still clearly outperform MHN_Euc, U-Hop, showing that ontology embeddings and hyperbolic retrieval are complementary rather than interchangeable.

On **WN18RR link prediction** (App. E.5), a knowledge-graph completion benchmark built from the same WordNet ontology, we reuse a single encoder for entities and relations and only swap the decoder. All three hyperbolic decoders (HypAttn, HypNN, HAMNs) substantially outperform Euclidean MHN_Euc on MRR and Hits@1/3/10, confirming that hyperbolic geometry is advantageous on strongly hierarchical graphs. Among them, HAMNs achieves the best overall performance with

slightly higher MRR and Hits@K while using the same encoder and memory size, indicating that the proposed CCCP-based hyperbolic energy minimization is practically effective as a drop-in decoder.

We refer the reader to App. E.8 for a focused comparison between HAMNs and hyperbolic attention.

**Takeaway.** Across these two WordNet-based tasks—one taxonomy classification and one knowledge-graph completion problem—HAMNs act as a robust hyperbolic retrieval module that can be plugged into standard pipelines, clearly outperform Euclidean MHNs on deep hierarchical label spaces, and remain competitive with other hyperbolic decoders.

### 4.3 WEAK/SHALLOW HIERARCHY TASKS: MIL AND MOLECULAR PROPERTY PREDICTION

**Multi–Instance Learning (MIL)** We evaluate on three classical MIL datasets—**Tiger**, **Elephant**, and **Fox**—to probe the bag–instance regime without instance-level labels (Dietterich et al., 1997), using the standard splits introduced by (Ilse et al., 2018; Küçükaşcı & Baydoğan, 2018; Carbonneau et al., 2018). We plug our **HypPooling** into the MIL pipeline: embedded instances serve as stored memories ($Y$), while a fixed set of learnable query vectors acts as state (query) patterns ($R$) on the same Poincaré ball; retrieval is performed via hyperbolic attention and on-manifold updates. See Appendix D for the layer design and Appendix E.2 for training protocol and hyperparameters. We compare against representative MIL baselines (e.g., attention-MIL (Ilse et al., 2018), mi-Net variants (Carbonneau et al., 2018), and Euclidean MHNs). Results show *competitive* overall performance and new SOTA on **Fox**; elsewhere the margins over Euclidean MHNs are modest (Table 5).

**Molecular property prediction** Experiments on four MoleculeNet datasets—**HIV**, **BACE** (Subramanian et al., 2016), **BBBP** (Martins et al., 2012), and **SIDER** (Kuhn et al., 2016)—probe the weak/shallow–hierarchy regime. The proposed **HypLayer** is inserted into standard pipelines: training samples serve as stored memories ($Y$), inputs as queries ($R$), followed by hyperbolic embedding and retrieval (exact layer design, training protocol, and hyperparameters are detailed in Appendix D). Comparisons cover representative baselines (classical ML, GNNs, and Euclidean MHNs). This approach yields *competitive* performance and sets SOTA on **BBBP** and **SIDER** (Appendix E.3); margins over Euclidean MHNs are small, consistent with the weak-hierarchy hypothesis.

### 4.4 COMPUTATIONAL COST AND PERFORMANCE

Using PyTorch's profiler on a single NVIDIA RTX 4090 GPU, we compare MHNs with the three hyperbolic decoders (HypAttn, HypNN, HAMNs) under the CIFAR-100 hierarchical classification and WN18RR link-prediction setups. For a fixed backbone and hidden size, HAMNs typically use fewer parameters and FLOPs than Euclidean MHNs, but incur higher runtime and peak memory because of hyperbolic operations (Möbius addition, exponential/logarithmic maps, Riemannian gradient transforms) and the associated memory-access overhead. Among the hyperbolic variants, HypAttn is the most expensive due to repeated geodesic distance and Fréchet-mean computations, HypNN is the most lightweight, and HAMNs lies in between. See Appendix E.6 for detailed.

### 4.5 ABLATIONS: CURVATURE AND NUMBER OF STORED PATTERNS

**Summary.** A *moderate curvature* provides the best trade-off; extremes are harmful (too small under-expresses hierarchy, too large degrades accuracy). Varying the *number of stored patterns* causes only small overall fluctuations: oversized memories slightly reduce top-level accuracy, moderate increases help mid-level, and fine-level performance peaks at higher counts. In practice, we recommend *moderate curvature* and a *modest memory size*. See Appendix E.7 for detailed.

## 5 RELATED WORK

**Associative memories and modern Hopfield networks** Classical Hopfield networks were first introduced by Hopfield (1982), who viewed the network as a recurrent dynamical system whose energy function stores binary patterns as attractors and retrieves them via energy minimization; continuous variants (Tank & Hopfield, 1986) extend the state space to real-valued vectors. In contemporary machine learning, *modern Hopfield networks* (MHNs) are used as differentiable associative-memory

layers inside deep architectures: for example, Ramsauer et al. (2021) derive one-step-convergent energies, analyze memory capacity, and clarify the correspondence between MHNs and attention. Subsequent work extends this framework along several axes, including kernelized or learnable query–memory similarities that improve capacity and robustness (Wu et al., 2024), sparse and structured variants based on sparse entropic regularizers or Fenchel–Young losses and SparseMAP (Hu et al., 2023; Santos et al., 2024), and latent-structured Hopfield networks that embed continuous attractor dynamics into autoencoder latent spaces for semantic association and episodic-style memory (Li et al., 2025). In this paper we adopt this *machine-learning perspective*: we treat MHNs as pluggable energy-based retrieval layers and focus on moving the Hopfield energy and dynamics to negatively curved manifolds, in order to study how geometry and optimization jointly affect hierarchical retrieval, rather than proposing new similarity kernels or a biologically realistic circuit.

**Hyperbolic Geometry** Nickel & Kiela (2017) first proposed using hyperbolic space to learn hierarchical representations of symbolic data, such as text and graphs, by embedding them into the Poincaré ball model. Since then, the application of hyperbolic geometry has been explored in various domains. Ganea et al. (2018) introduced hyperbolic neural network layers, which have enabled the development of hybrid architectures such as hyperbolic convolutional neural networks (Shimizu et al., 2021), hyperbolic graph convolutional networks (Chami et al., 2019), hyperbolic variational autoencoders (Ovinnikov et al., 2021), and hyperbolic attention networks (Gülçehre et al., 2019). These architectures have been successfully applied to tasks such as deep metric learning, object detection , and natural language processing. Beyond practical applications, theoretical investigations into hyperbolic spaces and their models have also deepened, demonstrating properties such as lower representation distortion (De Sa et al., 2018), better generalization ability (Bachmann et al., 2021), and stronger representation power in low-dimensional spaces (Sala et al., 2018). Unlike prior implicit uses of hyperbolic geometry, energy-based Hopfield retrieval is carried out directly in hyperbolic space, broadening applicability to hierarchical representation learning.

## 6 LIMITATIONS

The scope of this work is explicitly limited to *differentiable associative memory modules for machine learning*, rather than neurobiological models of memory. On the methodological side, we currently derive and implement our framework only for the case of *constant negative curvature*, and we instantiate it using the Poincaré ball model in our implementation. Other non-Euclidean geometric models are not yet explored experimentally in this paper. From a systems perspective, our results highlight a trade-off between theory and practice. Although under the same configuration HAMNs require fewer theoretical FLOPs and parameters than an Euclidean MHN layer, the current GPU kernels for hyperbolic operations incur substantial overhead, leading to longer wall-clock time and higher peak memory usage in practice. Finally, our empirical study mainly examines whether the model can exploit hierarchical geometric information and how reliable it is on deeply hierarchical tasks in practical applications. Extending HAMNs to larger models and modalities is future work.

## 7 CONCLUSION

We propose a plug-and-play, *model-agnostic* memory framework that systematically extends modern Hopfield networks from Euclidean to hyperbolic geometry, and formulates retrieval as energy minimization based on geodesic distance and its induced "Minkowski-like" similarity. Overall, our experimental results support the *hierarchy-sensitivity hypothesis*: as hierarchical depth increases, HAMNs achieve statistically significant performance gains over Euclidean MHNs, whereas in flat or shallow settings their performance is largely on par with Euclidean MHNs. On deep label spaces, HAMNs remain competitive with other hyperbolic decoders. The results further indicate that ontology embeddings primarily provide a *label prior*, while the hyperbolic memory layer determines how strongly the retrieval process exploits hierarchical geometry under this prior; the two components are therefore functionally complementary. Taken together, these findings suggest that HAMNs can serve both as a generic memory module—deployable in any downstream task that requires storing and retrieving hierarchical patterns—and as a practically useful way to obtain consistent gains whenever an explicit multi-level structure is present, offering a concrete starting point for designing geometry-aware associative memory modules in more complex hierarchical settings.

## REPRODUCIBILITY STATEMENT

We provide a runnable implementation of HAMNs instantiated on the Poincaré ball and the CIFAR-100 hierarchical experiments, submitted as supplementary materials. Code-level implementation details for **HypHopfield**, **HypPooling**, and **HypLayer** are given in Appx. D. For other common hyperbolic models (Lorentz, Klein, upper half-plane, hemisphere), model-agnostic replacement formulas are provided in Appx. C. The supplementary code package includes the training scripts and module-instantiation code required to reproduce the experiments, including the CIFAR-100 clustering-sensitivity study (Appx. E.1.1), the WordNet hypernym prediction task (Appx. E.4), and the WN18RR link-prediction experiment (Appx. E.5).

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

## A  HYPERBOLIC ENERGY-BASED OPTIMIZATION FRAMEWORK

### A.1  MINKOWSKI INNER PRODUCT AND HYPERBOLIC DISTANCE

We briefly recall how the Minkowski inner product controls hyperbolic distance in the Lorentz model, and how this justifies our design choice of using $-\langle \cdot, \cdot \rangle_L$ as the association term in the HAMNs energy. Formally, the $d$-dimensional hyperbolic space in Lorentz form is

$$\mathbb{H}^d = \left\{ u \in \mathbb{R}^{d+1} : \langle u, u \rangle_L = -1, \ u_0 > 0 \right\}, \tag{12}$$

where the Minkowski inner product is defined as

$$\langle u, v \rangle_L = -u_0 v_0 + \sum_{i=1}^{d} u_i v_i, \qquad u, v \in \mathbb{R}^{d+1}. \tag{13}$$

It is well known that the geodesic distance on $\mathbb{H}^d$ can be expressed as

$$d_{\mathbb{H}}(u, v) = \operatorname{arcosh}\left(-\langle u, v \rangle_L\right), \qquad u, v \in \mathbb{H}^d. \tag{14}$$

(Nickel & Kiela, 2018; Ratcliffe, 2006)

Since $\operatorname{arcosh}(\cdot)$ is strictly increasing on $[1, +\infty)$, Eq. equation 14 implies a simple monotonic relationship: reducing the hyperbolic distance $d_{\mathbb{H}}(u, v)$ is equivalent to reducing its argument $-\langle u, v \rangle_L$, or, equivalently, *increasing* the Minkowski inner product $\langle u, v \rangle_L$. In other words, the negative Minkowski inner product $-\langle u, v \rangle_L$ behaves as a curvature-aware dissimilarity measure that is strictly order-equivalent to the hyperbolic distance.

**Implication for the HAMNs energy**  In our associative memory module we use an energy of the form

$$E(z, \Xi) = - \sum_{m \in \Xi} \alpha_m \langle z, m \rangle_L + \lambda \, \Phi(z), \tag{15}$$

where $\Xi$ denotes the set of memory vectors, $\alpha_m \geq 0$ are association weights, and $\Phi(z)$ is an intrinsic regularizer. Because of equation 14, the association term $-\langle z, m \rangle_L$ can be interpreted as a smooth surrogate for the hyperbolic distance between the current state $z$ and memory $m$: decreasing this term aligns with moving $z$ closer to its relevant memories in the hyperbolic metric. The regularizer $\Phi(z)$ penalizes travelling too far along geodesics (e.g., towards the boundary), thereby balancing attraction towards memories with geometric stability.

**Time-like vs space-like components**  Any point $u \in \mathbb{H}^d$ in the Lorentz model can be written as $u = (u_0, \bar{u}) \in \mathbb{R}^{1+d}$ with $\langle u, u \rangle_L = -1$, which implies $u_0 = \cosh r_u$ and $\|\bar{u}\|_2 = \sinh r_u$ for some radial coordinate $r_u = d_{\mathbb{H}}(u, o)$. For two points $u, v \in \mathbb{H}^d$ with radii $r_u, r_v$ and angular separation $\theta$ between their spatial parts $\bar{u}, \bar{v}$, the Minkowski inner product decomposes as

$$\langle u, v \rangle_L = - \cosh r_u \cosh r_v + \sinh r_u \sinh r_v \cos \theta.$$

Thus the "time-like" component $-\cosh r_u \cosh r_v$ and the "space-like" component $\sinh r_u \sinh r_v \cos \theta$ contribute with opposite signs. For fixed angular term $\cos \theta$, increasing the radii $r_u, r_v$ makes $\langle u, v \rangle_L$ more negative, so that the surrogate dissimilarity $-\langle u, v \rangle_L$ grows roughly with the sum $r_u + r_v$. Consequently, the Minkowski-based energy is more sensitive to radial separation than to purely angular differences, which naturally encourages the model to encode hierarchical depth along the radial direction.

**Depth–radius correlation**  A further geometric intuition comes from the radial coordinate $\|z\|_{\mathbb{H}}$, defined as the hyperbolic distance from a reference origin $o \in \mathbb{H}^d$, i.e., $\|z\|_{\mathbb{H}} = d_{\mathbb{H}}(z, o)$. On the hierarchical CIFAR-100 experiment, we can directly probe this geometry: for each stored class memory we record its tree depth (top / super / coarse / fine) and its hyperbolic radius. As detailed in Appendix E.1.2, the average radius increases monotonically from top to fine classes, and the Spearman correlation between discrete depth and $\|z\|_{\mathbb{H}}$ is $\rho = 0.57$ (with $p \approx 1.6 \times 10^{-12}$). This provides empirical evidence that optimizing the Minkowski-based hyperbolic energy naturally organizes concepts according to hierarchical depth in the learned representation.

## A.2 ILLUSTRATIVE HYPERBOLIC ENERGY LANDSCAPE

To complement the analytic discussion above, Fig. 1 visualizes the retrieval energy $E(\xi)$ of a toy HAMNs on a 2D Poincaré disk. We embed a three-level hierarchy with *top*, *coarse*, and *fine* attractors placed at increasing radii. Red, orange, and cyan dots indicate the centers of the top-, coarse-, and fine-level patterns, respectively. The color map shows the value of $E(\xi)$ on a dense grid inside the disk.

We observe that (i) top-level patterns near the center give rise to broad and smooth basins, corresponding to coarse semantic groups; (ii) coarse-level patterns at intermediate radii carve out more localized wells while remaining separated; and (iii) fine-level patterns near the boundary induce sharp basins that densely tile the outer region without collapsing into each other. Visually, the energy field therefore changes systematically across hierarchical levels: as we move radially outward, basins become progressively sharper and more numerous. This illustrates how negative curvature allows us to allocate increasingly fine-grained attractors along the radial direction while preserving a smooth global energy landscape, in line with our capacity and margin discussion in App. B.

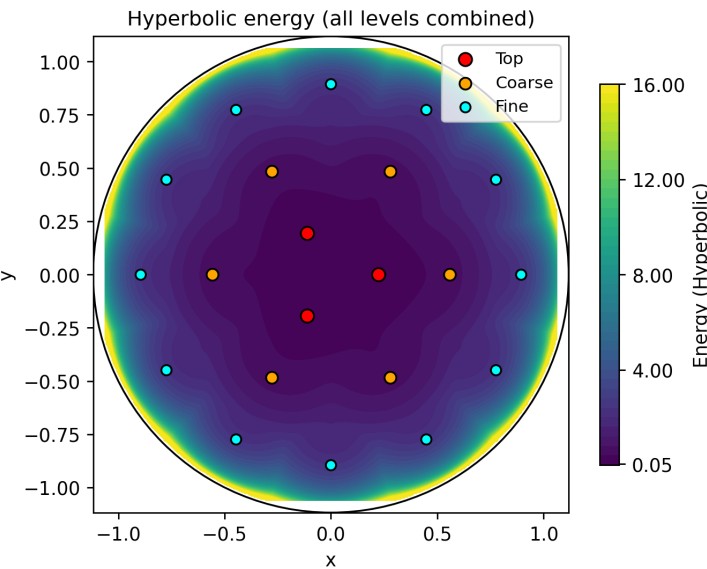

Figure 1: Hyperbolic retrieval energy on a 2D Poincaré disk for a toy three-level hierarchy. Red, orange, and cyan dots denote top-, coarse-, and fine-level attractors, respectively. Darker regions correspond to lower energy.

## A.3 BOUNDING THE ENERGY FUNCTION

We consider the energy

$$E(\xi) = -\frac{1}{\theta} \log\left( \sum_{i=1}^{S} \exp\big(\theta \langle x_i, \xi \rangle_M \big) \right) + \frac{1}{2} d_{\mathcal{M}}(\xi, p)^2, \tag{16}$$

where the (hyperbolic) similarity is $\langle x, y \rangle_M := -\cosh\big(d_{\mathcal{M}}(x,y)\big)$ on a complete, simply connected Riemannian manifold $(\mathcal{M}, g)$ of constant negative curvature $-c$.

**Setup and notation** Fix a base point $p \in \mathcal{M}$ and define

$$r_i := d_{\mathcal{M}}(x_i, p), \qquad r := d_{\mathcal{M}}(\xi, p).$$

Let $M_r := \max_i r_i$. We assume optimization is restricted (by standard clipping/projection) to a geodesic ball around $p$, i.e., $r \leq R_r$. Here $S$ is the number of stored patterns and $\theta > 0$ the inverse temperature.

### A.3.1 BOUNDING THE SIMILARITY

By the triangle inequality,

$$|r_i - r| \leq d_{\mathcal{M}}(x_i, \xi) \leq r_i + r.$$

Since $\cosh(\cdot)$ is strictly increasing on $[0, \infty)$ and $\langle x_i, \xi \rangle_M = -\cosh d_{\mathcal{M}}(x_i, \xi)$, we obtain for each $i$

$$-\cosh(r_i + r) \leq \langle x_i, \xi \rangle_M \leq -\cosh(|r_i - r|). \tag{17}$$

Consequently, using $r_i \leq M_r$ and $r \leq R_r$,

$$\boxed{-\cosh(M_r + R_r) \leq \langle x_i, \xi \rangle_M \leq -1} \qquad (\forall i), \tag{18}$$

because $\cosh(0) = 1$ and $|r_i - r|$ can be as small as 0.

### A.3.2 BOUNDING THE ENERGY

Write $E(\xi) = E_{\text{cave}}(\xi) + E_{\text{cvx}}(\xi)$ with

$$E_{\text{cave}}(\xi) = -\frac{1}{\theta} \log \sum_{i=1}^{S} e^{\theta z_i}, \quad z_i := \langle x_i, \xi \rangle_M, \qquad E_{\text{cvx}}(\xi) = \tfrac{1}{2} d_{\mathcal{M}}(\xi, p)^2.$$

For any $\theta > 0$, the log-sum-exp bounds yield

$$\max_i z_i \leq \frac{1}{\theta} \log \sum_i e^{\theta z_i} \leq \max_i z_i + \frac{\log S}{\theta} \implies -\max_i z_i - \frac{\log S}{\theta} \leq E_{\text{cave}}(\xi) \leq -\max_i z_i.$$

From equation 18 we have $\max_i z_i \in \left[ -\cosh(M_r + R_r), -1 \right]$. Hence

$$\boxed{1 - \frac{\log S}{\theta} \leq E_{\text{cave}}(\xi) \leq \cosh(M_r + R_r)}. \tag{19}$$

For the convex part (squared distance to $p$),

$$0 \leq E_{\text{cvx}}(\xi) = \tfrac{1}{2} d_{\mathcal{M}}(\xi, p)^2 \leq \tfrac{1}{2} R_r^2. \tag{20}$$

### A.3.3 FINAL BOUNDS

Combining equation 19 and equation 20 yields

$$\boxed{1 - \frac{\log S}{\theta} \leq E(\xi) \leq \cosh(M_r + R_r) + \tfrac{1}{2} R_r^2}. \tag{21}$$

The constants depend only on the maximal radial extents of memories and states $(M_r, R_r)$ and the inverse temperature $\theta$, but *not* on the specific hyperbolic model. Thus the boundedness of $E$—and hence the numerical stability of CCCP or Riemannian-gradient iterations—holds uniformly across all constant-curvature hyperbolic realizations.

## A.4 OPTIMIZATION OF THE ENERGY FUNCTION VIA CCCP

### A.4.1 CONCAVITY/CONVEXITY ON HADAMARD MANIFOLDS

We work on a Hadamard manifold $(\mathcal{M}, g)$ with constant negative curvature. Write

$$E(\xi) = E_{\text{cvx}}(\xi) + E_{\text{cave}}(\xi), \qquad E_{\text{cvx}}(\xi) = \tfrac{1}{2} d_{\mathcal{M}}(\xi, p)^2, \quad E_{\text{cave}}(\xi) = -\tfrac{1}{\theta} \log \sum_{i=1}^{N} e^{\theta s_i(\xi)},$$

where $s_i(\xi) := \langle x_i, \xi \rangle_M = -\cosh(d_{\mathcal{M}}(x_i, \xi))$. It is known that $d_{\mathcal{M}}(\cdot, \cdot)$ is geodesically convex on Hadamard manifolds; since $\cosh$ is convex and strictly increasing on $[0, \infty)$, the composition $\cosh \circ d_{\mathcal{M}}$ is geodesically convex, hence $s_i(\xi) = -\cosh(d_{\mathcal{M}}(x_i, \xi))$ is *geodesically concave*. For the concavity of $E_{\text{cave}}(\xi) = -\text{lse}_\theta(\{s_i(\xi)\}_i)$, let $F(\xi) = \text{lse}_\theta(\{s_i(\xi)\}_i)$. For any unit tangent

vector $u \in T_\xi \mathcal{M}$, the (Riemannian) Hessian admits the standard decomposition (see the derivation in §A.5):

$$\text{Hess}_\xi F[u, u] = \sum_{i=1}^{N} p_i(\xi) \, \text{Hess}_\xi \, s_i[u, u] + \theta \, \text{Var}_{p(\xi)}\big(\langle \text{grad} \, s_i(\xi), u \rangle_g\big),$$

where $p_i(\xi) = \frac{e^{\theta s_i(\xi)}}{\sum_j e^{\theta s_j(\xi)}}$. Each $s_i$ is geodesically concave, so $\text{Hess}_\xi \, s_i[\cdot, \cdot] \preceq 0$; the second term is a nonnegative variance term. Therefore $\text{Hess}_\xi(-F)[u, u] = -\sum_i p_i \, \text{Hess}_\xi \, s_i[u, u] - \theta \, \text{Var}_p(\cdot)$ is "a difference of a negative semidefinite and a positive semidefinite term." On a bounded geodesic ball, if there exists $\kappa > 0$ such that $-\text{Hess}_\xi \, s_i \succeq \kappa I$ and $L := \max_{i, \xi} \|\text{grad} \, s_i(\xi)\|_g < \infty$, then whenever $0 < \theta \leq \kappa/L^2$ we have $\text{Hess}_\xi(-F) \preceq 0$, hence $E_{\text{cave}}$ is geodesically concave. Under this temperature range, $E = E_{\text{cvx}} + E_{\text{cave}}$ satisfies the "convex + concave" requirement for CCCP. In practice, we also observe monotone decrease under typical training temperatures.

**Riemannian DC / CCCP viewpoint** Classical DC programming and CCCP are formulated in Euclidean spaces for functions $f = g - h$ with $g$ convex and $h$ convex; in our setting the ambient space is a Hadamard manifold and convexity is understood in the *geodesic* sense. The role of the Euclidean linearization $h(x) \approx h(x^{(t)}) + \langle \nabla h(x^{(t)}), x - x^{(t)} \rangle$ is played by the Riemannian first–order expansion

$$h(\xi) \approx h(\xi^{(t)}) + \big\langle \text{grad} \, h(\xi^{(t)}), \log_{\xi^{(t)}}(\xi) \big\rangle_{\xi^{(t)}},$$

where $\log_{\xi^{(t)}}(\cdot)$ is the Riemannian logarithm map. Replacing Euclidean convexity and linearization by geodesic convexity and the logarithm map is the only change; the surrogate $Q(\cdot \mid \xi^{(t)})$ in equation 22 is therefore the natural Riemannian analogue of the CCCP surrogate on Hadamard manifolds (see also standard references on Riemannian optimization, e.g. (Bonnabel, 2013)).

### A.4.2 CCCP Linearization and Surrogate

Let $\xi^{(t)}$ be the current iterate. A first-order (Riemannian) upper bound for the concave part yields

$$E_{\text{cave}}(\xi) \leq E_{\text{cave}}(\xi^{(t)}) + \big\langle a^{(t)}, \log_{\xi^{(t)}}(\xi) \big\rangle_{\xi^{(t)}}, \qquad a^{(t)} := \text{grad} \, E_{\text{cave}}(\xi^{(t)}).$$

Thus the "bound-minimization" surrogate for CCCP is

$$Q\big(\xi \mid \xi^{(t)}\big) = \tfrac{1}{2} d_{\mathcal{M}}(\xi, p)^2 + \big\langle a^{(t)}, \log_{\xi^{(t)}}(\xi) \big\rangle_{\xi^{(t)}} + \text{const.} \tag{22}$$

### A.4.3 Closed-Form Update (with Parallel Transport)

On a Hadamard manifold, the minimizer of equation 22 admits the closed form

$$\xi^{(t+1)} = \exp_p\Big( - \text{PT}_{\xi^{(t)} \to p}\big(a^{(t)}\big)\Big), \tag{23}$$

where $\text{PT}_{\xi^{(t)} \to p}$ denotes parallel transport along the unique geodesic from $\xi^{(t)}$ to $p$. For numerical stability, we employ a damped step with $\eta \in (0, 1]$:

$$\xi^{(t+1)} = \exp_p\Big( \eta \, v^{(t)} \Big), \qquad v^{(t)} := - \text{PT}_{\xi^{(t)} \to p}\big(a^{(t)}\big), \tag{24}$$

which reduces to equation 23 when $\eta = 1$.

### A.4.4 Softmax Weights and Riemannian Gradient of the Concave Term

Let $p_i^{(t)} = \frac{\exp(\theta \, s_i(\xi^{(t)}))}{\sum_j \exp(\theta \, s_j(\xi^{(t)}))}$. By the chain rule together with §A.5, equation 27, we obtain

$$a^{(t)} = \text{grad} \, E_{\text{cave}}(\xi^{(t)}) = -\sum_{i=1}^{N} p_i^{(t)} \, \text{grad}_\xi \, s_i(\xi)\Big|_{\xi = \xi^{(t)}}. \tag{25}$$

**Convergence note**    When $\theta$ satisfies the above sufficient condition, $E_{\text{cvx}}$ is geodesically convex and $E_{\text{cave}}$ is geodesically concave, so $E = E_{\text{cvx}} + E_{\text{cave}}$ fits the standard "convex + concave" setting of DC / CCCP on a Hadamard manifold. If $E$ is bounded from below and its sublevel set $\{\xi \in \mathcal{M} : E(\xi) \le E(\xi^{(0)})\}$ is compact, then exact minimization of the surrogate $Q(\cdot \mid \xi^{(t)})$ at each step yields a sequence $\{\xi^{(t)}\}$ such that the energies $E(\xi^{(t)})$ are monotonically nonincreasing and converge to a finite limit, and every accumulation point $\xi^{\star}$ of $\{\xi^{(t)}\}$ is a Riemannian stationary point, $\operatorname{grad} E(\xi^{\star}) = 0$. We do *not* claim that the whole sequence $\{\xi^{(t)}\}$ converges to a stationary point; our guarantee is restricted to energy monotonicity and stationary accumulation points, in direct analogy with classical CCCP in Euclidean spaces (Yuille & Rangarajan, 2001).

## A.5  RIEMANNIAN GRADIENT OF THE CONCAVE TERM

Consider the hyperbolic similarity

$$s_M(x, y) := \langle x, y \rangle_M = -\cosh\big(d_{\mathcal{M}}(x, y)\big).$$

Let $d_{\mathcal{M}}$ denote the geodesic distance and $\log_x : \mathcal{M} \to T_x\mathcal{M}$ the Riemannian logarithm at $x$. On a Hadamard manifold, for any $x \ne y$,

$$\operatorname{grad}_x d_{\mathcal{M}}(x, y) = -\frac{\log_x(y)}{\|\log_x(y)\|_g}, \tag{26}$$

where $\|\cdot\|_g$ is the norm induced by $g$ on $T_x\mathcal{M}$.

**Chain rule for the similarity gradient**    By the chain rule,

$$\operatorname{grad}_x s_M(x, y) = -\sinh\big(d_{\mathcal{M}}(x, y)\big) \operatorname{grad}_x d_{\mathcal{M}}(x, y),$$

and substituting equation 26 yields

$$\boxed{\operatorname{grad}_x s_M(x, y) = \sinh\big(d_{\mathcal{M}}(x, y)\big) \frac{\log_x(y)}{\big\|\log_x(y)\big\|_g}} \,. \tag{27}$$

The gradient points along the unit tangent from $x$ to $y$ with magnitude $\sinh(d_{\mathcal{M}}(x, y))$.

**Riemannian gradient of** $E_{\text{cave}}$    Let $s_i(\xi) = \langle x_i, \xi \rangle_M$ and $p_i(\xi) = \frac{e^{\theta s_i(\xi)}}{\sum_j e^{\theta s_j(\xi)}}$. Then

$$\boxed{\operatorname{grad} E_{\text{cave}}(\xi) = -\sum_{i=1}^{N} p_i(\xi) \operatorname{grad}_\xi s_i(\xi) = -\sum_{i=1}^{N} p_i(\xi) \sinh\big(d_{\mathcal{M}}(x_i, \xi)\big) \frac{\log_\xi(x_i)}{\big\|\log_\xi(x_i)\big\|_g}} \,. \tag{28}$$

**Coordinate gradient (Poincaré ball example)**    If the chosen coordinates are conformal (e.g., the Poincaré ball), then $g(\xi) = \lambda(\xi)^2 I$ with $\lambda(\xi) = \frac{2}{1-c\|\xi\|^2}$. The Euclidean (coordinate) gradient $\nabla_\xi$ and the Riemannian gradient $\operatorname{grad}_\xi$ satisfy

$$\nabla_\xi f = G(\xi)^{-1} \operatorname{grad}_\xi f = \lambda(\xi)^{-2} \operatorname{grad}_\xi f. \tag{29}$$

Plugging equation 28 into equation 29 yields an implementation-ready Euclidean gradient expression.

## A.6  OPTIONAL NEURAL COMPUTATION ANALOGIES

**(i) About equation 7 (softmax weights).**    Equation equation 7 represents the Gibbs/Luce choice rule, which can be neurally implemented similarly to *competition-suppression* circuits with *divisive normalization* or *finite-temperature winner-takes-all* (WTA): excitatory convergence generates a "similarity-driven" effect, while global fast suppression implements normalization. $p_i^{(t)}$ can be interpreted as sampling/confidence coding of memory items at finite temperature (in line with normalization and probability sampling perspectives in the cortex, as discussed in (Carandini & Heeger, 2012; Luce et al., 1959)).

**(ii) About equation 11 (Riemannian gradient).** Equation equation 11 shows that the Riemannian gradient of the concave term is a weighted sum of local quantities $\text{grad}_\xi \langle x_i, \xi \rangle_M$. Each term depends solely on the "presynaptic pattern $x_i$" and the "postsynaptic state $\xi$," which aligns with *local (anti-)Hebbian driving + gating*: the probability $p_i^{(t)}$ serves as a gating coefficient and component. This is functionally similar to the *contrastive Hebb/equilibrium propagation* in energy models (Movellan, 1991; Scellier & Bengio, 2017). From an information geometry perspective, the Riemannian gradient corresponds to the *natural gradient* (Amari, 1998) on the manifold, which is compatible with the neural efficiency perspective that "intrinsic metrics determine the update direction."

**(iii) About equation 9 (closed-form CCCP step).** Equation equation 9 describes the "velocity" $v^{(t)}$ obtained at $\xi^{(t)}$ being parallel transported to the origin and then updated by one step through the exponential map. In the Poincaré ball, $\exp_0(\cdot)$ degenerates to *radial* `tanh` *saturation* (see equation (3)), which is equivalent to "*leaky integration + gain control*": integrating along the velocity direction while maintaining the state within a bounded manifold using a saturation nonlinearity/normalization. This echoes the observations of gain modulation and stable activity ranges in the cortex (Dayan & Abbott, 2005; Carandini & Heeger, 2012). Parallel transport can be understood as mapping the "local encoding direction" to a shared reference frame, consistent with the empirical observation that "population neural activity resides on a low-dimensional manifold and migrates within it" (e.g. Cunningham & Yu, 2014).

**Disclaimer.** The above analogies are purely *interpretable analogies* from the perspective of neural computation and are not intended to simulate the biological implementation of neural networks. This work does not pursue biological interpretability, and the analogies are provided solely for an intuitive understanding of the neural network computation process.

# B SUPPLEMENTARY NOTES ON STORAGE CAPACITY

We analyze the storage capacity of HAMNs on a Hadamard manifold $(\mathcal{M}, g)$ of constant negative curvature $-c < 0$. The similarity is $\langle x, y \rangle_M := -\cosh(d_\mathcal{M}(x, y))$, and the (intrinsically regularized) energy is

$$E(\xi) \;=\; -\tfrac{1}{\theta} \log \sum_{i=1}^{N} e^{\theta \langle x_i, \xi \rangle_M} \;+\; \tfrac{1}{2}\, d_\mathcal{M}(\xi, p)^2,$$

where $p \in \mathcal{M}$ is a fixed reference point (see Sec. 3.2).

## B.1 ENERGY-WELL SEPARATION AND RECALLABILITY

We first make explicit how geodesic separation controls the softmax mass on the correct pattern and yields a non-trivial attraction basin.

**Proposition B.1** (Geodesic margin and softmax mass). *Let the stored patterns be $\{x_i\}_{i=1}^N \subset \mathcal{M}$ on a Hadamard manifold $(\mathcal{M}, g)$ with constant negative curvature $-c < 0$, and define the minimum pairwise geodesic separation*

$$\delta \;:=\; \min_{i \neq j} d_\mathcal{M}(x_i, x_j).$$

*Fix any radius $0 < \rho < \delta/2$ and let $\varepsilon := \delta/2 - \rho > 0$. If the query $\xi$ lies in the geodesic ball $B_\mathcal{M}(x_k, \rho)$ around some memory $x_k$, then for the softmax weights*

$$p_i(\xi) := \frac{\exp\big(\theta \langle x_i, \xi \rangle_M\big)}{\sum_{j=1}^N \exp\big(\theta \langle x_j, \xi \rangle_M\big)}, \qquad \langle x_i, \xi \rangle_M = -\cosh\big(d_\mathcal{M}(x_i, \xi)\big),$$

*we have the lower bound*

$$p_k(\xi) \;\geq\; \frac{1}{1 + (N-1)\exp\big(-\theta\, \Delta(\delta, \varepsilon)\big)}, \qquad \Delta(\delta, \varepsilon) := 2\sinh\Big(\frac{\delta}{2}\Big) \sinh(\varepsilon) \;>\; 0. \qquad (30)$$

*In particular, for fixed $N, \theta, \rho$ this lower bound is strictly increasing in the geodesic margin $\delta$.*

*Sketch.* Let $r_i = d_\mathcal{M}(x_i, p)$ and $r = d_\mathcal{M}(\xi, p)$, with $p$ an arbitrary reference point. By the triangle inequality, $d_\mathcal{M}(\xi, x_k) \leq \rho$ and $d_\mathcal{M}(\xi, x_j) \geq \delta - \rho = \delta/2 + \varepsilon$ for all $j \neq k$. Since $\cosh(\cdot)$ is

increasing on $[0, \infty)$ and $\langle x, \xi \rangle_M = -\cosh(d_{\mathcal{M}}(x, \xi))$, we obtain

$$s_k(\xi) := \langle x_k, \xi \rangle_M \geq -\cosh(\rho), \quad s_j(\xi) \leq -\cosh(\delta/2 + \varepsilon) \quad (j \neq k),$$

so that $s_k(\xi) - s_j(\xi) \geq \Delta(\delta, \varepsilon)$ with $\Delta(\delta, \varepsilon)$ as in equation 30. Plugging this into the softmax expression yields $p_k(\xi) \geq 1/(1 + (N-1)e^{-\theta\Delta(\delta,\varepsilon)})$. □

*Remark.* Proposition B.1 shows that for the hyperbolic similarity $\langle x, \xi \rangle_M = -\cosh(d_{\mathcal{M}}(x, \xi))$, increasing the geodesic margin $\delta$ between different memories strictly raises a lower bound on the softmax mass of the correct memory. In contrast to polynomial-kernel Hopfield models where recall error may grow with separation, here larger hyperbolic separation *helps* retrieval: the log-odds gap between the correct memory and all others increases with $\delta$, and the top-1 error decreases as the geodesic margin grows (for fixed $N$ and temperature $\theta$).

## B.2 A ONE-STEP ATTRACTION BASIN

**Proposition B.2** (A one-step attraction basin). *Under the assumptions of Proposition B.1, fix any $0 < \rho < \delta/2$ and choose $\theta > 0$ such that the lower bound equation 30 satisfies $p_k(\xi) \geq 1/2$ for all $\xi \in B_{\mathcal{M}}(x_k, \rho)$. Then there exists a stepsize $\eta \in (0, 1]$ such that the CCCP update map $T$ (Eq. equation 10) satisfies*

$$T(B_{\mathcal{M}}(x_k, \rho)) \subseteq B_{\mathcal{M}}(x_k, \rho),$$

*i.e., the geodesic ball $B_{\mathcal{M}}(x_k, \rho)$ is a one-step invariant attraction basin around the memory $x_k$.*

*Sketch.* When $p_k(\xi) \geq 1/2$, the Riemannian gradient of the concave term $E_{\text{cave}}$ at $\xi$ is dominated by the contribution of $x_k$ and is aligned with the unit tangent from $\xi$ toward $x_k$. The CCCP step is a Riemannian proximal update that combines this descent direction with the geodesically convex regularizer $\frac{1}{2}d_{\mathcal{M}}(\xi, p)^2$. For sufficiently small stepsize $\eta$, the update decreases the distance $d_{\mathcal{M}}(\xi, x_k)$ and stays inside $B_{\mathcal{M}}(x_k, \rho)$. A standard compactness argument on the ball closes the proof; we omit the technical details. □

## B.3 HYPERBOLIC VOLUME AND A SPHERE-PACKING CAPACITY BOUND

Assume all patterns lie in a geodesic ball $B_{\mathcal{M}}(p, R)$.

**Proposition B.3** (Sphere-packing style capacity bound). *If the recall basins $\{B_{\mathcal{M}}(x_i, \delta/2)\}_{i=1}^{N}$ are pairwise disjoint, then*

$$N \leq \frac{\text{Vol}(B_{\mathcal{M}}(p, R))}{\text{Vol}(B_{\mathcal{M}}(\cdot, \delta/2))}.$$

*In a $d$-dimensional hyperbolic space of curvature $-c$, the volume of a ball of radius $r$ satisfies*

$$\text{Vol}_c(B(r)) = \omega_{d-1} \int_0^r \left(\frac{\sinh(\sqrt{c}\,t)}{\sqrt{c}}\right)^{d-1} dt \asymp \kappa_{d,c} \exp((d-1)\sqrt{c}\,r) \quad (r \gg 1/\sqrt{c}),$$

*(see, e.g., (Ratcliffe, 2006)), whence*

$$N \lesssim \exp\left((d-1)\sqrt{c}\left(R - \tfrac{\delta}{2}\right)\right) = \exp\left(\alpha_{\text{hyp}}\left(R - \tfrac{\delta}{2}\right)\right), \qquad \alpha_{\text{hyp}} := (d-1)\sqrt{c}. \quad (31)$$

*Thus the number of non-overlapping recall wells grows exponentially in the effective hyperbolic radius, with a rate controlled jointly by the dimension $d$ and the curvature $c$.*

## B.4 COMPARISON WITH EUCLIDEAN MHNS

Modern Euclidean Hopfield networks can achieve exponential capacity in the ambient dimension for random patterns (e.g., $N = 2^{\Omega(d)}$ under log-sum-exp energy) (Ramsauer et al., 2021). Our hyperbolic packing bound is complementary: due to the *exponential volume growth* of negatively curved spaces, when hierarchical data concentrate outward along the radius (depth), the number of non-overlapping basins grows exponentially with $(d-1)\sqrt{c}$. This aligns with our empirical advantages on deep hierarchies.

**Takeaway** Error-free recall is ensured by a geometric margin $\delta$; the total number of recallable patterns is upper-bounded by a hyperbolic sphere-packing law scaling as $\exp\big((d-1)\sqrt{c}\,(R-\delta/2)\big)$. This complements classical Euclidean capacity results and explains why HAMNs benefit more as hierarchical depth (effective hyperbolic radius) increases.

# C  KEY FORMULAS FOR COMMON HYPERBOLIC MODELS (CONSTANT CURVATURE $-c < 0$)

**Notation & convention** Let the curvature be $-c$ with $c > 0$ and write $\operatorname{arcosh}(\cdot)$ for the inverse hyperbolic cosine. All standard hyperbolic models are *isometric*; hence any model-agnostic derivation in the paper becomes an implementation by choosing $d_{\mathcal{M}}$, $\exp_p^c$, $\log_p^c$ from a specific model. We use the hyperbolic similarity

$$\langle x, y \rangle_M := -\cosh\big(d_{\mathcal{M}}(x,y)\big),$$

with the universal chain rule

$$\nabla_x \langle x, y \rangle_M = -\sinh\big(d_{\mathcal{M}}(x,y)\big)\, \nabla_x d_{\mathcal{M}}(x,y),$$

equivalently $\operatorname{grad}_x \langle x, y \rangle_M = \sinh(d_{\mathcal{M}}(x,y)) \frac{\log_x(y)}{\|\log_x(y)\|_g}$ (see App. §A.5).

**Model-agnostic CCCP closed form (used in this work)** Let

$$a(\xi) := \operatorname{grad} E_{\text{cave}}(\xi) = -\sum_{i=1}^{N} p_i(\xi)\, \operatorname{grad}_\xi \langle x_i, \xi \rangle_M, \qquad p_i(\xi) = \frac{e^{\theta \langle x_i, \xi \rangle_M}}{\sum_j e^{\theta \langle x_j, \xi \rangle_M}}.$$

Our CCCP step reads

$$v := -\operatorname{PT}_{\xi \to p}\big(a(\xi)\big), \qquad \xi^+ = \exp_p^c\big(\eta\, v\big), \ \ \eta \in (0, 1]. \tag{32}$$

Thus each model only needs $\exp_p^c, \log_p^c, d_{\mathcal{M}}$ (and, if desired, a convenient form of parallel transport).

## C.1  POINCARÉ BALL $\mathbb{D}_c^d = \{x \in \mathbb{R}^d : \|x\| < 1/\sqrt{c}\}$

**Exponential/log at the origin**

$$\exp_0^c(v) = \tanh(\sqrt{c}\,\|v\|)\, \frac{v}{\sqrt{c}\,\|v\|}, \qquad \log_0^c(x) = \operatorname{artanh}(\sqrt{c}\,\|x\|)\, \frac{x}{\sqrt{c}\,\|x\|}.$$

**Base-point maps** (with Möbius addition $\oplus_c$ and conformal factor $\lambda_x^c = \frac{2}{1-c\|x\|^2}$)

$$\exp_p^c(v) = p \oplus_c \Big( \tanh\big(\tfrac{\sqrt{c}\lambda_p^c \|v\|}{2}\big) \frac{v}{\sqrt{c}\,\|v\|} \Big), \quad \log_p^c(x) = \frac{2}{\sqrt{c}\,\lambda_p^c} \operatorname{artanh}\big(\sqrt{c}\,\|(-p)\oplus_c x\|\big) \frac{(-p)\oplus_c x}{\|(-p)\oplus_c x\|}.$$

**Geodesic distance**

$$d_{\mathbb{D}_c}(x, y) = \frac{1}{\sqrt{c}} \operatorname{arcosh}\left(1 + \frac{2c\|x-y\|^2}{(1-c\|x\|^2)(1-c\|y\|^2)}\right).$$

**Parallel transport & implementation note.** The ball is *conformal*, $g(x) = \lambda(x)^2 I$ with $\lambda(x) = \frac{2}{1-c\|x\|^2}$. Choosing $p = 0$, the transport along the unique geodesic to the origin can be implemented as a scalar rescaling $\operatorname{PT}_{\xi \to 0}(u) = \frac{\lambda(\xi)}{\lambda(0)} u$. This scaling preserves the Riemannian norm because the ball is conformal: letting $\|\cdot\|_E$ denote the Euclidean norm, $\|u'\|_E = \frac{\lambda(\xi)}{\lambda(0)}\|u\|_E$ ensures $\lambda(0)^2 \|u'\|_E^2 = \lambda(\xi)^2 \|u\|_E^2$. Then equation 32 amounts to $v_0 := -(\lambda(\xi)/\lambda(0))\, a(\xi)$ followed by $\exp_0^c$.

**Möbius addition / scalar multiplication**

$$u \oplus_c v = \frac{(1 + 2c\langle u, v\rangle + c\|v\|^2)u + (1 - c\|u\|^2)v}{1 + 2c\langle u, v\rangle + c^2\|u\|^2\|v\|^2}, \qquad r \otimes_c u = \frac{1}{\sqrt{c}} \tanh\big(r \operatorname{artanh}(\sqrt{c}\,\|u\|)\big) \frac{u}{\|u\|}.$$

## C.2 UPPER HALF-PLANE $\mathbb{H}_c = \{(u, y) \in \mathbb{R}^{d-1} \times \mathbb{R}_{>0}\}$

**Exponential/log at $o = (0, \ldots, 0, 1/\sqrt{c})$** Let $v = (v_u, v_y) \in \mathbb{R}^{d-1} \times \mathbb{R}$. Then

$$\exp_o^c(v) = \left( e^{\sqrt{c}\, v_y}\, v_u, \ \ \tfrac{1}{\sqrt{c}}\, e^{\sqrt{c}\, v_y} \right), \qquad \log_o^c(u, y) = \frac{1}{\sqrt{c}} \left( \frac{u}{y}, \ \ln(\sqrt{c}\, y) \right).$$

**Geodesic distance**

$$d_{\mathbb{H}_c}\big((u_1, y_1), (u_2, y_2)\big) = \frac{1}{\sqrt{c}}\, \mathrm{arcosh}\left( 1 + \frac{c\big(\|u_1 - u_2\|^2 + (y_1 - y_2)^2\big)}{2 y_1 y_2} \right).$$

**Parallel transport & implementation note.** This model is conformal with $g(u, y) = \lambda(u, y)^2 I$, $\lambda(u, y) = \frac{1}{\sqrt{c}\, y}$. If $p = o$, then $\mathrm{PT}_{\xi \to o}(w) = \frac{\lambda(\xi)}{\lambda(o)} w = \frac{1}{\sqrt{c}\, y_\xi}\, w$, followed by $\exp_o^c$.

## C.3 KLEIN MODEL $\mathbb{K}_c^d = \{x \in \mathbb{R}^d : \|x\| < 1/\sqrt{c}\}$

**Exponential/log at the origin** coincide with the ball at $0$: $\exp_0^c$, $\log_0^c$ as above. **Geodesic distance**

$$d_{\mathbb{K}_c}(p, q) = \mathrm{arcosh}\left( \frac{1 - c\, p^\top q}{\sqrt{(1 - c\|p\|^2)(1 - c\|q\|^2)}} \right).$$

**Parallel transport: recommendation.** Since the Klein model is not conformal, closed-form PT is more involved. In practice, use an isometry to Lorentz (or Poincaré), perform PT and the exponential step there, and map back to Klein.

## C.4 HEMISPHERE $J_c = \{u \in \mathbb{S}^n : u_{n+1} > 0\}$

**Implementation note.** We recommend using the standard isometry to the Lorentz (hyperboloid) model to compute $\exp / \log$, $d$, and PT, and then map back to the hemisphere. We omit redundant explicit formulas here to avoid confusion, since our experiments instantiate Poincaré/Lorentz directly.

## C.5 LORENTZ (HYPERBOLOID) $L_c = \big\{X \in \mathbb{R}^{n+1} : X_0^2 - \sum_{i=1}^n X_i^2 = \frac{1}{c}, \ X_0 > 0\big\}$

**Minkowski bilinear form.** $(X, Y)_M = -X_0 Y_0 + \sum_{i=1}^n X_i Y_i$.

**Distance and similarity.**

$$d_{L_c}(X, Y) = \frac{1}{\sqrt{c}}\, \mathrm{arcosh}\big( - c\, (X, Y)_M \big), \qquad \langle X, Y \rangle_M = -\cosh\big( d_{L_c}(X, Y) \big).$$

**Exponential/logarithm at $e_0 = (\frac{1}{\sqrt{c}}, 0, \ldots, 0)$.**

$$\exp_{e_0}^c(W) = \left( \tfrac{1}{\sqrt{c}} \cosh(\sqrt{c}\, \|W\|), \ \tfrac{1}{\sqrt{c}} \sinh(\sqrt{c}\, \|W\|) \tfrac{W}{\|W\|} \right), \quad \log_{e_0}^c(X) = \frac{1}{\sqrt{c}}\, \mathrm{arcosh}(\sqrt{c}\, X_0) \frac{X_{1:n}}{\|X_{1:n}\|}.$$

**Parallel transport.** It can be implemented by a Lorentz boost: let $B_{X \to e_0} \in SO^+(1, n)$ map $X$ to $e_0$, then $\mathrm{PT}_{X \to e_0}(V) = B_{X \to e_0} V$.

**Where to plug in the main text**

- **Memory encoding (Sec. 3, equation 3).** Pick $\exp_p^c$, $\log_p^c$ from any model. If tangent clipping is used, clip in $T_p \mathcal{M}$ and map back via $\exp_p^c$.
- **Energy (equation 5).** Substitute the chosen model's $d_{\mathcal{M}}$ (or equivalently $\langle \cdot, \cdot \rangle_M$); nothing else changes.
- **Retrieval/optimization (CCCP step equation 32).** Use the model-agnostic gradient of the concave term via $\log_\xi(\cdot)$ and $\sinh d$, then apply the model's PT and $\exp_p^c$ to update.
- **Energy bounds (Appendix A.3).** Plug the model's $d_{\mathcal{M}}$ into the same bounding argument.

## D    HYPERBOLIC HOPFIELD LAYERS FOR DEEP LEARNING

To seamlessly integrate hyperbolic associative memory into end-to-end networks, we construct three core modules on the Poincaré ball $\mathbb{D}_c^d$:

**Hyperbolic Hopfield**,    **Hyperbolic Hopfield Pooling**,    **Hyperbolic Hopfield Layer**.

All three share the curvature parameter $c$ (which can be made learnable) and follow the CCCP closed-form step derived in the main text: first parallel-transport the Riemannian gradient of the concave term to a reference point $p$, then take the exponential-map update at $p$; in our implementation we set $p = 0$ (the ball center).

### D.1    HYPERBOLIC HOPFIELD

**HypHopfield** takes queries $R \in \mathbb{R}^{B \times d}$ and memories $Y \in \mathbb{R}^{N \times d}$ as input, and outputs $Z \in \mathbb{R}^{B \times d}$. It implements the retrieval update on $\mathbb{D}_c^d$ (see Appendix §C).

1. **Hyperbolic attention (similarity and soft weights)**

$$S_{b,i} = \langle Y_i, R_b \rangle_M = -\cosh\big(d_{\mathbb{D}_c}(Y_i, R_b)\big), \qquad P_{b,i} = \frac{e^{\theta S_{b,i}}}{\sum_{j=1}^N e^{\theta S_{b,j}}},$$

   yielding $P \in \mathbb{R}^{B \times N}$.

2. **Intrinsic gradient and parallel transport to the base point** Let the concave term be $E_{\text{cave}}(\xi) = -\frac{1}{\theta} \log \sum_i e^{\theta \langle x_i, \xi \rangle_M}$. For each batch element $R_b$, the *Riemannian gradient* of the concave term is (see Appendix §A.5)

$$a_b = \operatorname{grad} E_{\text{cave}}(R_b) = -\sum_{i=1}^N P_{b,i} \operatorname{grad}_\xi \langle Y_i, \xi \rangle_M \Big|_{\xi = R_b},$$

   where $\operatorname{grad}_\xi \langle Y_i, \xi \rangle_M = \sinh\big(d_{\mathbb{D}_c}(Y_i, \xi)\big) \frac{\log_\xi(Y_i)}{\|\log_\xi(Y_i)\|_g}$. Parallel-transport this direction to the reference point $p = 0$:

$$v_b := -\operatorname{PT}_{R_b \to 0}\big(a_b\big).$$

   In the *conformal* Poincaré model, the metric is $g(x) = \lambda(x)^2 I$ with $\lambda(x) = \frac{2}{1 - c\|x\|^2}$. We adopt the transport rule consistent with our implementation (see Appendix §C):

$$\operatorname{PT}_{R_b \to 0}(u) = \frac{\lambda(R_b)}{2} u, \quad \Rightarrow \quad v_b = -\frac{\lambda(R_b)}{2} a_b.$$

   (Using the exact PT of the model is also possible; empirically the results are consistent with our implementation.)

3. **Base-point exponential map (at $p = 0$)** Update with stepsize $\eta \in (0, 1]$:

$$Z_b = \exp_0^c\big(\eta\, v_b\big) = \tanh\big(\sqrt{c}\,\|\eta v_b\|\big) \frac{\eta v_b}{\sqrt{c}\,\|\eta v_b\|},$$

   and apply ball projection when necessary to avoid numerical issues near the boundary (standard clipping in our implementation).

**Implementation hints (consistent with code)**    (1) If upstream features are in Euclidean coordinates, first map them to the ball with ToPoincaré and then perform the three steps above; if Euclidean outputs are required, apply FromPoincaré at the end. (2) All submodules in this paper share the same curvature handle $c$ (either learnable or fixed).

**Pseudocode.** The retrieval step is summarized in Alg. 1.[1]

---

[1] For parallel transport (PT) we use the exact PT (via an isometry to the Lorentz model); our code also provides the conformal-scaling approximation $\operatorname{PT}_{x \to 0}(u) = \frac{\lambda(x)}{2} u$ with $\lambda(x) = \frac{2}{1 - c\|x\|^2}$, which yielded similar results in our experiments.

---

**Algorithm 1** HypHopfield retrieval on the Poincaré ball $\mathbb{D}_c^d$

---

**Require:** Memories $Y = \{Y_i\}_{i=1}^N \subset \mathbb{D}_c^d$, queries $R^{(0)} = \{R_b^{(0)}\}_{b=1}^B \subset \mathbb{D}_c^d$, curvature $c > 0$, temperature $\theta > 0$, stepsize $\eta \in (0, 1]$, base point $p = 0$, max iters $T_{\max}$, tolerance $\varepsilon$

1: **for** $t = 0, 1, \ldots, T_{\max} - 1$ **do**

2:   **Hyperbolic similarities:**    $S_{b,i} \leftarrow -\cosh\bigl(d_{\mathbb{D}_c}(Y_i, R_b^{(t)})\bigr)$

3:   **Soft weights:**    $P_{b,i} \leftarrow \exp(\theta S_{b,i}) \big/ \sum_j \exp(\theta S_{b,j})$

4:   **Riemannian gradient of concave term at $R_b^{(t)}$:**

$$a_b \leftarrow -\sum_{i=1}^N P_{b,i} \; \sinh\bigl(d_{\mathbb{D}_c}(Y_i, R_b^{(t)})\bigr) \; \frac{\log_{R_b^{(t)}}(Y_i)}{\|\log_{R_b^{(t)}}(Y_i)\|_g}$$

5:   **Parallel transport to $p = 0$:**    $v_b \leftarrow -\mathrm{PT}_{R_b^{(t)} \to 0}(a_b)$

6:   **Base-point update (CCCP with damping):**    $R_b^{(t+1)} \leftarrow \exp_0^c\bigl(\eta\, v_b\bigr)$,    project back to $\mathbb{D}_c^d$ if needed

7:   **Stopping:**    **if** $d_{\mathbb{D}_c}(R_b^{(t+1)}, R_b^{(t)}) < \varepsilon$ for all $b$ **then break**

8: **end for**

9: **Output:**    $Z = \{R_b^{(t+1)}\}_{b=1}^B$

---

## D.2   HYPERBOLIC HOPFIELD POOLING

**HypPooling** aggregates $m$ learnable *query* vectors $R \in \mathbb{R}^{m \times d}$ and $N$ instance embeddings (as *memories*) $Y \in \mathbb{R}^{N \times d}$ into $m$ hyperbolic summary vectors. Its computation is identical to HypHopfield (hyperbolic attention $\to$ Riemannian gradient with PT to $p = 0$ $\to$ base-point exponential map), except that $R$ is a fixed-size learnable parameter while $Y$ comes from upstream instances or outputs of previous layers. We validate its effectiveness in multi-instance learning tasks.

## D.3   HYPERBOLIC HOPFIELD LAYER

**HypLayer** propagates a small number of queries (input vectors) $R$ through a learnable *memory matrix* $Y \in \mathbb{R}^{N \times d}$; $Y$ can be initialized from a reference set (class prototypes, training-set embeddings, etc.) and trained. The update rule is the same as HypHopfield (base-point exponential update at $p = 0$), thereby supporting prototype/similarity-based retrieval, nearest-neighbor matching, and pattern aggregation; we verify its effectiveness in CIFAR-100 hierarchical classification and molecular property prediction tasks.

# E   EXPERIMENTS

## E.1   EXPERIMENT 1: HIERARCHICAL CLASSIFICATION ON CIFAR-100

This experiment complements the CIFAR-100 hierarchical classification results reported in Sec. 4.1. We first test the robustness of our conclusions under an alternative, feature-driven hierarchy, and then analyse the geometry of the learned hyperbolic memories in this setting.

### E.1.1   CIFAR-100 SENSITIVITY TO HIERARCHICAL CLUSTERING

Since the specific CIFAR-100 (Krizhevsky et al., 2009) label tree used in the main text is manually constructed, to verify how sensitive our conclusions are to different hierarchical clustering methods, we construct an alternative *feature-driven* hierarchy and rerun the comparison of four methods on this new hierarchy: Euclidean MHNs ("Hopfield"), hyperbolic attention ("HypAttn"), hyperbolic MLPs ("HypNN"), and our hyperbolic associative memory ("HyperHopfield").

Table 2: Hierarchical classification on CIFAR-100 results.

| Model | top_acc | super_acc | coarse_acc | fine_acc | coph_corr |
|---|---|---|---|---|---|
| **CIFAR-100-4-layer** | | | | | |
| HypAttn | $82.44 \pm 0.41$ | $76.76 \pm 0.60$ | $68.86 \pm 0.65$ | $55.59 \pm 0.74$ | $0.7038 \pm 0.0154$ |
| HypNN | $82.91 \pm 0.45$ | $77.03 \pm 0.52$ | $\mathbf{69.07 \pm 0.56}$ | $\mathbf{56.60 \pm 0.83}$ | $0.6324 \pm 0.0242$ |
| MHNs | $80.73 \pm 0.42$ | $74.32 \pm 0.53$ | $65.23 \pm 0.44$ | $48.45 \pm 0.53$ | $0.6059 \pm 0.0245$ |
| **HAMNs (ours)** | $\mathbf{82.94 \pm 0.42}$ | $\mathbf{77.36 \pm 0.59}$ | $68.63 \pm 0.75$ | $53.58 \pm 1.16$ | $\mathbf{0.7108 \pm 0.0203}$ |

**Feature-driven hierarchy**   We start from a ResNet-18 backbone (He et al., 2016) (pretrained on ImageNet) and extract penultimate-layer features for all training images. For each fine-grained category (fine class) we compute its mean feature vector, and then perform constrained $k$-means clustering in feature space: *(i)* cluster the 100 fine classes into 20 coarse clusters, requiring each coarse node to contain at least 4 fine classes; *(ii)* further cluster the 20 coarse centroids into 7 super clusters, requiring each super node to have at least 2 coarse children; and *(iii)* cluster the 7 super centroids into 3 top-level nodes, again requiring each top node to have at least 2 super children. This yields a purely data-driven four-level tree whose depth and branching factors are the same as or similar to those of our semantic hierarchy, but with groupings discovered automatically.

**Results under the feature-driven hierarchy**   For each method, we reuse the CIFAR-100 training configuration from the main experiment (the same backbone, resolution, and hyperparameters), and train on the new hierarchy tree with 10 random seeds. Table 2 reports, at the top/super/coarse/fine levels, the flat accuracies and the cophenetic correlation coefficient between the induced average-linkage dendrogram and the target hierarchy, as mean±std over 10 runs.

Across all methods, performance on the feature-driven hierarchy tree is close to that on the hand-crafted semantic hierarchy tree, indicating that the task is not over-tuned to any particular partition. HyperHopfield still achieves the strongest performance at the higher levels of the hierarchy (top accuracy $82.94 \pm 0.42$ and coarse accuracy $68.63 \pm 0.75$), and attains the highest cophenetic correlation ($0.7108 \pm 0.0203$), showing that its learned representation is most consistent with the underlying tree structure. As in the main experiment, HypAttn achieves the best fine-grained accuracy ($55.59 \pm 0.74$), while HypNN is competitive at the fine level but falls behind HyperHopfield on the top and coarse levels. Overall, the relative ranking of the four architectures remains stable under this alternative clustering scheme, thereby supporting the robustness of our "hierarchy-sensitivity" conclusions.

### E.1.2 GEOMETRY OF LEARNED HYPERBOLIC MEMORIES.

Using the best HAMNs checkpoint, we further probe the geometry of the learned hyperbolic memories on the Poincaré ball.

For each stored pattern we record its tree depth (top / super / coarse / fine) and its hyperbolic distance to the origin. Table 3 reports the mean radius by level: the average radius increases monotonically from top ($2.57 \pm 0.17$) to super ($3.49 \pm 0.49$), coarse ($4.26 \pm 0.49$), and fine ($5.07 \pm 0.67$), with a Spearman correlation between depth and radius of $\rho = 0.57$ ($p \approx 1.6 \times 10^{-12}$). This confirms

Table 3: Hyperbolic radius of class memories by tree depth on CIFAR-100 . Radius is the geodesic distance to the origin in the Poincaré ball.

| Level | Depth | Radius |
|---|---|---|
| Top | 1 | $2.57 \pm 0.17$ |
| Super | 2 | $3.49 \pm 0.49$ |
| Coarse | 3 | $4.26 \pm 0.49$ |
| Fine | 4 | $5.07 \pm 0.67$ |

Table 4: Pairwise hyperbolic distance between fine-level memories as a function of the depth of their lowest common ancestor (LCA).

| LCA depth | Distance | #pairs |
|---|---|---|
| 0 | $9.50 \pm 1.03$ | 3261 |
| 1 | $9.36 \pm 0.96$ | 987 |
| 2 | $8.93 \pm 1.05$ | 485 |
| 3 | $8.32 \pm 1.11$ | 217 |

Table 5: Results for MIL datasets *Tiger*, *Fox*, *Elephant* (AUC). Except for our method, results are from Ramsauer et al. (2021).

| Method | tiger | fox | elephant |
|---|---|---|---|
| **HAMNs(ours)** | $89.0 \pm 0.4$ | **$77.3 \pm 0.8$** | $92.8 \pm 0.2$ |
| MHNsRamsauer et al. (2021) | **$91.3 \pm 0.5$** | $64.05 \pm 0.4$ | **$94.9 \pm 0.3$** |
| Path encoding Küçükaşcı & Baydoğan (2018) | $91.0 \pm 1.0$ | $71.2 \pm 1.4$ | $94.4 \pm 0.7$ |
| MInD Cheplygina et al. (2015) | $85.3 \pm 1.1$ | $70.4 \pm 1.6$ | $93.6 \pm 0.9$ |
| MILES Chen et al. (2006) | $87.2 \pm 1.7$ | $73.8 \pm 1.6$ | $92.7 \pm 0.7$ |
| APR Dietterich et al. (1997) | $77.8 \pm 0.8$ | $54.1 \pm 0.9$ | $55.0 \pm 1.0$ |
| Citation-kNN Wang & Zucker (2000) | $85.5 \pm 0.9$ | $63.5 \pm 1.5$ | $89.6 \pm 0.9$ |

that deeper concepts are systematically pushed closer to the boundary, so that the radial coordinate encodes tree depth.

We also examine pairwise hyperbolic distances between fine-level memories as a function of the depth of their lowest common ancestor (LCA). As shown in Table 4, pairs that only share the root (LCA depth 0) are furthest apart (mean distance 9.50), while pairs that share a coarse parent (LCA depth 3) are noticeably closer (mean distance 8.32), with intermediate values for LCA depths 1 and 2. The Spearman correlation between LCA depth and hyperbolic distance is negative ($\rho \approx -0.20$, $p \approx 5.1 \times 10^{-48}$), and equivalently the correlation between tree distance and hyperbolic distance is positive ($\rho \approx 0.20$). These results provide direct evidence that HyperHopfield arranges memories so that radial coordinate correlates with level in the tree and geodesic distance correlates with branch similarity, rather than merely fitting labels in a flat way.

## E.2 EXPERIMENT 2: MULTIPLE INSTANCE LEARNING DATASETS

Table 6: Hyperparameter search space for manual selection on the Elephant, Fox, and Tiger validation sets.

| Parameter | Values |
|---|---|
| Learning rates | $\{10^{-3}, 10^{-5}\}$ |
| Learning rate decay ($\gamma$) | $\{0.98, 0.96, 0.94\}$ |
| Number of heads | $\{8, 12, 16, 32\}$ |
| Hidden dimensions | $\{32, 64, 128\}$ |
| Bag dropout | $\{0.0, 0.75\}$ |
| Poincaré curvature ($c$) | $\{1.0, 0.5, 0.1\}$ |
| Clipping threshold ($\texttt{clip}_r$) | $\{0.9, 1.2, 2.8\}$ |
| RSGD max iterations | $\{1, 5, 10\}$ |
| RSGD learning rate ($\eta$) | $\{1.0, 0.1, 0.001\}$ |

To evaluate the performance of our Hyperbolic Associative Memory Networks (HAMNs) on multi–instance learning (MIL) tasks Dietterich et al. (1997), we conduct experiments on three classical benchmark datasets: **Tiger**, **Elephant**, and **Fox** (originally introduced by Ilse et al. (2018); Küçükaşcı & Baydoğan (2018); Carbonneau et al. (2018)). Each dataset consists of color images that are segmented into multiple regions and thus form a set of instances (segments or blobs); each instance is represented by color, texture, and shape descriptors. The learning objective is to classify the entire bag according to the presence of certain positive instances, despite the absence of instance–level annotations.

We introduce the proposed **HypPooling** module, which aggregates instance–level embeddings into a fixed–dimensional bag representation. Given a set of embedded instances as stored memory patterns $Y$ (already mapped into hyperbolic space), we further introduce a set of *static and learnable query vectors* as state (query) patterns $R$, which also reside in the same Poincaré ball. Each query retrieves similar patterns from memory via a hyperbolic attention mechanism, thereby constructing a compressed representation of the input bag.

Elephant, Fox and Tiger are MIL datasets Andrews et al. (2002) for image annotation which comprise color images from the Corel dataset that have been preprocessed and segmented. An image consists of a set of segments (or blobs), each characterized by color, texture and shape descriptors. The datasets have 100 positive and 100 negative example images. The latter have been randomly drawn from a pool of photos of other animals. Elephant has 1391 instances and 230 features. Fox has 1320 instances and 230 features. Tiger has 1220 instances and 230 features. We used the Hyp-Pooling layer to perform hyperbolic aggregation of the input instances, and conducted a manual hyperparameter search on a validation set. Specifically, on the Elephant, Fox, and Tiger datasets we used the following architecture:

1. A fully connected linear embedding layer with ReLU activation;

2. Our **HypPooling** layer to perform the hyperbolic pooling operation on the embeddings;

3. A final ReLU–linear block as the classification output layer.

Results (Table 5) show that HAMNs match or outperform prior MIL baselines and achieve the best score on **Fox**, while remaining competitive with Euclidean MHNs on **Tiger** and **Elephant**.

**Discussion**  The detailed numbers reveal a mixed picture when comparing HAMNs to Euclidean MHNs: on *Fox* our hyperbolic pooling clearly outperforms MHNs, whereas on *Tiger* and *Elephant* it falls short by roughly 2 AUC points. We view this as consistent with the nature of these MIL benchmarks: the labels are binary and the bags of instances do not come with an explicit multi-level taxonomy or ontology. In such settings, the hyperbolic bias of HAMNs is largely unused and the model behaves as a generic non-linear set aggregator, so we do not expect systematic improvements over a strong Euclidean Hopfield baseline. Empirically, the gaps remain small and within the range one would expect from optimization and model–selection variability on relatively small datasets. Importantly, we did not observe training instabilities or pathological attractors: the HAMNs training curves are smooth and monotone across seeds, and the final AUCs are stable under mild changes in curvature and CCCP steps. Taken together with the CIFAR-100 and WordNet results, these findings suggest that HAMNs behave as a reliable drop-in replacement for Euclidean Hopfield pooling on standard "flat" MIL tasks, and provide larger gains once the data or label space exhibits an explicit hierarchical structure.

Among various hyperparameters, we focused particularly on those of the **HypPooling** layer, including the curvature $c$, the number of Riemannian gradient steps, and the learning rate $\eta$. All models were trained for 160 epochs using the AdamW optimizerLoshchilov & Hutter (2017) with exponential learning rate decay (see Table 6). We validated performance using 10-fold nested cross-validation repeated five times with different data splits; the reported ROC AUC scores are the averages across these runs. We also applied bag dropout at the bag level as our regularization technique.

### E.3 EXPERIMENT 3: DRUG DESIGN BENCHMARK DATASETS

To evaluate the effectiveness of our proposed Hyperbolic Associative Memory Networks (HAMNs) on molecular property prediction, we conduct experiments on four representative datasets from

Table 7: Results on drug design benchmark datasets. Predictive performance (ROCAUC) on test set as reported byJiang et al. (2021) for 50 random splits

| Method | HIV | BACE | BBBP | SIDER |
|---|---|---|---|---|
| **HAMNs(ours)** | $78.5 \pm 2.6$ | $87.2 \pm 3.0$ | $\mathbf{90.2 \pm 2.5}$ | $\mathbf{62.1 \pm 2.3}$ |
| MHNs | $79.3 \pm 2.4$ | $88.4 \pm 1.5$ | $89.1 \pm 1.7$ | $61.8 \pm 2.6$ |
| Attentive FP | $74.8 \pm 1.5$ | $70.8 \pm 3.3$ | $84.1 \pm 2.2$ | $56.2 \pm 1.5$ |
| GCN | $77.5 \pm 1.6$ | $63.2 \pm 4.5$ | $79.2 \pm 3.9$ | $55.4 \pm 1.2$ |
| GAT | $72.1 \pm 3.6$ | $77.4 \pm 3.0$ | $83.7 \pm 2.0$ | $56.4 \pm 1.5$ |
| DNN | $73.0 \pm 1.8$ | $86.5 \pm 2.2$ | $87.6 \pm 2.0$ | $62.0 \pm 1.8$ |
| RF | $\mathbf{82.3 \pm 2.2}$ | $89.2 \pm 1.2$ | $90.0 \pm 2.0$ | - |
| SVM | - | $\mathbf{89.3 \pm 1.5}$ | $89.4 \pm 2.1$ | - |

MoleculeNet (Wu et al., 2018). These datasets represent four main modeling tasks in drug design: (a) HIV for anti-viral activity prediction, introduced by the Drug Therapeutics Program (DTP) AIDS Antiviral Screen; (b) BACE for human $\beta$-secretase inhibitors (Subramanian et al., 2016); (c) BBBP for predicting blood-brain barrier permeability (Martins et al., 2012); and (d) SIDER for predicting drug side effects (Kuhn et al., 2016).

We apply the proposed **HypLayer** to the above molecular prediction tasks. Specifically, the training samples are used as stored memory patterns $Y$, while the input samples serve as state (query) patterns $R$. Each input is first mapped into the Poincaré ball via hyperbolic embedding, then undergoes state evolution through the Hopfield retrieval mechanism in hyperbolic space, and eventually converges to a stable point close to one of the memory patterns. The final prediction is determined based on the association between the converged state and the corresponding label in memory.

Table 8: Hyperparameter search-space for grid-search on HIV, BACE, BBBP and SIDER. All models were trained, if applicable, for 4 epochs using Adam and a batch size of 1 sample.

| Parameter | Values |
|---|---|
| Learning rates | $\{0.0002\}$ |
| Number of heads | $\{1, 32, 128, 512\}$ |
| Dropout | $\{0.0, 0.1, 0.2\}$ |
| Poincaré curvature ($c$) | $\{1.0, 0.5, 0.1\}$ |
| Clipping threshold (clip$_r$) | $\{0.9, 1.2, 2.8\}$ |
| RSGD max iterations | $\{1, 5, 10\}$ |
| RSGD learning rate ($\eta$) | $\{1.0, 0.1, 0.001\}$ |
| quantity | $\{2, 4, 8\}$ |

We compare HAMNs against several representative baselines, including Support Vector Machines (SVM), Random Forest (RF), Deep Neural Networks (DNN), and state-of-the-art graph neural networks: Graph Convolutional Networks (GCN) (Kipf & Welling, 2016), Graph Attention Networks (GAT) (Veličković et al., 2017), AttentiveFP (Xiong et al., 2019), and modern Hopfield networks (MHNs) (Ramsauer et al., 2021). All models follow the standard splitting protocol provided by MoleculeNet. We report the average AUC over 50 random splits for each dataset.

As shown in Table 7, our method achieves competitive performance across all datasets and sets a new state-of-the-art result on **BBBP**(AUC = 90.2±2.5), **SIDER** (AUC = 62.1±2.3). All hyperparameters were selected on separate validation sets and we selected the model with the highest validation AUC on five different random splits. (see Table 8)

**Discussion**  From Table 7 we see that HAMNs achieve state-of-the-art AUC on **BBBP** and **SIDER**, while being slightly behind MHNs, RF, or SVM on **HIV** and **BACE**. Again, these datasets are essentially flat binary classification problems without explicit hierarchical supervision on the label space or molecular graph, so there is no strong reason to expect a hyperbolic geometry to be uniformly superior to a Euclidean one. In this regime, our associative memory mainly acts as a flexible nonlinear transformation of molecular embeddings, and the small differences (typically within 1–1.5 AUC points) are better interpreted as task-specific idiosyncrasies than as evidence of a systematic weakness of the hyperbolic dynamics. We also note that training remains stable across all four datasets: the CCCP-based energy update monotonically decreases the HAMNs energy and we do not observe divergent behavior or highly variable performance across random seeds. Thus, on standard drug-design benchmarks that do not expose a clear hierarchical structure, HAMNs behave as a competitive associative module on par with Euclidean MHNs, while their advantages are more pronounced on explicitly hierarchical problems such as hierarchical CIFAR-100, WordNet hypernym prediction, and WN18RR link prediction.

### E.4 EXPERIMENT 4: HYPERNYM PREDICTION ON WORDNET

To further verify whether hyperbolic associative memories can exploit *real* hierarchical structure beyond CIFAR-100, we consider a hypernym prediction task on WordNet nouns (Miller, 1995). Following the standard setting, each training sample is a pair $(x, y)$, where $x$ is a target synset and $y$ is one of its (immediate) hypernyms. We restrict the data to noun synsets whose primary hypernym

Table 9: Hypernym prediction on WordNet (nouns). We report test accuracy (%) and the average shortest-path distance in the WordNet hypernym–hyponym graph (lower is better).

| Model | Acc (%) | Avg. hier. dist. |
|---|---|---|
| MHN_Euc | 26.68 | 7.33 |
| U-Hop | 30.09 | 7.33 |
| HypAttn | **52.46** | **4.75** |
| HypNN | 47.54 | 5.25 |
| HAMNs | 52.20 | 4.78 |

Table 10: Hypernym prediction on WordNet (nouns) with an additional ontology embedding front-end (**+OntEuc**). We report test accuracy (%) and the average shortest-path distance in the WordNet hypernym–hyponym graph (lower is better).

| Model | Acc (%) | Avg. hier. dist. |
|---|---|---|
| MHN_Euc + OntEuc | 89.61 | 1.03 |
| U-Hop+OntEuc | 91.06 | 0.89 |
| HypAttn + OntEuc | **96.36** | **0.36** |
| HypNN + OntEuc | 93.63 | 0.63 |
| HAMNs + OntEuc | 96.24 | 0.37 |

appears at least 5 times in the corpus, yielding about 7,700 noun–hypernym pairs and approximately 760 distinct hypernym labels.

Each synset is encoded as a sentence by concatenating its lemmas, gloss, and usage examples in order. We use the `all-MiniLM-L6-v2` Sentence-BERT encoder (Reimers & Gurevych, 2019) to obtain $L_2$-normalized features in $\mathbb{R}^{d_{in}}$, followed by a linear projection to a 512-dimensional representation that is fed into a memory/retrieval block. We compare five architectures: (i) Euclidean modern Hopfield networks (**MHN_Euc**), (ii) a kernelized Euclidean Hopfield baseline (**U-Hop**, (iii) a hyperbolic attention layer (**HypAttn**), (iv) a hyperbolic MLP block (**HypNN**), and (v) our hyperbolic associative memory layer (**HAMNs**). The output of the retrieval module is then passed to a linear classifier $\mathbb{R}^{512} \to \mathbb{R}^{|\mathcal{Y}|}$. We report both the (flat) top-1 accuracy and the average *hierarchical distance* between the predicted and true hypernyms, measured as the shortest-path length on the undirected hypernym–hyponym graph constructed from WordNet (smaller is better).

On this WordNet task—where SBERT already provides a very strong semantic representation and the decision boundary is close to that of a flat classifier over hypernym labels—the additional flexibility of HAMNs does not yield a pronounced performance advantage. Instead, HAMNs performs comparably to HypAttn in terms of accuracy and hierarchical distance, while offering the *same* memory interface that we use in our hierarchical CIFAR-100 experiments and knowledge-graph experiments, where its advantages become more apparent. More concretely, HypAttn remains the best-performing model on this task, whereas HAMNs maintains competitive performance and achieves a similar improvement in hierarchical consistency.

**Adding a ontology embedding**   Recent work on ontology embeddings has shown that even in Euclidean spaces, explicitly encoding the label graph can substantially improve hierarchical prediction performancee (Kulmanov et al., 2021; Smaili et al., 2018). To test whether such techniques would change our conclusions, we add a simple Euclidean *ontology embedding* front-end (denoted **OntEuc**) on top of the SBERT features. Concretely, we train a lightweight two-layer MLP on the WordNet hypernym graph to predict the hypernym label from SBERT features. We then freeze this ontology encoder and use its output as the common input representation for all five decoders (MHN_Euc, U-Hop, HypAttn, HypNN, HAMNs), keeping the label space and supervision identical across variants.

With the ontology encoder in place, all models improve dramatically: even the Euclidean Hopfield baseline rises 26.7% to 89.6% accuracy and its average hierarchical distance drops from 7.33 to 1.03, while U-Hop+OntEuc further improves to 91.06% with distance 0.89. However, hyperbolic

decoders still clearly dominate: both HypAttn and HAMNs reach about $96\%$ accuracy and reduce the average graph distance to $\approx 0.36$–$0.37$, cutting the residual hierarchical error of the Euclidean Hopfield network by roughly a factor of three. These results show that ontology embeddings and hyperbolic retrieval are *complementary*: the ontology encoder provides a strong, graph-aware label prior that benefits all methods, while the hyperbolic memory layer remains better matched to the multi-level structure of the taxonomy under the same input representation and label prior. In principle, any off-the-shelf ontology embedding model could be plugged into our framework as a drop-in front-end to initialize or refine label representations.

Overall, the WordNet experiments lead to three takeaways. First, on this deep hypernym-prediction task, Euclidean Hopfield variants (MHN_Euc and U-Hop) are clearly inadequate as standalone solutions: under the same SBERT encoder and classifier capacity, their accuracies remain far below the hyperbolic decoders and their average graph distances are much larger (around 7.33 vs. 4.75–5.25), indicating that a flat Euclidean retrieval does not exploit the multi-level taxonomy well. Second, both hyperbolic attention (HypAttn) and our HAMNs provide strong and practically usable retrieval layers for such hierarchical label spaces: without any ontology pre-training they already achieve substantially higher accuracy and much smaller hierarchical error than MHN_Euc, and after adding the ontology encoder they remain significantly ahead of the Euclidean Hopfield network. In this setting HypAttn attains the best numbers, while HAMNs stays competitive and offers the same associative-memory interface that we use for CIFAR-100 and WN18RR. Third, the ontology encoder and our hyperbolic memory are complementary rather than alternatives: the former injects a graph-aware label prior that can be shared by any downstream model, whereas the latter determines how well the retrieval layer can respect hierarchical geometry under a fixed input representation and label prior.

### E.5    EXPERIMENT 5: LINK PREDICTION ON WN18RR

**Dataset**    To complement the WordNet hypernym classification in App. E.4, we also evaluate HAMNs on *knowledge–graph completion* using the WN18RR benchmark (Dettmers et al., 2018). WN18RR is a cleaned subset of the WordNet graph that removes inverse and redundant relations so that link prediction really requires modeling the underlying lexical hierarchy rather than memorizing simple shortcuts. Each entity corresponds to a WordNet synset and edges encode semantic relations such as `hypernym`, `hyponym`, `instance_hypernym`, `also_see`, etc. The standard split contains 40,943 entities and 11 relation types, with 86,835 training triples, 3,034 validation triples, and 3,134 test triples. Following the common "reciprocal relations" protocol, we augment the graph with an inverse relation $r^{-1}$ for every $r$, so the effective number of relations becomes 22 in our code.

**Task and evaluation**    A triple $(h, r, t)$ is interpreted as a query $(h, r, ?)$ (and, via the reciprocal trick, also as $(t, r^{-1}, ?)$). During evaluation we rank all entities $e \in \mathcal{E}$ as possible tails and compute *filtered* metrics: for each query, all entities that form any known true triple with $(h, r)$ (except the target $t$ itself) are masked out before ranking. We report Mean Reciprocal Rank (MRR) and filtered Hits@1/3/10, averaged over predicting both head and tail.

**Model variants**    We reuse a single encoder for entities and relations and only change the retrieval / decoder module:

- **MHN_Euc**: a Euclidean modern Hopfield memory that attends over $K = 16$ relation–specific memory slots using dot-product attention, exactly analogous to a standard multi-head attention layer.

- **HypAttn**: a purely geometric hyperbolic attention decoder. Given a query $q = e_h + e_r$ in the tangent space, it maps $q$ and the relation-specific memory slots to the Poincaré ball, computes scores from (scaled) geodesic distances, applies a sharpened temperature-controlled softmax, and returns a Fréchet mean on the manifold.

- **HypNN**: a hyperbolic MLP-style decoder that applies two Möbius-linear layers with non-linearities on the Poincaré ball, followed by a map back to the tangent space at the origin.

- **HAMNs**: our hyperbolic Hopfield decoder implemented via a lightweight `HAMNs_Algo` module. `HAMNs_Algo` applies the CCCP-based intrinsic energy update derived in Sec. 3

Table 11: Link prediction results on WN18RR (filtered, with reciprocal relations). All numbers are from a single run with seed 42.

| Decoder | MRR | Hits@1 | Hits@3 | Hits@10 |
|---------|-----|--------|--------|---------|
| MHN_Euc | 34.31 | 29.13 | 37.03 | 43.54 |
| HypAttn | 38.32 | 31.56 | 42.10 | 50.37 |
| HypNN | 39.33 | 33.09 | 42.34 | 50.62 |
| HAMNs | **39.56** | **33.44** | **42.70** | **50.73** |

with one gradient step and an intrinsic quadratic regularizer on the Poincaré ball, but is reparameterized for the knowledge-graph setting so that memories are relation-specific and the query is the composed vector $e_h + e_r$. Compared to the `Hyperbolic_HopfieldLayer` used in the CIFAR-100 experiments, the update rule is the same while the interface and parameterization are adapted to link prediction on WN18RR.

All variants use $d = 200$-dimensional embeddings for entities and relations, curvature parameter $c = 1.0$, and $K = 16$ memory slots per relation (when applicable).

**Training details**  We train with negative sampling: for each positive triple $(h, r, t)$ we draw $K_{\text{neg}} = 50$ random negative tails, compute scores for one positive and $K_{\text{neg}}$ negatives, and optimize a binary cross-entropy objective. We use AdamW with learning rate $2 \times 10^{-3}$, weight decay $10^{-4}$ and batch size 1024 for 100 epochs (we reduce the learning rate by a factor of 2 for HAMNs to avoid overly aggressive energy updates). All runs use the same random seed and mixed-precision training (AMP) for efficiency.

**Results**  Table 11 summarizes the filtered test performance of all decoders on WN18RR. The Euclidean Hopfield baseline (MHN_Euc) is clearly dominated by all three hyperbolic decoders, improving MRR by about $+0.04$ and Hits@10 by more than $+7$ absolute points. Among the hyperbolic variants, HAMNs achieves the best overall performance, obtaining the highest MRR (0.3956) and Hits@1/3/10 (0.3344/0.4270/0.5073) and slightly outperforming both HypNN and HypAttn on all metrics. The improvements over HypNN are modest but consistent, indicating that the proposed energy-based retrieval can serve as a competitive hyperbolic decoder for hierarchical knowledge-graph completion, rather than being only of theoretical interest.

Overall, this experiment shows that (i) hyperbolic decoders are consistently beneficial for link prediction on a knowledge graph with inherent lexical hierarchy, (ii) the Euclidean Hopfield baseline MHN_Euc, although strong on standard Euclidean benchmarks, systematically underperforms all hyperbolic variants on WN18RR, and (iii) HAMNs achieves slightly better performance than the other hyperbolic decoders, indicating that the proposed energy-based retrieval is not only theoretically well-founded but also practically effective as a drop-in decoder for hierarchical knowledge-graph completion.

E.6    EXPERIMENT 6: RUNTIME AND MEMORY ANALYSIS OF DIFFERENT DECODERS

To better understand the practical cost of different decoders, we profile four variants—Euclidean modern Hopfield networks (**MHN_Euc**), hyperbolic attention (**HypAttn**), a hyperbolic MLP decoder (**HypNN**), and our hyperbolic associative memory (**HAMNs**)—under two representative settings. All measurements are obtained with PyTorch's profiler on a single NVIDIA RTX 4090 GPU; numbers are indicative rather than hardware-independent.

**CIFAR-100 hierarchical classification**  In the CIFAR-100 experiments (Sec. 4.1) HAMNs are instantiated via the generic `Hyperbolic_HopfieldLayer` module, which closely mirrors the Euclidean `HopfieldLayer`: it keeps the same multi-head Hopfield core (query/key/value projections, hidden dimension, number of heads, etc.) but replaces the dot-product similarity with a hyperbolic energy and performs one CCCP step on the Poincaré ball. As a result, HAMNs and MHN_Euc have comparable parameter counts and FLOPs, while the hyperbolic operations (Möbius

Table 12: Decoder cost on CIFAR-100 hierarchical classification (batch size 128, input $3 \times 224 \times 224$). FLOPs are for a single forward pass.

| Decoder | FLOPs (G) | Params (M) | Fwd (ms) | Fwd+Bwd (ms) |
|---|---|---|---|---|
| HAMNs | $5.19 \times 10^2$ | 14.5 | 41.8 | 144.1 |
| HypAttn | $4.65 \times 10^2$ | 12.1 | 17.0 | 59.7 |
| HypNN | $4.64 \times 10^2$ | 11.8 | 15.9 | 53.9 |
| MHN_Euc | $6.81 \times 10^2$ | 19.8 | 23.3 | 80.8 |

Peak GPU memory (GB): HAMNs $\approx 6.4$, HypAttn $\approx 3.2$, HypNN $\approx 2.8$, MHN_Euc $\approx 4.2$.

Table 13: Decoder cost on WN18RR link prediction (batch size 1024, $K_{\text{neg}} = 50$ negatives per positive). FLOPs are reported per batch forward pass.

| Decoder | FLOPs (G) | Params (M) | Fwd (ms) | Fwd+Bwd (ms) |
|---|---|---|---|---|
| MHN_Euc | $4.15 \times 10^2$ | 8.38 | 8.4 | 24.9 |
| HypAttn | $4.23 \times 10^2$ | 8.38 | 51.5 | 228.9 |
| HypNN | $3.24 \times 10^1$ | 8.30 | 3.7 | 13.3 |
| HAMNs | $2.20 \times 10^2$ | 8.34 | 40.6 | 169.8 |

Peak GPU memory (GB): MHN_Euc $\approx 2.7$, HypAttn $\approx 13.1$, HypNN $\approx 0.83$, HAMNs $\approx 9.3$.

addition, exponential/logarithmic maps, Riemannian preconditioning) introduce additional runtime and memory overhead.

Table 12 reports FLOPs, parameter counts, wall-clock time and peak GPU memory for the four decoders when plugged into our ResNet-18 backbone on CIFAR-100 with an input of $128 \times 3 \times 224 \times 224$.

We observe that `Hyperbolic_HopfieldLayer` indeed uses fewer FLOPs and parameters than MHN_Euc, but the additional hyperbolic operations and intermediate states make its wall-clock time and peak memory *larger*. This gap is largely an implementation artifact: the current hyperbolic layer is written as a generic drop-in replacement for the Euclidean Hopfield core and has not yet been optimized with custom GPU kernels or fused operations.

**WN18RR link prediction** In the WN18RR link-prediction experiments (App. E.5) the decoder interface is different: for each query $(h, r, ?)$ we attend over $K$ relation-specific memory slots and $K_{\text{neg}}$ negative tails. Here we use a lighter HAMNs implementation, denoted `HAMNs_Algo`, instead of the generic `Hyperbolic_HopfieldLayer`. `HAMNs_Algo` keeps a single linear projection for queries, an `Embedding` table for relation-specific memories, and performs a single CCCP update in the Poincaré ball; there is no full Hopfield core with separate key/value projections. All decoders share the same entity/relation encoder and training protocol (batch size 1024, $K_{\text{neg}} = 50$ negatives).

Table 13 summarizes the runtime and memory statistics on WN18RR.

Two trends are worth highlighting. First, the hyperbolic decoders are more expensive than the Euclidean Hopfield baseline, as expected: both HypAttn and HAMNs must map queries and memories back and forth between the tangent space and the Poincaré ball and compute geodesic distances for each candidate tail and each relation-specific memory slot. Among them, HypAttn is the heaviest: it performs a full Fréchet-mean style hyperbolic attention for every $(h, r, ?)$ query and all negatives, which explains its highest FLOPs, longest runtime (forward $\approx 51$ ms, forward+backward $\approx 229$ ms), and largest memory footprint ($\approx 13$ GB). Second, the custom `HAMNs_Algo` lies between HypNN and HypAttn: it is substantially lighter than HypAttn (around $1.3\times$ faster and $\approx 30\%$ less peak memory) but still slower and more memory-hungry than the Euclidean MHN_Euc and the very compact HypNN decoder.

Overall, these measurements emphasize that (i) our hyperbolic associative memory can be implemented either as a generic drop-in replacement for the Euclidean Hopfield core (`Hyperbolic_HopfieldLayer`, used in CIFAR-100) or as a task-specific lightweight module (`HAMNs_Algo`, used in WN18RR), and (ii) in both cases the extra geometric operations—rather

than a fundamentally higher algorithmic complexity—are the main source of runtime and memory overhead. We expect more optimized kernel implementations to substantially narrow this gap without changing the underlying model.

### E.7 EXPERIMENT 7: ABLATIONS: CURVATURE AND NUMBER OF STORED PATTERNS

We finally study how the performance of HAMNs depends on the curvature parameter $c$ and the number of stored patterns (memory capacity) on the hierarchical CIFAR-100 benchmark with four levels (top / super / coarse / fine).

Table 14: Comparison of curvature $c$ (higher is better).

| $c$ | flat_top | flat_super | flat_coarse | flat_fine |
|---|---|---|---|---|
| 0.1 | 0.8834 | 0.7501 | 0.5891 | 0.3524 |
| 0.2 | 0.8784 | 0.7366 | 0.5614 | 0.2798 |
| 0.3 | 0.8656 | 0.7225 | 0.5888 | 0.3498 |
| 0.4 | 0.8942 | 0.7629 | 0.6342 | 0.3757 |
| 0.5 | 0.8897 | 0.7439 | 0.6142 | 0.3617 |
| 0.6 | 0.8755 | 0.7687 | 0.6493 | 0.4372 |
| 0.7 | 0.9030 | 0.7774 | 0.6695 | 0.4823 |
| 0.8 | 0.8841 | 0.7571 | 0.6204 | 0.4505 |
| 0.9 | 0.8898 | 0.7675 | 0.6514 | 0.4563 |
| 1.0 | 0.8818 | 0.7592 | 0.6522 | 0.4715 |
| 2.0 | 0.8919 | 0.7570 | 0.6455 | 0.4737 |
| 3.0 | 0.8902 | 0.7624 | 0.6461 | 0.4467 |
| 4.0 | 0.8924 | 0.7711 | 0.6459 | 0.4112 |
| 5.0 | 0.8577 | 0.7541 | 0.6395 | 0.4099 |
| 6.0 | 0.8807 | 0.7429 | 0.5745 | 0.2680 |
| 7.0 | 0.8860 | 0.7521 | 0.6262 | 0.3536 |
| 8.0 | 0.8704 | 0.7446 | 0.6035 | 0.3288 |
| 9.0 | 0.8715 | 0.7280 | 0.5661 | 0.2162 |
| 10.0 | 0.8617 | 0.6864 | 0.4368 | 0.1468 |

*The curvature $c$ comparison data above come from hierarchical classification results on CIFAR-100 after imposing a four-level structured hierarchy.*

Table 15: Comparison of number of stored patterns on CIFAR-100 (4-level hierarchy).

| stored_n | flat_top | flat_super | flat_coarse | flat_fine |
|---|---|---|---|---|
| 100 | 0.8898 | 0.7517 | 0.6419 | 0.4498 |
| 150 | 0.8893 | 0.7680 | 0.6563 | 0.4670 |
| 200 | 0.8826 | 0.7394 | 0.6413 | 0.4754 |
| 250 | 0.8710 | 0.7466 | 0.6494 | 0.4642 |
| 300 | 0.8917 | 0.7563 | 0.6463 | 0.4596 |
| 350 | 0.8889 | 0.7641 | 0.6431 | 0.4505 |
| 400 | 0.8954 | 0.7630 | 0.6379 | 0.4558 |
| 450 | 0.8945 | 0.7649 | 0.6282 | 0.4290 |
| 500 | 0.8851 | 0.7586 | 0.6458 | 0.4728 |
| 550 | 0.8928 | 0.7674 | 0.6400 | 0.4647 |
| 600 | 0.8932 | 0.7491 | 0.6426 | 0.4740 |
| 650 | 0.8916 | 0.7796 | 0.6507 | 0.4643 |
| 700 | 0.8893 | 0.7676 | 0.6513 | 0.4749 |
| 750 | 0.8932 | 0.7473 | 0.6419 | 0.4547 |
| 800 | 0.8836 | 0.7778 | 0.6575 | 0.4780 |

**Effect of curvature** Table 14 reports flat accuracy at each hierarchy level for different values of the Poincaré curvature $c$. We observe a clear "sweet spot": choosing a *moderate* negative curvature

(roughly $c \in [0.7, 2.0]$) yields the best trade-off across all levels, while very small or very large curvatures lead to degraded performance. Intuitively, when $c$ is too small the geometry becomes almost Euclidean and cannot efficiently represent tree-like structure, so deeper classes cannot be pushed significantly farther away in radial distance. Conversely, when $c$ is too large, the space becomes extremely contracted near the boundary; small changes in radius then correspond to very large changes in geodesic distance, making the energy landscape overly sharp and harder to optimize. The best-performing values of $c$ are precisely those that allow the hyperbolic radius to encode hierarchical depth in a smooth but nontrivial way.

**Effect of stored-pattern count** Table 15 varies the number of stored patterns from 100 up to 800 on the same four-level hierarchy. Since CIFAR-100 has 100 classes, the lower end of this range corresponds to roughly one attractor per class, while the upper end corresponds to multiple attractors per class or per subtree. We see that the accuracies at each level are not strictly monotonic: fine-level accuracy tends to benefit from higher capacity (up to around 800 patterns), whereas the top-level accuracy can slightly decrease when too many patterns are added.

This behaviour is consistent with the role of stored patterns in the hyperbolic energy. With too few patterns, a single attractor must explain several semantically distinct classes or subtrees, so the retrieved states become blurred and fine-level decisions suffer from under-capacity. As we increase the number of stored patterns, the model can allocate more specialised attractors to different branches of the hierarchy, which helps coarse and fine levels. However, when the capacity becomes very large, the energy landscape starts to fragment into many local minima that lie within the same top-level branch. From the perspective of a top-level classifier that aggregates over all these attractors, this over-fragmentation can slightly hurt robustness: some queries are pulled towards spurious but semantically redundant attractors inside the wrong branch, which explains the mild drop in top-level accuracy for the largest capacities. Overall, the fluctuations remain small (on the order of one percentage point), indicating that HAMNs are quite robust to the precise choice of memory size as long as it lies in a reasonable range.

### E.8 RELATION TO HYPERBOLIC ATTENTION DECODERS

**Mechanism: one-shot attention vs. energy-based retrieval** HypAttn is a *single-step geometric attention layer* on the Poincaré ball. Given a query $q$ in the tangent space and a set of memory slots $\{m_k\}$, it maps them to hyperbolic space, computes scores from (scaled) geodesic distances $d_{\mathbb{H}}(q, m_k)$, applies a temperature-controlled softmax, and returns a Fréchet mean on the manifold. Thus the output is a weighted average of memory points in hyperbolic space, with weights depending on the current query but without an explicit global energy function or retrieval dynamics.

By contrast, HAMNs define an *explicit Hopfield-style energy* on a hyperbolic manifold (instantiated via the Minkowski inner product, see App. A.1) and retrieve memories by approximately minimizing this energy via a CCCP update on the manifold. In simplified form, the energy takes the structure

$$E(z, \Xi) = -\sum_{m \in \Xi} \alpha_m \langle z, m \rangle_L + \lambda \, \Phi(z),$$

where $\Xi$ is the memory set, $\langle \cdot, \cdot \rangle_L$ is the Lorentzian inner product (monotonically related to hyperbolic distance), and $\Phi$ is an intrinsic regularizer. The CCCP-based update performs (intrinsic) gradient steps that provably decrease $E$ until reaching a critical point. This yields genuine *associative retrieval dynamics* with well-defined attractors in hyperbolic space, rather than a one-shot averaging operation.

Importantly, the equivalence between modern Hopfield networks and attention in the Euclidean case arises because the Hopfield update can be written *exactly* as a softmax attention rule (Ramsauer et al., 2021). Hyperbolic attention layers (Gülçehre et al., 2019), in contrast, are defined by directly replacing dot products with functions of geodesic distance and computing a single Fréchet mean per query; there is no known global Hopfield energy on a negatively curved manifold whose gradient flow reproduces this one-shot update. Deriving HAMNs therefore requires constructing a bona fide hyperbolic Hopfield energy and its CCCP retrieval dynamics, rather than mechanically "lifting" the Euclidean MHN=attention correspondence to a curved space. In other words, the Euclidean MHN–attention equivalence does not carry over to hyperbolic manifolds in a trivial reverse-engineering way.

In summary, HypAttn answers "where to attend on the manifold in one step", whereas HAMNs answer "which memory configuration minimizes a global hyperbolic energy, reached via a stable iterative update". HypAttn is closer to a hyperbolic analogue of multi-head attention, while HAMNs are hyperbolic modern Hopfield networks.

**Memory geometry and hierarchy**   The explicit energy in HAMNs lets us reason about and control the geometry of attractors. On CIFAR-100, we show that the learned hyperbolic memories organize themselves so that radial coordinate correlates with tree depth and geodesic distance correlates with branch similarity (App. E.1.2). This behaviour follows naturally from the Minkowski-based association term and the radius regularizer in the HAMNs energy. HypAttn, in contrast, has no explicit notion of an energy landscape or attractors; it produces a Fréchet mean conditioned on the current query but does not define a shared set of stable memory states whose geometry we can inspect in the same way.

**Empirical comparison**   Empirically, HypAttn and HAMNs show complementary strengths across tasks:

- **Hierarchical CIFAR-100.**   On shallow hierarchies (2-layer), HypAttn attains the best coarse/fine accuracies, reflecting its strength as a local geometric aggregator. As depth increases to 4 layers, HAMNs achieve the best top/super/coarse accuracies and the highest cross-level coherence (`coph_corr`), while HypAttn remains strongest at fine-grained recognition (Sec. 4.1, App. E.1). This matches the intuition that an energy-based memory with explicit attractors is particularly helpful for aligning predictions across multiple levels of a deep tree.

- **WordNet hypernym prediction.** On the SBERT-only setting, both HypAttn and HAMNs substantially outperform the Euclidean Hopfield baseline (MHN_Euc), with HypAttn slightly ahead and HAMNs essentially tied in terms of accuracy and hierarchical distance (App. E.4). After adding the same ontology encoder (**OntEuc**) in front of *all* decoders, every model improves, but HypAttn+OntEuc and HAMNs+OntEuc remain clearly better than MHN_Euc+OntEuc. This indicates that, under a strong graph-aware label prior, both hyperbolic attention and hyperbolic associative memories can exploit the WordNet hierarchy effectively.

- **WN18RR link prediction.** On WN18RR, a strongly hierarchical knowledge graph, all three hyperbolic decoders (HypAttn, HypNN, HAMNs) significantly outperform Euclidean MHN_Euc on MRR and Hits@1/3/10. HAMNs achieves the best overall scores among the hyperbolic decoders (App. E.5), suggesting that the CCCP-based energy minimization provides a mild but consistent advantage when the retrieval problem is genuinely multi-hop and hierarchical.

**Conceptual takeaway**   Hyperbolic attention and HAMNs share the same underlying manifold but embody different design philosophies. HypAttn is a *direct hyperbolic attention mechanism*: it leverages geodesic distances to compute attention weights and produce a single Fréchet mean per query. HAMNs are *energy-based hyperbolic memories*: they define a global Hopfield energy on the manifold and use a provably convergent CCCP update to retrieve attractors. Our experiments suggest that when the data or label space exhibits deep hierarchical structure (CIFAR-100 4-layer tree, WordNet taxonomy, WN18RR), this energy-based retrieval provides more stable and hierarchy-consistent behavior than a single-step hyperbolic attention layer.

## THE USE OF LARGE LANGUAGE MODELS (LLMS)

We used large language models (LLMs) as a general-purpose assistant only for: (i) translation and grammar correction; (ii) text polishing and wording refinement; and (iii) suggesting intermediate steps or equivalent formulations in a small subset of mathematical derivations. Specifically, the LLM provided text-level assistance when drafting or rewriting the following parts: the model-agnostic gradient form of the hyperbolic similarity (Appendix A.5) and the structured presentation of upper/lower bounds for the energy function (Appendix A.3). All assumptions, derivations, and final proofs were independently re-derived, verified, and corrected by the authors as needed.

