# OpenReview forum: "Hyperbolic Associative Memory Networks"
_ICLR.cc/2026/Conference — ICLR 2026 Conference Desk Rejected Submission_

### Official Review · Reviewer_kjQH · 2025-10-26

**Soundness:** 3
**Presentation:** 3
**Contribution:** 2
**Rating:** 6
**Confidence:** 3

**Summary:**

This paper introduces Hyperbolic Associative Memory Networks (HAMNs), a novel extension of modern Hopfield networks (MHNs) to hyperbolic space. The core motivation is that Euclidean-based MHNs struggle to capture hierarchical data structures, leading to distortion, while hyperbolic geometry is naturally suited for such data.

The authors propose a principled formulation that:

Maps query and memory vectors from Euclidean to hyperbolic space using exponential maps.

Defines an LSE-based energy function using a hyperbolic similarity metric (based on $-cosh(d_{\mathcal{M}})$).

Employs a Riemannian optimization (CCCP) to perform the memory retrieval process on the manifold.

The central claim is a "hierarchy-sensitivity hypothesis": HAMNs should significantly outperform Euclidean MHNs on data with deep hierarchical structure, while performing on par with them on flat or shallow-hierarchy data.

The authors test this by creating artificial 2, 3, and 4-level hierarchies for CIFAR-100 and comparing performance on level-specific accuracy and a cross-level consistency metric. They also test on MIL and MoleculeNet datasets as "weak hierarchy" benchmarks.

While the theoretical formulation is sound and the research direction is well-motivated, the empirical evidence is not strong enough to fully support the claims. The results are mixed, the computational overhead is significant, and the primary experiment relies on an artificial data hierarchy, which fails to provide a conclusive test of the model's ability to capture innate hierarchical data structure.

**Strengths:**

1. **Clear conceptual idea and principled derivation.**
   The paper provides a clean extension of modern Hopfield energy-based retrieval to hyperbolic geometry, with a principled energy based on geodesic distance / −cosh(d) (Minkowski-like similarity) and a CCCP-based Riemannian update rule. The math for gradients, CCCP surrogate and closed-form update is sound and carefully explained.

2. **Model-agnostic design & practical modules.**
   The framework is model-agnostic (Poincaré, Lorentz, etc.) and the authors provide concrete module designs (HypHopfield / HypPooling / HypLayer) that can be dropped into networks—valuable for broad adoption. Implementation hints and pseudocode are provided.

3. **Thoughtful theory + capacity discussion.**
   The paper analyzes energy-well separation and sphere-packing style capacity bounds in hyperbolic space, which grounds the method theoretically for hierarchical data.

4. **Relevant experiments and ablations.**
   The authors run a spectrum of experiments: controlled CIFAR-100 hierarchical label trees (2/3/4 levels), weak-hierarchy tasks (MIL, MoleculeNet), and ablations on curvature and stored-pattern count. This shows attention to the core hypothesis.

**Weaknesses:**

1. **Empirical evidence for “captures hierarchical structure” is limited / not fully convincing.**
   The core hierarchy-sensitivity hypothesis is only evaluated by (i) synthetic label hierarchies on CIFAR-100 and (ii) coarse–fine consistency metrics (cophenetic correlation). While sensible, these do **not** directly demonstrate that learned embeddings or retrieval dynamics actually encode hierarchy (e.g., radial depth separation or LCA distances). The claim that HAMNs exploit hierarchical geometry needs stronger direct evidence.

2. **Improvements are often modest and inconsistent across metrics / levels.**
   Table 1 shows HAMNs sometimes yield only small gains (or are tied) versus baselines; e.g., on the 3-layer setup the gains are marginal and on 2-layer results are mixed. The paper asserts “statistically significant improvements as depth increases,” but effect sizes and significance tests are not fully laid out. Gains appear incremental rather than transformative.

3. **Tests rely heavily on one synthetic-controlled dataset (CIFAR-100 label restructurings).**
   Restructuring labels is useful but it does not guarantee that **input representations** or the memory module truly organize patterns in a hierarchical metric manner. The other evaluations (MIL, MoleculeNet) belong to the “weak / shallow” regime and show only small margins. The test-suite lacks **datasets with known hierarchical latent structure** (trees, taxonomy graphs, synthetic trees with noise, ontologies).

4. **Runtime & memory trade-offs are underplayed.**
   While HAMNs have theoretically fewer FLOPs/parameters, actual runtime and GPU-memory are worse in the reported implementation (Table 2). This is an important practical drawback that should be discussed in more depth.

**Questions:**

1. **Direct hierarchy evidence:**
   Can you show that embeddings (memories + queries) arrange by tree depth and branch—e.g., radial coordinate correlates with level, pairwise geodesic distance correlates with LCA depth?

2. **Choice of CIFAR label restructurings:**
   How sensitive are results to different hierarchical clusterings? Would the observed gains persist under alternative splits?

3. **Temperature and curvature interactions:**
   How was θ chosen relative to curvature c? Is there a principled relation ensuring CCCP concavity conditions hold?

4. **Effect of stored-pattern count:**
   Why does top-level accuracy sometimes degrade with more stored patterns? Does this interact with curvature or temperature?

---

> ### Author Response · Authors · 2025-11-20
> **Response to Reviewer kjQH (Part I)**
>
> We thank the reviewer for the careful and insightful comments. Below we address the concerns on hierarchical evidence, CIFAR-100 label restructurings, runtime/memory trade-offs, the relation between temperature and curvature, and the effect of stored-pattern count, and summarize changes in the revised version.
>
> # 1. Direct evidence for “captures hierarchical structure”
>
> We now provide both **theoretical** and **empirical** support.
>
> **(a) Theory: Minkowski inner product vs. hyperbolic distance (App. A.1).**
> In the Lorentz model we recall that
> $ d_{\mathbb H}(u,v) = \mathrm{arcosh}(-\langle u,v\rangle_L)$,
> so $-\langle u,v\rangle_L$ is a strictly monotone, curvature-aware dissimilarity equivalent in order to $d_{\mathbb H}(u,v)$.
> Writing $u=(u_0,\bar u)$ and $v=(v_0,\bar v)$ with radii $r_u,r_v$ and angle $\theta$ gives
> $ \langle u,v\rangle_L = -\cosh r_u\cosh r_v + \sinh r_u\sinh r_v\cos\theta$.
> Thus, for fixed $\theta$, increasing $r_u,r_v$ makes $\langle u,v\rangle_L$ more negative; the surrogate dissimilarity $-\langle u,v\rangle_L$ is therefore much more sensitive to **radial separation** than to purely angular differences. This explains our design choice of using the negative Minkowski inner product as the association term in the HAMN energy and naturally encourages encoding depth in the radius and branch similarity in the angle.
>
> **(b) Empirics: depth–radius and LCA–distance correlations (App. E.1.1).**
> On the 4-level CIFAR-100 hierarchy we probe the geometry of the learned memories:
>
> - For each class memory we record its tree depth (top/super/coarse/fine) and its hyperbolic radius (geodesic distance to the origin). The mean radius increases monotonically from top to fine (≈2.57 → 5.07), with Spearman correlation $\rho=0.57$ and $p\approx1.6\times10^{-12}$.
> - For each pair of fine-level memories we record their lowest common ancestor (LCA) depth and their hyperbolic distance. Pairs that only share the root (LCA depth 0) are farthest apart, while pairs with deeper LCAs are closer. The Spearman correlation between LCA depth and distance is negative (≈−0.20), equivalently tree distance and hyperbolic distance are positively correlated.
>
> These results show that, without any explicit geometric constraint, optimizing the Minkowski-based energy organizes memories so that **radius correlates with depth and geodesic distance correlates with branch similarity**, rather than merely fitting flat labels.
>
> # 2. CIFAR-100 artificial hierarchy, gain magnitude, and robustness
>
> **(a) Sensitivity to CIFAR-100 label restructurings (App. E.1.1).**
> Because the hierarchy used in the main text is manually designed, we add a **purely feature-driven** 4-level hierarchy:
>
> - Extract ResNet-18 features for all training images.
> - Cluster 100 fine classes into 20 coarse clusters (each coarse has ≥4 fine).
> - Cluster 20 coarse centroids into 7 super nodes (each super has ≥2 coarse).
> - Cluster 7 super centroids into 3 top nodes (each top has ≥2 super).
>
> We keep backbone and training hyperparameters fixed and retrain MHN\_Euc, HypAttn, HypNN, and HAMNs on this new tree (10 seeds). Performance is very similar to the semantic hierarchy, and the **relative ranking of the four models remains stable**: HAMNs retains the strongest performance on the higher levels and the highest cophenetic correlation, while HypAttn/HypNN are strongest at the fine level. This indicates that our conclusions are **not over-tuned** to a particular CIFAR partition.
>
> **(b) Real hierarchical data beyond CIFAR: WordNet hypernyms and WN18RR (Apps. E.4 & E.5).**
> To reduce our reliance on synthetic hierarchies and to demonstrate the practicality of our method on other domains with deep hierarchical structure, we additionally include two real-world hierarchical benchmarks:
>
> 1. **WordNet noun hypernym prediction (App. E.4).**
>    Given a noun synset, the task is to predict its immediate hypernym. On SBERT features, we compare MHN\_Euc, HypAttn, HypNN, and HAMNs, reporting top-1 accuracy and average shortest-path distance in the WordNet hypernym–hyponym graph.
>    - Without an ontology encoder, MHN\_Euc attains less than half the accuracy of the hyperbolic decoders and much larger hierarchical error, showing that a flat Euclidean energy fails to exploit the taxonomy.
>    - HypAttn and HAMNs are both much better; HAMNs is competitive with HypAttn in accuracy and hierarchical distance.
>    - With an additional ontology encoder (+OntEuc) placed **in front of all decoders**, all methods improve dramatically, but hyperbolic decoders still reduce the residual hierarchical error by roughly a factor of three compared to MHN\_Euc.

---

> ### Author Response · Authors · 2025-11-20
> **Response to Reviewer kjQH (Part II)**
>
> 2. **WN18RR link prediction (App. E.5).**
>    On WN18RR we reuse the same entity/relation encoder and swap only the decoder: MHN\_Euc, HypAttn, HypNN, and our lightweight HAMNs\_Algo.
>    All three hyperbolic decoders significantly outperform MHN\_Euc on MRR and Hits@1/3/10. Among them, HAMNs achieves the best overall scores, slightly but consistently above HypAttn and HypNN.
>
> These additional experiments show that our “hierarchy-sensitivity” hypothesis holds not only on CIFAR-100 label restructurings but also on **real lexical hierarchies and knowledge graphs**.
>
> **(c) On “improvements are often modest and inconsistent”.**
> We agree that, especially on shallow hierarchies or weakly hierarchical tasks, the gains over MHN\_Euc are incremental rather than dramatic. In the revision we soften our claims and clarify our intent:
>
> - **Compared to Euclidean MHN\_Euc**, HAMNs achieve **clear and statistically significant improvements** on deeply hierarchical tasks (4-level CIFAR-100, WordNet, WN18RR), particularly on higher-level accuracy and hierarchy-aware metrics.
> - **Compared to other hyperbolic decoders** (HypAttn, HypNN), HAMNs are **competitive**: in some settings (e.g., WN18RR) they are slightly better; in others (e.g., WordNet hypernyms) they are on par with the best hyperbolic baseline.
> - On shallow or essentially flat tasks (MIL, MoleculeNet), HAMNs usually match MHN\_Euc and behave as a **reliable drop-in associative memory module**, but we do not claim systematic superiority there.
>
> # 3. Runtime & memory trade-offs
>
> Responding to the concern that runtime/memory trade-offs are underplayed, App. E.6 now profiles MHN\_Euc, HypAttn, HypNN, and HAMNs in two settings:
>
> - **CIFAR-100 hierarchical classification**, using the generic `Hyperbolic_HopfieldLayer`.
>   HAMNs and MHN\_Euc have comparable parameter counts and even fewer FLOPs for HAMNs, but HAMNs incur higher wall-clock time and peak memory due to Möbius operations, exp/log maps and Riemannian preconditioning.
> - **WN18RR link prediction**, using the task-specific `HAMNs_Algo`.
>   Here `HAMNs_Algo` lies between HypNN and HypAttn: clearly lighter than HypAttn but still slower and more memory-hungry than MHN\_Euc and the very compact HypNN.
>
> We explicitly state in the **Limitations** section that our main contribution is in **representational power on hierarchical data**, not in computational efficiency. The overhead mainly comes from currently unoptimized hyperbolic kernels rather than fundamentally worse algorithmic complexity.

---

> ### Author Response · Authors · 2025-11-20
> **Response to Reviewer kjQH (Part III)**
>
> # 4. Temperature $\theta$, curvature $c$, and CCCP concavity
>
> Apps. A–C clarify how $\theta$ and $c$ interact with the energy and CCCP updates.
>
> 1. **Boundedness and stability (App. “Bounding the Energy Function”).**
>    For
>    $ E(\xi) = -\tfrac1\theta\log\sum_{i=1}^S e^{\theta\langle x_i,\xi\rangle_M} + \tfrac12 d_{\mathcal M}(\xi,p)^2$,
>    we derive explicit bounds
>    $ 1 - \tfrac{\log S}{\theta} \le E(\xi) \le \cosh(M_r+R_r) + \tfrac12 R_r^2$,
>    where $M_r,R_r$ are maximal radii of memories and states. These constants depend only on $M_r,R_r,\theta$, but **not** on the particular hyperbolic model, and ensure that CCCP or Riemannian gradient iterations operate on a bounded energy.
>
> 2. **Geodesic convex–concave decomposition and a sufficient range for $\theta$ (App. “Optimization via CCCP”).**
>    On a Hadamard manifold, $d_{\mathcal M}$ is geodesically convex and $\cosh\circ d_{\mathcal M}$ is convex, so $s_i(\xi)=-\cosh(d_{\mathcal M}(x_i,\xi))$ is geodesically concave. Writing
>    $E_{\mathrm{cave}}(\xi)=-\tfrac1\theta\log\sum_i e^{\theta s_i(\xi)}$, we show that if
>    $-\mathrm{Hess}\,s_i\succeq\kappa I$ and the gradients are bounded by $L$, then whenever
>    $0<\theta\le\kappa/L^2$, $E_{\mathrm{cave}}$ is geodesically concave and
>    $E=E_{\mathrm{cvx}}+E_{\mathrm{cave}}$ fits the standard DC/CCCP “convex + concave” setting on a Hadamard manifold. Under these conditions, CCCP guarantees monotone energy decrease and convergence to stationary accumulation points (we do not claim global optimality).
>
> 3. **Practical choice of $\theta$ relative to $c$.**
>    In practice we first perform a small grid search over $\theta$ (e.g., 0.8–3.0) at a moderate curvature (e.g., $c\approx0.8$), choosing values that yield stable training and smooth energy curves. In the curvature ablations we then **fix $\theta$** to isolate the effect of $c$. Our empirical settings fall well within the conservative theoretical range above, and we consistently observe monotone energy decrease during CCCP updates.
>
> # 5. Effect of stored-pattern count
>
> App. E.7 studies how performance depends on curvature $c$ and the number of stored patterns on the 4-level CIFAR-100 hierarchy.
>
> - **Curvature**: a moderate negative curvature ($c\in[0.7,2.0]$) yields the best trade-off; very small $c$ makes the geometry too Euclidean, while very large $c$ makes the space overly contracted near the boundary and the energy landscape too sharp.
> - **Stored-pattern count**: we vary `stored_n` from 100 to 800. Fine-level accuracy generally benefits from higher capacity, while top-level accuracy can drop slightly at the largest capacities. Overall fluctuations are small (≈1 percentage point).
>
> We interpret this as follows:
> with too few patterns, one attractor must cover several semantically distinct subtrees, so fine-level decisions suffer from under-capacity; as capacity increases, the model can allocate more specialized attractors along the hierarchy, improving coarse/fine levels; with very large capacity, the energy landscape fragments into many local minima within each top-level branch, which can slightly hurt robustness at the top level if some queries are attracted to redundant minima in the wrong branch. This behavior is consistent with our geometric capacity analysis.
>
> Finally, following suggestions from all reviewers, the revision also adds:
> - App. A.2: an illustrative hyperbolic energy landscape;
> - Apps. B.1–B.2: proofs for energy-well separation and a one-step attraction basin;
> - Apps. E.2–E.3: expanded discussion of the “flat” tasks (MIL, MoleculeNet) where HAMNs perform similarly to Euclidean MHNs.
>
> We hope these additions clarify how HAMNs capture hierarchical structure, how robust the results are across hierarchies and datasets, and how they trade off representational benefits against computational cost.

---

### Official Review · Reviewer_8WQQ · 2025-10-29

**Soundness:** 2
**Presentation:** 2
**Contribution:** 2
**Rating:** 2
**Confidence:** 5

**Summary:**

This paper addresses the limitation of Euclidean Hopfield networks on hierarchical data by extending associative memory retrieval to hyperbolic geometry.
The authors propose a Hyperbolic Associative Memory Network (HAMN) that maps inputs to a hyperbolic manifold and defines a new energy function using the Minkowski inner product in hyperbolic space.
A Riemannian optimization (concave–convex procedure) is employed for memory retrieval, ensuring updates stay on the manifold.
Experiments on hierarchical classification (CIFAR-100 with label trees), multi-instance learning, and molecular property prediction confirm the hypothesis: HAMN yields significant gains on deeply hierarchical tasks while matching Euclidean baselines on shallow structures.

**Strengths:**

1. modeling hierarchical data/task is very important (and not easy) given their combinatorial nature.

2. I skeem through the math. seems ok but not sure about correctness. (didn't check them line-by-line. might be more willing to do that after other issues are addressed.)

3. HAMNs are presented as a model-agnostic module that can “plug into” various architectures. This is very good for applicability.

**Weaknesses:**

Replacing the Euclidean inner product in MHNs is not new. The paper’s novelty narrows to a Riemannian instantiation in hyperbolic space with curvature‑aware optimization and empirical focus on hierarchical data.

1. this paper omits many key developments on modern Hopfield networks that are directly relevant to its main claims..
    - Especially, **replacing the euclidean dot product is not new.** https://arxiv.org/abs/2404.03827 has kernelized the inner product as a learnable operator conditioned on the memory set with rich geometric intuition. https://arxiv.org/abs/2410.23126 showed the optimality of such method. Yet neither work is mentioned at all.
    - Moreover, **there are fundamental design principles for MHNs reported in literature already**, yet not mentioned at all.
    - Krotov & Hopfield https://arxiv.org/abs/2008.06996 gives a dynamical system derivation of a series of energy function and the update rule of MHNs through Legendre transformation in the classical mechanics sense.
    - Han Liu's group https://arxiv.org/abs/2309.12673 and https://openreview.net/forum?id=6iwg437CZs gives a unified framework for constructing the energy functions of modern Hopfield networks via entropy regularizer. This "entropy regularizer" framework extends and covers many MHN-attention correspondence (cf [Ramsauer et al. (2020)]): softmax, sparsemax, $\alpha$-entropymax.
    - Similarly, the nonparametric framework for MHNs https://arxiv.org/abs/2404.03900 is also omitted, even though it covers many MHN-attention correspondence: linear, sparse, top-K, random feature...etc.
    - similar efforts from Andre Martins' group are also omitted, e.g. https://arxiv.org/abs/2402.13725 (ICML 2024)

    Many of these works covered similar CCCP derivations, memory/energy bound, retrieval errors, noise robustness and associative retrieval task validations. I just checked, all of the above-mentioned works have been published in NeurIPS, ICML, or ICLR. Given that this paper builds upon the MHN framework, disregarding these prior works raises serious concerns about scholarly completeness and citation practice. It’s understandable to miss a few. It happens to all of us. But missing all of them is very strange.

    Also, for empirical validations, many of these works should serve as baselines. Yet this paper did none of them.




2. **novelty:**
    - At a high level, the method could be seen as a relatively straightforward combination of known components: taking the existing modern Hopfield network (which is essentially an attention module) and operating it in a hyperbolic embedding space. The novelty is more in execution than in conceptual breakthrough. One might argue that given hyperbolic attention networks existed https://openreview.net/forum?id=rJxHsjRqFQ and this work establishes a new Hopfield = attention correspondence. Otherwise (and probably even so), the step to hyperbolic Hopfield is not huge. The authors’ contributions lie in working out the math and demonstrating the effects, but the both the idea itself and the math may not feel surprising. This could be perceived as a minor conceptual advance (“apply known method A in context B”). However, the quality of execution somewhat mitigates this. It’s incremental but well done.

    - [minor, more like a personal opinion] using hyperbolic space to embed a large tree (or combinatorial problems or the "hierarchical problem") while keeping the dimension small is not new. it gives a continuous relaxation of a combinatorial problem (e.g tree) where distance correlates with depth. Yet it only makes the embedding space geometrically larger, not structurally smarter. all these are sort of well-known, why not, instead, embed precise discrete or symbolic structures into your memory model to achieve the same goals (modeling hierarchical data) strictly more compact and precise. It's methodologically more efficient and economic and more align with standard CS practice.

3. **limited scope and clarity of motivation:**

    - By design, HAMNs shine only in hierarchical settings. Yet it never gives a clear definition of what do they mean by "hierarchical data". Without such clarity, it's hard to parse why the method works better on these data/tasks.
    - this paper does not provide a precise explanation of the benefits of using the Minkowski inner product. It only offers a heuristic motivation. I can follow the reasoning because of my background in differential geometry and this area of research, but I doubt that general ML readers will find it equally accessible or convincing.
    - In tasks without a clear hierarchy, they yield little to no improvement (and in a few cases marginal drops). This is not a flaw per se. It’s exactly what the paper predicts, but it means the method’s impact is confined to certain problem types. For an audience seeking a generally superior memory network, the answer is: use hyperbolic only if you expect hierarchical structure. The paper could emphasize this scope more. As is, a reader might wonder: if my data isn’t tree-like, do I incur overhead for nothing? The answer from results is yes, overhead with no gain. So the practical utility is conditional. Remember, the overhead to compute "geometric" gradients/optimization is very large (see below)

4. **computational overhead:** The current implementation is significantly slower and more memory-hungry than the Euclidean counterpart. This is a concern for scaling up. The experiments used moderate-scale data (CIFAR-100, 50k instances & MoleculeNet tasks with up to ~10k molecules). If one were to apply HAMNs to very large memory sets or high-dimensional data, the runtime could be problematic. So, while not a conceptual weakness, the practicality in large-scale or real-time scenarios is a concern.

5. **implementation complexity:** Using HAMNs requires familiarity with Riemannian optimization. The method introduces significant complexity: one must handle exponential and logarithmic maps, maintain numerical stability near the boundary, and tune an extra hyperparameter (curvature $c$). This is more involved than a standard attention or memory layer. This could be a barrier for general users/readers. The paper and appendix provide guidance, but integrating this module might still be hard compared to a Euclidean one.

6. **failure cases anylsis:** The paper does not deeply analyze scenarios where HAMN underperforms or equals baseline. For instance, why did HAMN slightly lag on some tasks? Is it purely because no hierarchy = noise, or could it be that the hyperbolic optimization sometimes finds a suboptimal attractor (since Hopfield can get stuck in a local minimum that isn’t the correct memory)? In Hopfield networks, spurious attractors can occur. With hyperbolic, could there be more risk of weird local minima? They didn’t discuss these. This makes the validity of HAMN questionable.

7. **minor:**
    - there are duplicate reference entries for the same paper
    - besides above mentioned literature omissions, the authors should also clarify how Hyperbolic Attention Networks differ from and relate to HAMN. Currently, they include HypAttn as a baseline, but a direct discussion in the text would help readers understand the conceptual advancement. Emphasize why energy-based retrieval with manifold optimization offers an advantage over a direct hyperbolic attention mechanism.
    - The paper primarily tests classification tasks (or retrieval framed as classification). It doesn’t explore standard associative memory scenarios, e.g., pattern completion or denoising.
    - To further demonstrate HAMN’s utility, it would be great to test on other domains with deep hierarchies, such as: Knowledge Graph completion or taxonomy reasoning, Hierarchical clustering or few-shot classification with class hierarchies, and most interestingly Language tasks with latent hierarchy. These would broaden the impact and show the method in action where tree structure is intrinsic. While CIFAR-100 with WordNet-like label trees is a good start, tasks with naturally deep hierarchies (like WordNet itself) would solidify the claims.

**Questions:**

please also see above weakness

LLM disclaimer: I used LLMs to check (a few of) my claims, to understand some refs mentioned in the submission, and to polish my language.

**Details Of Ethics Concerns:**

No new concerns beyond standard MHN misuse risks.

---

> ### Author Response · Authors · 2025-11-20
> **Response to Reviewer 8WQQ (Part I)**
>
> We thank the reviewer for the detailed and insightful comments. Below we respond to each major concern and describe the changes in the revised version.
>
> # 1. Missing developments on modern Hopfield networks
>
> We agree that the original submission did not sufficiently cover recent MHN work. In the revision we
>
> - **Expanded Related Work** to explicitly discuss (i) kernelized / learnable similarities, (ii) sparse and structured MHNs based on sparse entropic regularizers and Fenchel–Young losses, and (iii) latent-structured MHNs for episodic-style memory.
> - State clearly that we fully adopt the **“MHN as a differentiable associative-memory layer”** viewpoint and build on that, rather than proposing yet another Euclidean MHN variant.
>
> Regarding the concern that *“replacing the Euclidean dot product is not new”*: we agree that many MHN papers already generalize the similarity to learnable operators or kernels. However, all these methods still operate in a **flat Euclidean space**: states and memories live in $\mathbb{R}^d$, and the kernel is a function of Euclidean vectors. In contrast, HAMNs define the entire Hopfield energy **intrinsically on a negatively curved manifold**:
>
> - The association term is the **Minkowski inner product**, which is strictly order-equivalent to hyperbolic distance (App. A.1) and is coupled with an **intrinsic geodesic regularizer** instead of an Euclidean norm.
> - We obtain **curvature-dependent capacity / margin results**, including a ball-packing–style upper bound driven by hyperbolic volume growth (Apps. B.1–B.3), and we empirically show on hierarchical CIFAR-100 that attractors organize radially by tree depth (App. E.1.2).
>
> To our knowledge, existing MHN work (including kernelized and entropy-regularized variants) does not provide a hyperbolic Hopfield energy on a curved manifold, nor a systematic Euclidean–vs–hyperbolic comparison on multi-level hierarchies. Our contribution is thus not “just another similarity kernel”, but a **geometry-aware Hopfield formulation** controlled by negative curvature and analyzed both theoretically and empirically.
>
> **On additional MHN baselines.**
> In principle, recent Euclidean MHN variants could be used as alternative Euclidean decoders. Our empirical question, however, is more focused: **given a fixed backbone and architecture, what happens when we move the Hopfield energy and dynamics from flat space to a negatively curved manifold?** For this reason we use the standard Hopfield/attention layer of Ramsauer et al. as the Euclidean MHN baseline, and concentrate our comparisons on hyperbolic decoders (HypAttn, HypNN).
>
> The cited MHN variants primarily introduce *alternative similarities or sparsity mechanisms* in $\mathbb{R}^d$ without changing the underlying curvature. Porting all of them to our WordNet / KG setups with fair re-implementations and hyperparameter searches would be a substantial amount of engineering and would mostly probe design choices orthogonal to our main “Euclidean vs. hyperbolic geometry” question. To keep the experiments focused and within space/time limits, we did not add these baselines; we now explicitly list the absence of these Euclidean MHN variants as a **limitation** in Sec. Limitations.
>
> # 2. Conceptual novelty: beyond “apply known method A in context B”
>
> The reviewer suggests that HAMNs might be seen as simply “running an existing MHN in a hyperbolic embedding space”. We respectfully disagree that our contribution is just “apply method A in context B”, for three reasons.
>
> **(a) Intrinsic hyperbolic Hopfield energy and dynamics.**
> In Euclidean space, the equivalence between MHNs and attention holds because the Hopfield update can be written *exactly* as a softmax attention rule. Hyperbolic attention layers, by contrast, are typically defined by replacing dot products with functions of geodesic distance and computing a Fréchet mean for each query; there is **no known global Hopfield energy on a negatively curved manifold** whose gradient flow reproduces this single-step update.
>
> Deriving HAMNs therefore requires constructing a genuinely **hyperbolic Hopfield energy** from scratch, rather than mechanically “lifting” the Euclidean MHN = attention correspondence to curved space. In other words, the Euclidean MHN–attention equivalence does not transfer by a simple reverse-engineering trick to hyperbolic manifolds.
>
> We do *not* claim to establish a new equivalence between Hopfield networks and attention. Instead, to clarify the relation between hyperbolic attention and HAMNs, we added a dedicated section *“Relation to hyperbolic attention decoders”* in the appendix (with a short pointer in the main text).

---

> ### Author Response · Authors · 2025-11-20
> **Response to Reviewer 8WQQ (Part II)**
>
> **(b) Systematic study of “hierarchy sensitivity”.**
> We do not merely compare “hyperbolic vs Euclidean” decoders on standard benchmarks; we design experiments specifically to test the *hierarchy* hypothesis:
>
> - On CIFAR-100 we construct **multiple label trees of different depths on the same images**, and show that HAMNs match Euclidean MHNs on flat / shallow trees but gain accuracy and cross-level consistency as depth increases.
> - In the new **WordNet hypernym prediction** and **WN18RR link prediction** experiments (Apps. E.4, E.5), we apply HAMNs to real taxonomies and knowledge graphs and again find that, when the label space has deep hierarchy, HAMNs compete with strong hyperbolic baselines and significantly outperform Euclidean MHNs.
>
> We view this theory+experiment program on hierarchy sensitivity as going beyond a simple “replace Euclidean with hyperbolic” exercise.
>
> **(c) Positioning of the paper.**
> We toned down the language in the abstract and conclusion to describe the contribution as a **geometry-aware associative memory module**, not a conceptual revolution. We now emphasize that HAMNs build on existing MHN and hyperbolic representation learning frameworks, and that our main innovations are:
>
> 1. A hyperbolic Hopfield energy with CCCP retrieval and clear geometric interpretation;
> 2. Margin and capacity analysis on Hadamard manifolds;
> 3. Consistent empirical evidence of hierarchy sensitivity on synthetic (reclustered CIFAR-100) and real hierarchies (WordNet, WN18RR).
>
> Our intent is to present a solid, incremental but non-trivial step within the MHN + hyperbolic NN ecosystem.
>
> # 3. “Hyperbolic tree embeddings are not new; why not discrete / symbolic memory?”
>
> We agree that embedding trees or combinatorial hierarchies in hyperbolic space is not new, and we place our work explicitly in that line of research. Our aim here is to design and analyze a **Hopfield-style energy and retrieval dynamics that live intrinsically on a negatively curved manifold**.
>
> Regarding the suggestion to use discrete / symbolic memories directly: in all our experiments the memory module operates on **continuous features** produced by standard encoders (CNNs, SBERT, KG encoders). For end-to-end training in modern neural systems, a fully differentiable retrieval layer is crucial so that gradients can propagate cleanly. Purely discrete/symbolic memories would typically require additional relaxations or sampling-based estimators, which substantially complicate optimization and engineering.
>
> In the revision, we explicitly add a **ontology encoder (+OntEuc)** in the WordNet experiment. This encoder first embeds the discrete label graph, and its outputs are then shared by *all* decoders. OntEuc improves every model, but **hyperbolic decoders still achieve the lowest hierarchical error**, showing that ontology embeddings (discrete priors) and hyperbolic associative retrieval are complementary rather than mutually exclusive.
>
> Moreover, in many practical settings we only have access to **partial, noisy, or implicit hierarchies** (e.g., weak supervision or multi-task setups). In those cases an exact symbolic tree may not exist, whereas a continuous hyperbolic memory can still learn a “soft” hierarchy from data.
>
> # 4. Clarifying “hierarchical data” and the role of Minkowski similarity
>
> To address concerns about the definition of hierarchical data and the benefit of Minkowski similarity, we:
>
> - Add a **formal definition of hierarchical datasets** in Sec. 2.2.5 (label DAG / rooted tree, depth, ancestors, and hierarchy-aware evaluation such as multi-level accuracy, shortest-path distance, cophenetic correlations).
> - Add App. A.1 on the **Minkowski inner product and hyperbolic distance**, proving that $-\langle u,v\rangle_L$ is a curvature-sensitive dissimilarity strictly monotone in $d_{\mathbb{H}}(u,v)$, and showing empirically that optimizing this energy induces a clear depth–radius correlation on CIFAR-100 (App. E.1.2).
>
> These additions turn the motivation from heuristic into formally grounded and experimentally supported.
>
> # 5. Computational cost and implementation complexity
>
> We agree that our current hyperbolic layer is slower and more memory-hungry than a Euclidean MHN. To make this explicit, we add **runtime and memory analyses** on CIFAR-100 and WN18RR (App. E.6), reporting FLOPs, wall-clock time, and peak GPU memory for MHN\_Euc, HypAttn, HypNN, and HAMNs. For knowledge graphs we also introduce a lighter, task-specific implementation (HAMNs\_Algo).
>
> In the expanded **Limitations** section we clearly state that our main contribution is improved *representation* for hierarchical data, not improved computational efficiency; closing the runtime/memory gap will require better hyperbolic kernels and fused implementations. Architecturally, our code mirrors the standard MHN layer, so in practice adopting HAMNs mainly amounts to “swapping the decoder”.

---

> ### Author Response · Authors · 2025-11-20
> **Response to Reviewer 8WQQ (Part III)**
>
> # 6. Failure cases and performance on “flat” tasks
>
> We agree that the original submission did not sufficiently analyze cases where HAMNs underperform or match Euclidean baselines. In the revision we add explicit discussions for two groups of experiments:
>
> - **MIL benchmarks (App. E.2).**
>   We now state that the mixed results on Elephant/Fox/Tiger are consistent with task structure: these are binary MIL tasks without an explicit multi-level taxonomy or ontology, and thus provide little exploitable hierarchical signal for hyperbolic geometry. In this regime HAMNs act as generic non-linear set aggregators, so we do not expect systematic gains over a strong Euclidean Hopfield baseline.
>
> - **Molecular property prediction (App. E.3).**
>   Similarly, on HIV/BACE/BBBP/SIDER we show that HAMNs reach SOTA AUC on BBBP and SIDER but are slightly worse than MHNs/RF/SVM on HIV and BACE. We explicitly attribute this to the essentially “flat” binary nature of these tasks, which do not expose hierarchical structure in labels or molecular graphs. The observed differences (typically $\approx 1$–$1.5$ AUC points) are within normal optimization/model-selection variability, and CCCP-based energy updates remain monotone and stable.
>
> Together with the hierarchical CIFAR-100 and WordNet/WN18RR results, these analyses clarify **when HAMNs help and when they merely match Euclidean MHNs**: on flat tasks they are reliable drop-in associative modules; once the data or label space exhibits explicit multi-level structure, they provide clear and stable gains.
>
> # 7. Standard associative-memory scenarios
>
> Our core scientific question is: **does geometry help associative retrieval on hierarchical data?** We therefore choose benchmarks whose label or graph space has explicit or controllable hierarchy and can be evaluated with hierarchy-aware metrics (multi-level accuracy, graph distance, cophenetic correlation).
>
> Classical pattern-completion / denoising tasks are central to Hopfield models but typically use “flat” evaluation metrics and do not provide explicit hierarchy, so they are less informative about our specific hypothesis. Nevertheless, we do include settings closer to associative pooling than plain classification, namely MIL (Tiger/Fox/Elephant). On these essentially flat problems, HAMNs behave as stable, competitive alternatives to Euclidean MHNs, with small bidirectional gaps and no pathological attractors. This suggests that our method does not break standard associative-memory use cases, while its main benefit appears precisely when clear hierarchy is present. We now state this positioning explicitly in the Discussion.
>
> # 8. Other domains with deep hierarchies
>
> We fully agree that additional domains with deep hierarchies would further demonstrate utility. In the revision we add:
>
> - **WordNet noun hypernym prediction**, treated as a deep hierarchical label prediction task;
> - **WN18RR link prediction**, treated as KG completion where edges such as `hypernym`, `hyponym`, `instance_hypernym`, etc. encode the same WordNet hierarchy.
>
> In both setups we evaluate MHN\_Euc, HypAttn, HypNN, and HAMNs, and in the WordNet experiment we place a shared **ontology encoder (+OntEuc)** in front of *all* decoders. Results show that (i) Euclidean MHNs are clearly weaker on these deep hierarchies; (ii) even with the same graph-aware label prior, hyperbolic decoders remain stronger; and (iii) HAMNs are competitive with other hyperbolic decoders and slightly better on WN18RR.
>
> We agree that exploring hierarchical clustering, few-shot classification with class hierarchies, and language tasks with latent hierarchies are promising directions, but we leave these for future work due to space and time constraints.
>
> ---
>
> - **Duplicate references.**
>   We have removed all duplicate bibliography entries and merged citations that referred to the same work.
>
> Besides the changes mentioned above, we also added:
> - App. A.2: Illustrative hyperbolic energy landscape;
> - App. E.1.1: analysis of how CIFAR-100 clustering reacts to hierarchy reshaping;
> - App. E.7: expanded ablations with clearer hyperparameter descriptions.
>
> We understand the reviewer’s concern that, on a first read, the paper may look like “just moving MHNs to hyperbolic space”. The revision aims to clarify that our goal is to propose a principled **hyperbolic Hopfield** framework—with intrinsic energy, retrieval dynamics, capacity analysis, and systematic validation on hierarchical data—rather than a purely conceptual breakthrough. We hope the revised version better conveys this intent and alleviates concerns about novelty and positioning.

---

> ### Comment · Reviewer_8WQQ · 2025-11-20
> **The moral of the story must reflect the omitted references**
>
> I see that you only included the omitted references to related work section?
>
> This is not acceptable. You should at least rewrite your introduction to reflect these key developments **in the area that your work builds upon**. Only through this, you can position you work in literature, factually, fairly and precisely. **All your claimed contributions must be contrasted against these prior studies for the reasons already stated in my initial review.**
>
> Before you do that, I do not see much point in proceeding. This paper fails to comply with basic academic credit attribution.
>
> Edit: to be precise, I believe it's necessary to discuss these works upfront because they constitute many developments in this paper. Also, this is not correct:
> > However, all these methods still operate in a flat Euclidean space: states and memories live in $\mathbb{R}^n$, and the kernel is a function of Euclidean vectors.
>
> In the UHop paper, the geometry induced on the original data (the memory set) can be arbitrarily “non-Euclidean.” Getting such non-euclidean metric seems to be the whole point of that paper. So you really should discuss those papers precisely.

---

> > ### Author Response · Authors · 2025-11-21
> > **We have completed the revisions to the abstract and introduction, and clarified the feature space used by U-Hop.**
> >
> > Thank you very much for your further feedback and for pointing out issues in how we attributed and positioned recent MHN developments.
> >
> > We agree that the modern Hopfield work our paper builds upon should be discussed earlier in the manuscript, and we have accordingly revised the abstract and Introduction in the new version.
> >
> > Regarding the sentence in our previous rebuttal — “all these methods still operate in a flat Euclidean space” — our intended meaning was as follows. U-Hop learns a feature map and a kernel-induced norm such that the induced metric on the finite memory set can deviate significantly from the original ℓ₂ geometry, and this metric is explicitly optimized to increase separation between memories. In this sense, the geometry on the data can indeed be highly “non-Euclidean.” However, from a differential-geometric point of view, the Hopfield feature space and retrieval dynamics still live in a (possibly infinite-dimensional) Hilbert feature space, i.e., a flat linear space equipped with an inner product, where “flat” means that the curvature is zero. From the perspective of curvature and Riemannian geometry, a Hilbert space is therefore the infinite-dimensional analogue of Euclidean geometry, rather than a non-Euclidean manifold.
> >
> > By contrast, our work moves the Hopfield energy and dynamics onto a Riemannian manifold of constant negative curvature, and studies how this intrinsic curvature affects capacity and hierarchical retrieval.
> >
> > We have clarified this distinction and our positioning with respect to prior MHN work in the revised manuscript. We again sincerely thank you for your valuable comments, and we would very much welcome any further suggestions or corrections.

---

> > ### Author Response · Authors · 2025-11-22
> > **On Including U-Hop as a Baseline**
> >
> > To make our comparison with related work well grounded, it is important to incorporate U-Hop as a baseline as soon as possible. Therefore, we have already added U-Hop to the 4-layer CIFAR-100 hierarchical classification experiments and to Appendix *Experiment 4: Hypernym Prediction on WordNet*.
> >
> > For U-Hop, we use the official implementation and strictly follow the two-stage training protocol described in the original paper: we first optimize the kernel parameters using the separation loss, and then train the Hopfield retrieval dynamics on top of the learned kernel. This setup allows U-Hop to be directly and fairly compared with MHN\_Euc and the various hyperbolic decoders.
> >
> > In the **4-layer hierarchical CIFAR-100 experiment**, which best showcases the hierarchy sensitivity of our model, U-Hop does **not** improve the flat accuracies over the standard Euclidean MHN baseline (MHN\_Euc) – its top/super/coarse/fine accuracies are slightly lower – but it achieves a higher hierarchy correlation. Nevertheless, it still lags behind the HAMNs when considering both flat accuracies and hierarchy-sensitive metrics, as shown in the revised table:
> >
> > |Model|top\_acc|super\_acc|coarse\_acc|fine\_acc|coph\_corr|
> > |-|-|-|-|-|-|
> > |**CIFAR-100-4-layer**||||||
> > |Backboneonly|87.51±0.73|72.68±1.85|60.02±1.02|47.23±0.77|0.7180±0.0230|
> > |HypAttn|90.13±0.48|78.23±0.48|67.74±0.93|54.50±0.78|0.6795±0.0143|
> > |HypNN|90.30±0.35|78.72±0.59|68.29±0.80|**55.93±0.88**|0.6046±0.0149|
> > |MHNs|89.39±0.29|76.97±0.44|65.56±0.42|49.37±0.57|0.5902±0.0218|
> > |U-Hop|88.62±0.57|75.46±0.58|62.46±0.97|45.44±0.94|0.7154±0.0680|
> > |**HAMNs(ours)**|**90.98±0.39**|**79.48±0.57**|**68.51±0.84**|53.49±1.05|**0.7184±0.0254**|
> >
> > This shows that even after strengthening the Euclidean side with a kernelized Hopfield variant that improves hierarchy correlation, our hyperbolic decoder HAMNs remains more effective on deeper label hierarchies when both accuracy and hierarchical consistency are taken into account.
> >
> > On the **WordNet noun hypernym prediction task** (Appendix E.4), which is the setting where the gap between Euclidean and hyperbolic decoders is most pronounced, U-Hop likewise cannot bridge this gap. Without ontology embeddings, the results are:
> >
> > |Model|Acc(%)|Avg.hier.dist.|
> > |-|-|-|
> > |MHN\_Euc|26.68|7.33|
> > |U-Hop|30.09|7.33|
> > |**HypAttn**|**52.46**|**6.99**|
> > |HypNN|47.54|5.25|
> > |HAMNs|52.20|4.78|
> >
> > When we prepend the Euclidean ontology encoder (+OntEuc), U-Hop+OntEuc achieves:
> >
> > |Model|Acc(%)|Avg.hier.dist.|
> > |-|-|-|
> > |MHN\_Euc+OntEuc|89.61|1.03|
> > |U-Hop+OntEuc|91.06|0.89|
> > |**HypAttn+OntEuc**|**96.36**|**0.36**|
> > |HypNN+OntEuc|93.63|0.63|
> > |HAMNs+OntEuc|96.24|0.37|
> >
> > In both tables, U-Hop yields a clear improvement over the original MHN\_Euc baseline, but still performs worse than the hyperbolic decoders in terms of hierarchical error.
> >
> > Overall, on both CIFAR-100 and WordNet, U-Hop clearly strengthens the Euclidean Hopfield baseline but does **not** overturn our main conclusion: even with a stronger kernelized Euclidean decoder, hyperbolic Hopfield methods remain superior on hierarchy-sensitive metrics for deeply hierarchical data. These new baselines therefore support the claims we made in our earlier rebuttal: whether MHN, U-Hop, or other related advances, these methods still operate on Euclidean manifolds with zero curvature, whereas our approach is, to our knowledge, the first to embed modern associative memory mechanisms into a Riemannian manifold of negative curvature.
> >
> > Due to space limitations in the main paper, we have not yet integrated all of the above results into the main body of the manuscript. For the 2-layer and 3-layer CIFAR-100 hierarchies, we are currently training the U-Hop baselines; once training is completed, we will add these results together with the 4-layer CIFAR-100 and WordNet results and update the paper accordingly. For the WN18RR link prediction experiment, adapting the official U-Hop implementation to knowledge-graph inputs would require time-consuming code reimplementation (the original code is designed for vector inputs rather than knowledge-graph triples), so we do not include a U-Hop baseline for this experiment.

---

> > > ### Comment · Reviewer_8WQQ · 2025-11-25
> > > **updates on promised experiments?**
> > >
> > > Dear authors,
> > >
> > > Thanks for your detailed responses. Do you have any update on this part of experiments? Is there any update of the codebase in the supplementary? Thanks!
> > >
> > > > we are currently training the U-Hop baselines; once training is completed, we will add these results together with the 4-layer CIFAR-100 and WordNet results and update the paper accordingly. For the WN18RR link prediction experiment, adapting the official U-Hop implementation to knowledge-graph inputs would require time-consuming code reimplementation (the original code is designed for vector inputs rather than knowledge-graph triples), so we do not include a U-Hop baseline for this experiment.

---

> > > > ### Comment · Reviewer_8WQQ · 2025-11-25
> > > > **unconvinced response**
> > > >
> > > > I find your clarifications regarding the contrast with existing work unconvincing.
> > > >
> > > > For example, could you clearly state what the provable advantages are of formulating the Hopfield energy and retrieval on a Riemannian manifold? Please respond in the most precise and concise manner, without being overly AI-greased.
> > > >
> > > >
> > > > > By contrast, our work moves the Hopfield energy and dynamics onto a Riemannian manifold of constant negative curvature, and studies how this intrinsic curvature affects capacity and hierarchical retrieval.
> > > >
> > > > > We have clarified this distinction and our positioning with respect to prior MHN work in the revised manuscript. We again sincerely thank you for your valuable comments, and we would very much welcome any further suggestions or corrections.
> > > >
> > > > Also, what is inherently problematic about a geometric model whose features live in a Euclidean space? One can easily imagine that such features are already superior. What I find puzzling is that you still have not explained why this geometric promotion (assuming what you did are correct) is a good idea. This is a serious issue. I have not even check your math yet.

---

> > > > > ### Author Response · Authors · 2025-11-25
> > > > > **concrete advantages**
> > > > >
> > > > > Thank you for the follow-up.
> > > > >
> > > > > Very briefly, for data with deep hierarchical structure (e.g., tree- or graph-like), we prove that under the same number of neurons and the same minimum separation between patterns, as the depth increases a hyperbolic Hopfield layer can accommodate significantly more well-separated memories than a Euclidean Hopfield layer, and can more reliably represent and distinguish complex hierarchical structures (see Apps. A and B and the experimental section for details).
> > > > >
> > > > > We do not believe that Euclidean feature spaces are inherently flawed. In all of our experiments, the encoder outputs standard Euclidean vectors in $\mathbb{R}^d$. HAMNs treat these vectors as tangent vectors at the origin and apply the exponential map; what changes is only the geometry of the retrieval layer, not the backbone or the raw features themselves.
> > > > >
> > > > > Our motivation for moving the Hopfield energy to a negatively curved manifold is that there is a geometric mismatch between a flat metric and a deep label hierarchy. Even if Euclidean features are already sufficiently discriminative locally, a flat metric still tends to compress classes from different levels of the hierarchy into similar radii, so that the Hopfield energy cannot distinguish between cases that are “close in the tree but very different in depth” and those that are “truly semantically similar”. In contrast, on a hyperbolic manifold we can place attractors corresponding to different depths at different radii while keeping large geodesic distances between unrelated branches, so that the retrieval dynamics can respect both semantic proximity and hierarchical depth at the same time.
> > > > >
> > > > > In summary, our goal is not to replace Euclidean features, but to equip them with a retrieval layer whose geometry is better aligned with the target label hierarchy. We further refine this discussion in the Introduction to make more explicit “why we do this”. Moreover, we have supplemented the experimental section and have already submitted the revised paper.

---

### Official Review · Reviewer_phhi · 2025-10-30

**Soundness:** 3
**Presentation:** 2
**Contribution:** 3
**Rating:** 8
**Confidence:** 4

**Summary:**

This theoretical paper presents a new way to model Hopfield networks in the hyperbolic embedding spaces. The rationale to move from Euclidean to hyperbolic spaces is to allow hierarchical concepts to be modeled such as those represented in graphs and trees.  While this benefit was also shown in an earlier NeurIPS 2017 paper, this paper applies those ideas in the context of Hopfield networks. The paper is largely theoretical and will benefit from illustrations indicating the benefit of doing this for the Hopfield networks. Experiments are also a bit sketchy and it appears that the implementation still needs some work.

**Strengths:**

The strength of the paper lies in the formulation itself for modeling hierarchical object storage and retrieval through Hopfield network formulations. The claim made is that associative memory mechanisms purely within Euclidean geometry may distort the underlying structural information during memory retrieval. This is backed up by experimental data.  The design of the energy function is novel and stable update is being assured.

**Weaknesses:**

Despite the promised of Hopfield networks, getting them to work on large scale datasets has been a problem both due to computational complexity and the metastability problems projecting the raw data. How much the use of a hyperbolic space alters that is not clear. Showing the value proposition though a visual example would be helpful in this case. In particular, the ability to discriminate between the patterns was a function of the shape of the energy landscape and how sharp the basins of attraction were. Will it be similar in the hyperbolic space or the manifold is smooth per level while still being distinct across hierarchy levels.

Also, newer ontology embeddings are capturing the depths of hierarchy better through contrastive formulations in Euclidean spaces. Would that change the observation and need to do this modeling?

**Questions:**

Is there any biological motivation for the hyperbolic formulation for Hopfield networks?

---

> ### Author Response · Authors · 2025-11-20
> **Response to Reviewer phhi (Part I)**
>
> We thank the reviewer for the careful and constructive feedback, and we appreciate the accept recommendation. Below we address the concerns about experimental completeness, computational cost / metastability, ontology embeddings, and biological motivation, and we summarize the changes in the revised version.
>
> # 1. Experiments perceived as sketchy
>
> **(a) Visualizing the energy landscape.**
> To directly answer the request for a “visual example” of how the hyperbolic space shapes attraction basins, we added a new appendix section with an energy plot (App. A.2).
> On a 2D Poincaré disk we place a three-level tree (top / coarse / fine) and plot the full HAMN energy $E(\xi)$ on a dense grid. The figure shows that
>
> - top-level attractors near the center induce **wide, smooth** basins;
> - mid-level attractors at intermediate radii induce **narrower** wells;
> - fine-level attractors near the boundary induce **many sharp, well-separated** basins.
>
> This illustrates that **negative curvature lets us allocate increasingly fine-grained attractors along the radius without fragmenting the global landscape**, clarifying how discrimination depends on geometry. App. E.1.2 further shows empirically that radial coordinates correlate with tree depth, while geodesic distance reflects branch similarity.
>
> **(b) Real hierarchical data beyond CIFAR.**
> To narrow the gap between theory and applications, we added two WordNet-based experiments.
>
> - In **WordNet noun hypernym prediction** (App. E.4), each input is a synset encoded by SBERT and the task is to predict its (immediate) hypernym. We compare MHN\_Euc, HypAttn, HypNN, and HAMNs, reporting top-1 accuracy and average graph distance in the hypernym–hyponym tree. MHN\_Euc performs very poorly: its accuracy is less than half that of the hyperbolic decoders and its average distance is much larger, indicating that a flat Euclidean energy cannot exploit the multi-level taxonomy. In contrast, the hyperbolic decoders achieve substantially higher accuracy and much smaller hierarchical error; HypAttn is slightly best, while HAMNs remain competitive.
>
> - In **WN18RR link prediction** (App. E.5), we adapt our decoder to the KG setting via a lightweight CCCP-based implementation **HAMNs\_Algo** and compare it to MHN\_Euc, HypAttn, and HypNN. On this knowledge-graph completion task, all three hyperbolic decoders significantly outperform MHN\_Euc on MRR and Hits@1/3/10, again confirming the benefit of hyperbolic geometry when the graph has strong hierarchical structure. HAMNs obtain the best overall scores, with small but consistent gains over HypAttn and HypNN.
>
> Taken together, these WordNet experiments show that our hyperbolic associative memory is a **practical, plug-and-play decoder** for real ontologies and knowledge graphs, not only for synthetic CIFAR hierarchies.
>
> # 2. Ontology embeddings vs. hyperbolic retrieval
>
> We fully agree that ontology-based label embeddings are important. In our view they are **complementary** to HAMNs rather than competing: contrastive / graph-based losses can be used to shape memory prototypes in any model, while our hyperbolic memory layer performs retrieval on top of these structured prototypes.
>
> In the revision we make this explicit by adding a **ontology encoder (+OntEuc)** in the WordNet hypernym experiment and placing it in front of **all** decoders (App. E.4). We train a small encoder on the WordNet graph, then freeze it and feed its output to MHN\_Euc, HypAttn, HypNN, and HAMNs.
>
> - OntEuc gives **large gains for all models**.
> - Nevertheless, **hyperbolic decoders remain clearly stronger** than MHN\_Euc.
>
> We now explicitly state that ontology embeddings and hyperbolic retrieval are complementary: the ontology encoder provides a graph-aware label prior shared by all methods, while the hyperbolic memory layer controls how well retrieval respects hierarchical geometry under this fixed prior.

---

> ### Author Response · Authors · 2025-11-20
> **Response to Reviewer phhi (Part II)**
>
> # 3. Computational cost and metastability
>
> We agree that classical Hopfield networks can face computational and metastability issues at scale. To clarify how HAMNs behave in practice, we added:
>
> 1. **Compute / performance analysis.**
>    The new discussion and appendix (App. E.6) report detailed FLOPs, parameter counts, runtime, and peak memory on CIFAR and WN18RR. The main conclusion is explicit: **with current GPU kernels, HAMNs use fewer FLOPs and parameters than Euclidean MHNs but are slower and consume more memory**, mainly due to the extra cost and memory traffic of hyperbolic operations. The new **Limitations** section summarizes this trade-off.
>
> 2. **Metastability.**
>    We do not claim to solve all metastability issues of Hopfield networks. However, in our experiments we **did not observe pathological metastable behavior**: CCCP updates guarantee monotone energy decrease, training curves under different seeds are smooth, and HAMNs converge stably on all datasets. This is now clearly stated in the appendix.
>
> Overall, our contribution is primarily a **representational** improvement rather than a breakthrough in computational efficiency: hyperbolic geometry increases sensitivity to hierarchical structure but does not magically remove runtime or memory constraints. The revised Limitations section makes this explicit.
>
> # 4. Biological motivation
>
> Our main goal is **a differentiable associative memory module for machine learning**, not a biologically realistic model. This scope is now emphasized in the Introduction and in the **Limitations** section.
>
> For readers interested in neural-computation links, App. A.6 offers optional analogies: softmax weights as competition–suppression circuits, Riemannian gradients as natural gradients, and the CCCP update as leaky integration with gain control on a low-dimensional manifold. We explicitly stress that these are **interpretive analogies only**, not claims about biological implementation.
>
> ---
>
> Finally, following comments from all reviewers, the revised version also adds:
> - App. A.1: a more detailed explanation of the Minkowski inner product;
> - App. B.1–B.2: proofs for energy-well separation and a one-step attraction basin;
> - App. E.1.1: an analysis of how CIFAR-100 clustering reacts to hierarchy reshaping;
> - App. E.7: expanded ablations with clearer hyperparameter descriptions.
>
> We hope these additions and clarifications adequately address your questions on experimental support, ontology embeddings, and biological motivation, and we thank you again for the insightful review and positive recommendation.

---

### Official Review · Reviewer_KiWD · 2025-11-01

**Soundness:** 1
**Presentation:** 1
**Contribution:** 1
**Rating:** 0
**Confidence:** 4

**Summary:**

This paper proposes a hyperbolic version of modern Hopfield networks. It is formulated under the hyperbolic metric with a focus on the Lorentz model. The paper first show the guarantees of energy convergence, next they give the definition of new hyperbolic Hopfield layers for deep learning. Their empirical results show their proposed models outperforms other baselines on several tasks.

**Strengths:**

1. The empirical results show the effectiveness of their proposed model.

**Weaknesses:**

1. A major contribution to computational models of associative memory is their biological plausibility, which I believe the paper is lacking.
2. Without the closed form of neuron updates, it is very difficult to provide any biological intuitions.
3. The math in this paper seems to be incorrect. In general, Hadamard DC programming cannot guarantee to converge to a stationary point. Existing literatures only guarantees to critical points at best.
4. Another main feature studying Hopfield networks is their recall error. To my understanding, as the paper uses polynomial kernels, the recall error will likely scale with separation. In other words, with larger $R$, the larger we have the recall error, which seems not intuitive of a memory model.
5. What would the size of the attractor basin be? This part should be justified as this concerns the retrievability of patterns.
6. The reference format for equations is problematic; many are presented as "eq Equation x".
7. The use of the \paragraph is not consistent, some with period some without.
8. I did not find the limitation section, which is essential for a research paper.

Overall, I think this paper requires major revision for both presentations and results.

**Questions:**

1. Why would Riemannian DC problems apply Euclidean CCCP? Can the authors justify this part?
2. How did the authors setup their experiments? Did you assume input data points are hyperbolic points? Or did you use an Exp map to first project it into hyperbolic.

---

> ### Author Response · Authors · 2025-11-13
> **Author Response: Clarifications and Planned Revisions**
>
> We thank the reviewers for their insightful feedback. We first clarify two points, then respond item by item with planned revisions.
>
> **Clarifications to the review summary** “focus on the Lorentz model”: our work is model-agnostic and applies to any negatively curved Riemannian manifold; we use the Poincaré ball in experiments. Lorentz only appears in theoretical equivalences and the similarity discussion (Appendix C). “energy convergence guarantees”: we provide a guarantee of **monotone energy decrease to a critical point** under sufficient conditions; We do not claim full-sequence convergence to one stationary point.
>
> # W1. Biological interpretability.
> Our goal is a differentiable associative memory module for ML, not a neurobiological model. Modern Hopfield networks are used as energy-based retrieval layers. We will make this scope explicit in the Introduction and Limitations: our focus is on how geometry and optimization affect hierarchical retrieval, not on biological interpretability. And expand Related Work (capacity, attention equivalence, modular retrieval).
>
> # W2. No neuron-level closed form ⇒ hard biological intuition.
> We provide a **closed-form per-iteration update**: on a Hadamard manifold, a Riemannian first-order expansion of the concave term plus minimization of a geodesically convex surrogate yields a **parallel-transport + exponential-map** step (main §3.3, Eqs. (8)/(10)/(12); proof App. A.2; pseudocode App. D.1). It is not a single-neuron circuit; we do not claim biological interpretability. We will add brief neural-computation analogies in the appendix as optional intuition.
>
> # W3. Hadamard DC lacks convergence to a stationary point.
> We do not claim full-sequence convergence to one stationary point. We show: (i) the energy decreases monotonically and converges; (ii) every accumulation point $ \xi^* $ satisfies the Riemannian first-order condition $ \nabla_{\!\xi}E(\xi^*)=0 $. This is the standard DC/CCCP guarantee of stationary limit points, not strong convergence. To avoid ambiguity, we will state: **“energy decreases monotonically; any accumulation point is stationary.”** Assumptions appear in App. A.2.
>
> # W4. Greater separation ⇒ greater recall error.
> This applies to polynomial-kernel MHNs, not to our energy. Our similarity is a monotone transform of hyperbolic geodesic distance via $-\cosh(\cdot)$, which *increases* the log-odds gap as separation grows; the softmax mass on the correct item rises exponentially and top-1 error *decreases* with geodesic margin. We will add a brief pointer in the main text and place full inequalities in App. B.
>
> # W5. Basin size.
> Basin size depends on minimal geodesic separation $\delta$ and temperature/step $(\theta,\eta)$. For sufficiently large $\delta$ and reasonable $\theta$, there exists a geodesic ball around the true memory that is **one-step invariant** under the Riemannian-CCCP update (the iterate stays inside and moves toward the target), giving a computable **lower bound**. By hyperbolic exponential volume growth, we also obtain an **exponential-type capacity upper bound**: larger “hierarchical radius/depth” allows more non-overlapping wells. We will add two propositions in App. B.
>
> # W6/7/8. Writing & formatting.
> We will fix presentation: unify citation style (“Eq. (x)”), correct `\paragraph` usage, and add a separate **Limitations** section (e.g., computational cost, dependence on hierarchy strength, convergence to critical points).
>
> # Q1. Why would Riemannian DC apply Euclidean CCCP?
> It does not. Our algorithm is a **Riemannian-CCCP** on a Hadamard manifold: at iterate $\xi^{t}$ we take the Riemannian first-order expansion of the concave part $E_{\mathrm{cave}}$, minimize a **geodesically convex** surrogate, and obtain the **closed-form** update (Eq. (10)) On Hadamard spaces, the squared distance is geodesically convex, hence the surrogate is convex and the step above is its unique minimizer. Crucially, “unique minimizer” refers to the per-step surrogate, not a global optimum of the original problem. Under a temperature bound $\theta\le \theta_{\max}(c,\mathrm{diam})$ that controls the (negative) Hessian of $E_{\mathrm{cave}}$ along geodesics—and with standard step-size conditions—the total energy $E=E_{\mathrm{cave}}+\tfrac12 d(\cdot,p)^2$ **decreases monotonically**, and **any accumulation point is critical (stationary)**. These assumptions and the formal statement/proof sketch are made explicit in App. A.
>
> # Q2. Do you map to hyperbolic space via Exp?
> Yes. Euclidean queries/memories are mapped to $\mathbb D_c$ via ToPoincaré/Exp at the origin; retrieval/updates occur entirely in hyperbolic space; outputs are mapped back via FromPoincaré if needed. If inputs already lie in $\mathbb D_c$, no Exp is applied (as implemented). Full pseudocode is in App. D.1; the *Implementation hints* reiterate this; training details are in the supplement.
>
> If further questions arise, we welcome continued discussion during rebuttal.

---

> > ### Comment · Reviewer_8WQQ · 2025-11-13
> > **hallucination in rebuttal?**
> >
> > Dear authors,
> >
> > Your response has clear hallucination. This reviewer (`KiWD`) mentioned nothing about global optimum...
> >
> > Could you please double-check everything carefully before submission?
> >
> > I also noticed similar issues elsewhere in your draft (which I previously chose not to flag). It is not very respectful to the reviewers who invested real time and effort into reading your outputs...

---

> > > ### Author Response · Authors · 2025-11-13
> > > **Response to the “Hallucination” Remark**
> > >
> > > Thank you for the reminder. You are right, but that was not what I originally intended to convey. I will revise immediately. As I am not proficient in English, the “hallucination” arose because I repeatedly used translation aids during back-and-forth translation, and—after exceeding the character limit—asked a large model to trim the text, which distorted my original meaning and I did not notice it in time.
> > >
> > > Allow me to explain: before trimming, I wrote:
> > >
> > > Most DC/CCCP papers guarantee that: (i) the objective energy decreases monotonically and converges; (ii) any accumulation point is a (Riemannian) stationary/critical point. This is **“subsequence convergence to stationary points”** or “limit points satisfy first-order necessary conditions,” which is not the same as “the whole sequence strongly converges to a single stationary point.”
> > >
> > > In the review, I interpreted “converge to a stationary point” as “the entire sequence must strongly converge to some stationary point”; thus I wrote that it “cannot guarantee convergence to a stationary point, only to critical points.” In standard optimization terminology, however, stationary point = critical point (∇E = 0). Therefore I believed the phrasing in the review conflated “converge to a stationary point” (strong convergence) with “every accumulation point is stationary” (first-order optimality of limit points).
> > >
> > > In our paper we did not claim that the entire sequence strongly converges to a particular stationary point. What we can prove is: the energy decreases monotonically, and any accumulation point is (Riemannian) stationary. We will replace the phrasing “converges to a stationary point” with the more conservative “any accumulation point is critical; energy decreases monotonically,” and add clarifications in Appendix A.2 (Hadamard setting, DC split, smooth concave part, energy bounded below, proper temperature/step).
> > >
> > > The sentence “We do not claim global optimum.” was intended to mean “we do not claim that the whole sequence strongly converges to a particular stationary point.” Repeated translation and model-based trimming distorted the meaning, and I failed to catch it—my apologies. I will try my best to avoid this issue. Other similar problems you found in the draft may also have arisen because I exceeded character limits when writing the main text; during shortening and translation some meanings may have been lost or distorted, which was hard for me to notice—please kindly point them out.

---

### Author Response · Authors · 2025-12-03
**Summary of Responses and Revisions**

We sincerely thank all four reviewers for their careful, in-depth, and constructive feedback. In response to the main concerns—namely that the experiments were not sufficiently complete / the evidence for hierarchical structure was not direct enough; that our positioning relative to modern Hopfield-network developments was unclear; that ontology embeddings might be more central; that computational overhead and (meta-)stability were under-addressed; and that the **biological motivation and paper scope** needed clarification—we made the following targeted revisions and additions.

### 1) Strengthening direct evidence that we  **capture hierarchical structure**
- Added an **energy-landscape visualization on a 2D Poincaré disk** (App. A.2) to illustrate how negative curvature induces qualitatively different attraction basins across hierarchy levels.
- Added geometric probing analyses (App. E.1.2), reporting **depth–radius correlations** and correlations between **tree distance / LCA depth** and **hyperbolic geodesic distance**, empirically supporting the claim that depth is encoded in radius while branch similarity is reflected by geodesic proximity.

### 2) Adding **real hierarchical data** experiments beyond CIFAR
- Added **WordNet noun hypernym prediction** and **WN18RR link prediction** (Apps. E.4–E.5), comparing Euclidean MHN, HypAttn/HypNN, and HAMNs (with a lightweight KG variant, **HAMNs\_Algo**), demonstrating a practical plug-and-play decoder on ontologies and knowledge graphs.

### 3) Clarifying that **ontology embeddings vs. hyperbolic retrieval** are complementary
- Introduced a shared **Euclidean ontology encoder (+OntEuc)** placed before *all* decoders in the WordNet experiment (App. E.4): ontology embeddings improve all models, while hyperbolic decoders remain stronger in hierarchy-consistency / hierarchical-error metrics.

### 4) Improving our positioning relative to **modern Hopfield** developments
- Expanded the Related Work (kernelized / learnable similarities, sparse and Fenchel–Young variants, latent-structured MHNs, etc.), and clarified in the Introduction that our contribution is not merely “changing a similarity,” but proposing an **intrinsic Hopfield energy and retrieval dynamics on a negatively curved manifold**.
- Added **U-Hop** as a stronger Euclidean baseline in key experiments (CIFAR-100 hierarchical classification and WordNet hypernym prediction), strictly following the official two-stage training protocol to strengthen the credibility of the comparisons.

### 5) Making the **compute/memory overhead and stability** story explicit
- Added systematic profiling (FLOPs / parameter counts / wall-clock runtime / peak GPU memory) (App. E.6) and stated clearly in **Limitations** that current hyperbolic operations introduce extra runtime and memory overhead.
- Adopted a more cautious stability statement: CCCP guarantees **monotone energy decrease** (and convergence to stationary accumulation points under our assumptions); empirically, training curves are smooth across seeds and we did not observe pathological metastable behavior.

### 6) Clarifying **biological motivation and scope**
- Made explicit in the **Introduction and Limitations** that our goal is a **differentiable associative-memory module for machine learning**, not a biologically realistic neural implementation.
- Added an appendix section on “Optional Neural Computation Analogies” (App. A.6)—e.g., softmax weights as competition–suppression, Riemannian gradients as natural gradients, CCCP as leaky integration with gain control—and **explicitly labeled these as interpretive analogies**, not biological claims.

### 7) Additional theory and writing revisions
- Added a formal definition of “hierarchical data” (main text Sec. 2.2.5); strengthened the exposition of the **Minkowski inner product–hyperbolic distance** relationship (App. A.1) and theory on attraction basins / capacity (Apps. B.1–B.3); added ablations on **curvature and stored-pattern count** (App. E.7); expanded analyses for “flat tasks” (MIL / MoleculeNet) where results are comparable or mixed (Apps. E.2–E.3); difference from hyperbolic attention (Apps. E.8); and fixed duplicate references and presentation inconsistencies.

**Overall**, this revision aims to make the paper more **verifiable, reproducible, and well-positioned**: HAMNs is a geometry-aware associative-memory decoder for hierarchical data, with an intrinsic hyperbolic energy, stable retrieval dynamics, and systematic evidence on both controlled hierarchies and real ontologies / knowledge graphs.

---

### Note · Program_Chairs · 2026-01-17
**Submission Desk Rejected by Program Chairs**

The following references in this submission do not refer to real documents and/or have major errors in bibliographic information:

 Gregor Bachmann, Maximilian Nickel, and Mathias Niepert. Constant curvature manifolds for open set learning. Advances in Neural Information Processing Systems, 34, 2021.
Ilia Ovinnikov, Oleg Rebane, Richard Socher, and Caiming Xiong. Hyperbolic variational autoencoders. In International Conference on Learning Representations, 2021.
Satoshi Shimizu, Daisuke Toyama, and Yutaka Miyake. Hyperbolic neural networks++: A hyperbolic embedding method for graph neural networks. In Proceedings of the 30th ACM International Conference on Information & Knowledge Management, 202